# Communication-Efficient Federated Low-Rank Update Algorithm and its Connection to Implicit Regularization

## Abstract

Federated Learning (FL) faces significant challenges related to communication efficiency and performance reduction when scaling to many clients. To address these issues, we explore the potential of using low-rank updates and provide the first theoretical study of rank properties in FL. Our theoretical analysis shows that a client's loss exhibits a higher-rank structure (i.e., gradients span higher-rank subspaces of the Hessian) compared to the server's loss, and that low-rank approximations of the clients' gradients have greater similarity. Based on this insight, we hypothesize that constraining client-side optimization to a low-rank subspace could provide an implicit regularization effect while reducing communication costs. Consequently, we propose **FedLoRU**, a general low-rank update framework for FL. Our framework enforces low-rank client-side updates and accumulates these updates to form a higher-rank model. We are able to establish convergence of the algorithm; the convergence rate matches FedAvg. Experimental results demonstrate that FedLoRU performs comparably to full-rank algorithms and exhibits robustness to heterogeneous and large numbers of clients.

## 1 Introduction

Federated learning (FL, (McMahan et al., 2017)) is a collaborative learning framework designed to enhance privacy preservation by training models on clients' local data without sharing raw information. Nevertheless, it trades off some performance compared to centralized learning, largely due to communication overhead (Zheng et al., 2020) and heterogeneity (Ye et al., 2023; Kairouz et al., 2021). These limitations are further magnified when scaling to large client populations or training large language models on edge devices, where resource and data heterogeneity not only exacerbate communication costs but also complicate inter-client regularization (Ye et al., 2024). To address the two main challenges of communication overhead and performance reduction with increasing local clients in FL, we analyze the rank nature of loss landscape in FL and leverage low-rank updates.

There has been substantial research focusing on low-rank characteristics in centralized learning. By low rank, we refer to gradients spanning a low rank subspace of Hessian at any weights or the weight matrix being of the form $AB$ where the number of columns of $A$ is low. Methods such as LoRA (Hu et al., 2021), DyLoRA (Valipour et al., 2022), and QLoRA (Dettmers et al., 2024) utilize this scheme to decrease the number of trainable parameters, thus conserving memory and computational resources. Further observations (Huh et al., 2021; Ji & Telgarsky, 2018) indicate that over-parameterized models tend to find low-rank solutions, which provide implicit regularization effects.

However, the spectral properties of the loss landscape in FL remain under-explored. Herein, we first analyze the difference in the stable rank—defined as the squared ratio of the Frobenius norm to the spectral norm—between client Hessians and the server Hessian of any weights, discovering that a client exhibits a higher-rank structure. We also show that low-rank approximations of local gradients align better in direction than their full-rank counterparts. Based on this insight, we hypothesize that the client's higher-rank Hessian amplifies cross-client discrepancies, and that restricting client-side updates could offer both implicit regularization and reduced communication costs.

To address this, we propose the Federated Low-Rank Updates (FedLoRU) algorithm, which mitigates communication overhead and accommodates many clients through low-rank updates. FedLoRU factorizes client-side update matrices into $\boldsymbol{A}$ and $\boldsymbol{B}$ and applies iterative optimization to these low-rank factorized matrices. Clients and the server share the factorized matrices, which the server then aggregates. Matrices $\boldsymbol{A}$ and $\boldsymbol{B}$ are being communicated between the clients and server, rather than the much larger matrix $\boldsymbol{AB}$. To make the model's weight rank high, FedLoRU successively accumulates low-rank matrices. We also generalize the low-rank update strategy within federated learning for various heterogeneous settings. Our comprehensive approach underscores the potential of low-rank updates not only to enhance communication efficiency but also to impose implicit regularization.

In summary, this work presents the following principal contributions.

1. We provide, to our knowledge, the first theoretical study of spectral characteristics of client and server loss landscapes in FL. We show that, under stochastic sampling and a sufficiently large model, the stable rank of the Hessian of the loss function increases with smaller sample sizes.

2. Building on this analysis, we show that restricting local optimization to low-rank subspaces can improve cross-client alignment (implicit regularization effect), and we empirically find that the benefit of low-rank local training becomes more pronounced as the number of clients increases.

3. Motivated by these insights, we propose FedLoRU, which combines low-rank client-side optimization with periodic accumulation of low-rank updates to recover higher-rank global expressiveness while retaining communication efficiency. We rigorously show that its convergence rate is asymptotically equivalent to that of classical FedAvg. Moreover, we derive variants of FedLoRU for personalization and model heterogeneity settings.

4. Empirical results demonstrate that, on average, FedLoRU improves state-of-the-art communication-efficient federated learning algorithms on a variety of datasets, including LLM fine-tuning, and exhibits superior performance as the number of clients increases.

## 2 Related Work

**Communication-Efficient Federated Learning**  Extensive research has addressed communication challenges in FL (Shahid et al., 2021). FedPAQ (Reisizadeh et al., 2020) and AdaQuantFL (Jhunjhunwala et al., 2021) employ quantization to reduce the precision of weights. Meanwhile, pruning techniques are applied to remove less important weights, as seen in Fed-Dropout (Caldas et al., 2018) and FedMP (Jiang et al., 2023), with recent advancements like FedADSR (Wang et al., 2025) further optimizing this approach through dynamic, layer-wise adaptive pruning based on weight importance. Since quantization and sparsification do not alter the core network structure, they can be easily combined with other algorithms (e.g., FedLoRU) to reduce communication overhead.

In contrast, model compression techniques modify the model structure by compressing it before communication and restoring it afterward. FedDLR (Qiao et al., 2021) uses low-rank approximation for bidirectional communication but reverts to the full model for local training. FedHM (Yao et al., 2021) compresses only during server-to-client communication, where clients train factorized low-rank models that are aggregated by the server. Although both methods reduce communication overhead, their server-side compression can lead to performance degradation. While a recent work (Li et al., 2025) attempts to address this by optimizing low-rank decomposition and aggregation strategies to preserve model utility, they still struggle with reconstruction trade-offs. To intrinsically mitigate potential information loss, we focus on client-side factorization, avoiding compression processes.

**Low-rank nature of centralized and federated learning**  Numerous studies (Gur-Ari et al., 2018; Li et al., 2018; Sagun et al., 2016) assert that deep learning training inherently possesses a low-rank nature. Low-Rank Adaptation (LoRA, Hu et al. (2021)) is a representative algorithm that leverages this low-rank characteristic for fine-tuning by freezing pre-trained weights and applying low-rank updates via the decomposition $\boldsymbol{W} = \boldsymbol{W}_0 + \boldsymbol{AB}$, where $\boldsymbol{W}_0 \in \mathbb{R}^{m \times n}$, $\boldsymbol{A} \in \mathbb{R}^{m \times r}$, $\boldsymbol{B} \in \mathbb{R}^{r \times n}$, $r \ll m, n$. However, effectively

leveraging the low-rank structure in pre-training remains a challenge, as the weights do not inherently exhibit a low-rank nature (Yu & Wu, 2023; Zhao et al., 2024). To address this, ReLoRA (Lialin et al., 2023) seeks to achieve a higher-rank model by accumulating multiple low-rank updates, expressed as $\boldsymbol{W} = \boldsymbol{W}_0 + \sum_{i=1}^{M} \boldsymbol{A}_i \boldsymbol{B}_i$ where $\boldsymbol{A}_i \in \mathbb{R}^{m \times r}$, $\boldsymbol{B}_i \in \mathbb{R}^{r \times n}$.

In federated learning, some research has aimed to exploit the low-rank nature observed in centralized learning. LBGM (Azam et al., 2021) and FedLRGD (Jadbabaie et al., 2023) approximate gradients using past or sampled gradients, assuming gradients lie in a low-rank subspace. However, there is a noticeable gap in analyzing rank characteristics specific to federated learning. In the context of federated learning, there is a complex loss landscape involving multiple client-side and a single server-side optimization, and leveraging a low-rank structure needs to consider their respective rank structures. To our knowledge, no prior work has examined the rank structure in federated learning contexts without making very stringent assumptions. Our study is pioneering in addressing this gap, using analytical results and insights to develop a novel algorithm.

**Low-Rank Adaptation in Federated Learning**  Recent studies have studied the application of LoRA within federated learning frameworks. Notable algorithms, such as FedLoRA (Wu et al., 2024; Yi et al., 2023), FFALoRA (Sun et al., 2024), and Hyperflora (Lu et al., 2024), employ LoRA adapters to facilitate personalization. These methods apply low-rank adaptation to a pre-trained model during the local personalization training phase. On the other hand, other works (Zhang et al., 2023; Kuo et al., 2024; Cho et al., 2023) apply LoRA for fine-tuning within federated learning environments, with recent advancements like FedEx-LoRA (Singhal et al., 2024) ensuring exact aggregation by mitigating the update errors typically caused by standard federated averaging.

These approaches use only one low-rank matrix that restricts the model to a low-rank subspace. In contrast, we utilize multiple accumulated low-rank matrices allowing the model to achieve higher rank. Specifically, we extend the concept of LoRA by incorporating client-side low-rank updates and server-side accumulation to address the low-rank limitation of LoRA as well as the challenges posed by communication and client-server rank disparity. We also generalize the low-rank strategy within federated learning for both pre-training and fine-tuning, and for heterogeneous environments.

## 3 Low-Rank Characteristics in FL

In centralized learning, neural network losses exhibit a low-rank structure, indicating that the gradient lies within the subspace spanned by the Top-$k$ eigenvectors of the Hessian during training (Gur-Ari et al., 2018). While efforts have been made to utilize this low-rank structure to enhance federated learning algorithms, there is a lack of studies analyzing the rank structure of federated learning. In federated learning, the clients and server have distinct losses, resulting in different rank structures. Understanding these differing rank structures of client and server losses is crucial for developing low-rank-inspired algorithms tailored for federated learning. We provide, to our knowledge, the first theoretical characterization of this structure and show how low-rank local updates enhance inter-client gradient alignment.

**Notation and problem setup**  Suppose $\psi(\boldsymbol{x}, \boldsymbol{y})$ is a data generating distribution for an input-output pair $(\boldsymbol{x}, \boldsymbol{y}) \in \mathbb{R}^{d_x} \times \mathbb{R}^{d_y}$. We consider the problem of finding a prediction function $h^R(\cdot; \cdot) : \mathbb{R}^{d_x} \times \mathbb{R}^R \to \mathbb{R}^{d_y}$ parameterized by a $R$-dim weight vector $\omega^R \in \mathbb{R}^R$. Given a loss function $\ell(\cdot, \cdot) : \mathbb{R}^{d_y} \times \mathbb{R}^{d_y} \to \mathbb{R}$, the true risk is $\mathcal{L}_{\text{true}}(h^R, \omega^R) = \int \ell(h^R(\boldsymbol{x}; \omega^R), \boldsymbol{y}) d\psi(\boldsymbol{x}, \boldsymbol{y})$ and the corresponding true Hessian is $\boldsymbol{H}_{\text{true}}(h^R, \omega^R) = \nabla^2 \mathcal{L}_{\text{true}}(h^R, \omega^R)$. If $\mathcal{D}_N = \{(\boldsymbol{x}_i, \boldsymbol{y}_i)\}_{i=1}^{N}$ is a dataset generated from the distribution $\psi$, the empirical loss and Hessian for $\mathcal{D}_N$ are $f_N(h^R, \omega^R) = \sum_{(x,y) \in \mathcal{D}_N} \frac{1}{N} \ell(h^R(x; \omega^R), y)$ and $\boldsymbol{H}_N(h^R, \omega^R) = \sum_{(x,y) \in \mathcal{D}_N} \frac{1}{N} \frac{\partial^2}{\partial (\omega^R)^2} \ell(h^R(x; \omega^R), y)$.

We consider a random selection of $M$ samples without replacement from $\mathcal{D}_N$ to form a sub-dataset $\mathcal{D}_M \subseteq \mathcal{D}_N$. Let $f_M(h^R, \omega^R)$ and $\boldsymbol{H}_M(h^R, \omega^R)$ denote the loss and Hessian for the sub-dataset $\mathcal{D}_M$. In federated learning, $f_N$ can be considered as the loss that the server optimizes, while $f_M$ represents the loss of a local client assuming the homogeneous setting.

### 3.1 Higher Rank Nature of Clients in FL

In this section, we demonstrate that the local Hessian possesses a higher stable rank than the server's Hessian when the model size is large. This indicates that the loss landscape at a client is more complex than that of the server, which may contribute to divergence of local training.

**Stable rank**  To compare the rank properties of Hessians of a client and the server, we use the stable rank $\mathrm{srank}(\boldsymbol{A}) = \frac{\|\boldsymbol{A}\|_F^2}{\|\boldsymbol{A}\|_2^2} = \frac{\sum_{i=1}^n \sigma_i^2(\boldsymbol{A})}{\sigma_1^2(\boldsymbol{A})}$, where $n$ is the rank of matrix $\boldsymbol{A}$ and $\sigma_i(\boldsymbol{A})$ denotes its $i$-th singular value. Unlike traditional rank, which discretely counts non-zero singular values, the stable rank provides a continuous and more informative proxy, effectively capturing the low-rank nature of deep learning since stable rank is sensitive to the distribution of the singular values. This property is particularly useful in deep learning, where gradient descent trajectories are often dominated by a few large eigenvalues, and the subspace spanned by the corresponding eigenvectors critically influences training dynamics (Gur-Ari et al., 2018; Sagun et al., 2016; Sabanayagam et al., 2023). By emphasizing the contribution of large eigenvalues, the stable rank serves as a practical tool for quantifying the curvature of the loss landscape.

Moreover, the stable rank exhibits robustness to small perturbations in the Hessian. In practice, minor changes in model parameters or data points can lead to significant variations in the traditional rank, but these do not substantially affect the stable rank. This robustness ensures that stable rank provides consistent insights to the loss landscape, even under small variations in the training process. Additional discussion and details on stable rank are provided in Appendix A.

**Stable rank gap between client and server Hessians.**  For given $p, q \in \mathbb{N}$, let $\theta_1 > \cdots > \theta_p > 0 > \theta_{p+1} > \cdots > \theta_{p+q}$ be deterministic non-zero real numbers. Let $\Omega^R(\theta_1, \ldots, \theta_{p+q})$ be the set of parameter pairs $(h^R, \omega^R)$ whose *true* Hessian has eigenvalues $\theta_1 > \cdots > \theta_{p+q}$. Let $\bar{R}$ be the smallest integer for which $\Omega^{\bar{R}}(\theta_1, \cdots, \theta_{p+q})$ is non-empty. Let $\Omega(\theta_1, \cdots, \theta_k) = \bigcup_{R \geq \bar{R}} \Omega^R(\theta_1, \cdots, \theta_k)$, representing the union of $\Omega^R(\theta_1, \cdots, \theta_k)$ over all dimensions $R \geq \bar{R}$. We aim to show that the difference in the stable rank between the Hessians of the server and a client eventually becomes positive as dimension $R$ approaches infinity within the space of $\Omega(\theta_1, \cdots, \theta_k)$, which contains infinitely many $R$ for which $\Omega^R(\theta_1, \cdots, \theta_k) \neq \emptyset$, as proved in Appendix A.2.

For any $R \geq \bar{R}$ with $(h^R, \omega^R) \in \Omega^R$, we model the server and client Hessians as two decoupled additive perturbed model:

$$\boldsymbol{H}_N^R = \boldsymbol{H}_{\mathrm{true}}^R + \epsilon_N^R, \qquad \boldsymbol{H}_M^R = \boldsymbol{H}_{\mathrm{true}}^R + \epsilon_M^R. \tag{1}$$

Here, $\epsilon_N^R, \epsilon_M^R \in \mathbb{R}^{R \times R}$ are random error matrices associated with each Hessian. These matrices are assumed to be scaled according to $\epsilon_N^R = s_N X^R$, where $X^R \in \mathbb{R}^{R \times R}$ is a random real symmetric matrix where each element is independently drawn from a distribution with mean 0 and variance $\sigma^2/R$. The scaling factor $s_N = s(N)$ is defined as a monotonic decreasing function mapping $\mathbb{N}$ to $(0, 1)$. For simplicity in notation, we use $\boldsymbol{H}_N^R = \boldsymbol{H}_N(h^R, \omega^R)$ and $\boldsymbol{H}_{\mathrm{true}}^R = \boldsymbol{H}_{\mathrm{true}}(h^R, \omega^R)$ whenever the context is clear. A precise formalization of the problem framework, together with an in-depth discussion of its defining characteristics, is presented in Appendix A.

Next, we determine the limiting eigenvalues of the Hessians $\boldsymbol{H}_N^R$ in relation to the eigenvalues of $\boldsymbol{H}_{\mathrm{true}}^R$ as $R \to \infty$.

**Proposition 3.1** (Limiting eigenvalues of $\boldsymbol{H}_N^R$ (modified from Baskerville et al. (2022))). *Let $\boldsymbol{H}_N^R$ defined as in (1). If $\lambda_i(\boldsymbol{H}_N^R)$ denotes the $i$-th eigenvalue of $\boldsymbol{H}_N^R$, then for $i = 1, \cdots, p$, the following holds:*

$$\lambda_i(\boldsymbol{H}_N^R) \to \begin{cases} g_N^{-1}(\theta_i) & \text{if } g_N^{-1}(\theta_i) > U_N \\ U_N & \text{otherwise} \end{cases} \tag{2}$$

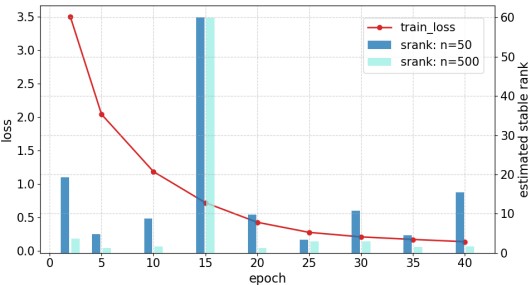 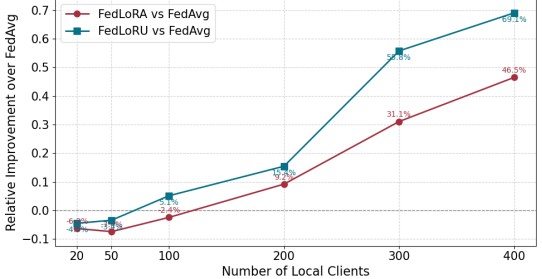

Figure 1: The estimated stable ranks of the Hessians are compared for dataset sizes of 50 and 500 (averaged over multiple runs). The estimated stable rank for the size of 50 consistently exceeds that of 500. For details of the experiment, see Appendix D.3

Figure 2: The relative difference in test accuracy between two algorithms is measured by the number of clients. The relative difference of $\text{Alg}_1$ to $\text{Alg}_2$ is defined as $\frac{\text{Alg}_1 - \text{Alg}_2}{\text{Alg}_1}$.

*as $R \to \infty$, and for $i = 0, \cdots, q-1$, we have*

$$\lambda_{R-i}(\boldsymbol{H}_N^R) \to \begin{cases} g_N^{-1}(\theta_{p+q-i}) & \text{if } g_N^{-1}(\theta_{p+q-i}) < L_N \\ L_N & \text{otherwise.} \end{cases} \tag{3}$$

*Here, $g_N^{-1}(\theta) = \theta + \frac{\sigma^2 s_N^2}{\theta}$, $U_N = 2\sigma s_N$, and $L_N = -2\sigma s_N$. In addition, for $p < i \leq R - q$, we have $\lambda_i(\boldsymbol{H}_N^R) \to \{L_N, U_N\}$.*

Convergence in our analysis is almost sure uniform convergence. The detailed proof is provided in Appendix A.3. In the following theorem, we demonstrate that a smaller dataset results in a higher stable rank in the limit except for the extremely ill-conditioned situation. Furthermore, given that modern neural network models typically possess a very large number of parameters, this finding is applicable to contemporary models.

**Theorem 3.2.** *Let $\boldsymbol{H}_N^R$ and $\boldsymbol{H}_M^R$ be the Hessians as defined in (1) and define $\theta_0 = \theta_1 \cdot \mathbf{1}_{|\theta_1| \geq |\theta_{p+q}|} + \theta_{p+q} \cdot \mathbf{1}_{|\theta_1| < |\theta_{p+q}|}$. Assume $\theta_0^2 \geq \sigma^2 s_M^2$. Then the difference in the limiting stable rank between $\boldsymbol{H}_N^R$ and $\boldsymbol{H}_M^R$ is positive and bounded below as follow*

$$\hat{\text{srank}}(\boldsymbol{H}_M) - \hat{\text{srank}}(\boldsymbol{H}_N) \geq \frac{s_M^2 - s_N^2}{g_M^{-1}(\theta_0)^2 g_N^{-1}(\theta_0)^2}$$

$$\left[ \sum_{j \in \mathcal{P}_N \cup \mathcal{Q}_N} 8\sigma^4 s_M s_N \left| \frac{\theta_0}{\theta_j} - \frac{\theta_j}{\theta_0} \right| + 4\sigma^2 B_N \left( \theta_0^2 - \frac{\sigma^4 s_M^2 s_N^2}{\theta_0^2} \right) \right], \tag{4}$$

*where $B_N = |\{i : \lambda_i(\boldsymbol{H}_N^R) \to U_N \text{ or } L_N\}|$, $\mathcal{P}_N = \{i \leq p : g_N^{-1}(\theta_i) > U_N\}$, and $\mathcal{Q}_N = \{i > p : g_N^{-1}(\theta_i) < L_N\}$. Furthermore, the lower bound decreases with $M$.*

This theorem characterizes the stable rank difference between $\boldsymbol{H}_M^R$ and $\boldsymbol{H}_N^R$ by showing that it is bounded below by a term proportional to $(s_M^2 - s_N^2)$. As $M$ decreases relative to $N$, this term increases. In the special case where $\theta_0^2 \leq \sigma^2 s_M^2$, the gap can become negative; however, this scenario arises only when the Hessian is extremely ill-conditioned, meaning that the largest singular value is extremely small. Under a typical scaling assumption such as $s_M = 1/M$, $\sigma^2 s_M^2$ remains sufficiently small in most practical settings, making such ill-conditioning unlikely. Consequently, except for highly degenerate settings, the stable rank difference between local and global Hessians remains strictly positive, and the magnitude of this difference grows as the size of each local dataset becomes smaller relative to the aggregate dataset. Our empirical results in Figure 1 further support this by demonstrating that smaller datasets exhibit higher estimated stable ranks.

### 3.2 Gradient Alignment Effect of Local Low-Rank Updates

In this section, we study how low-rank restrictions can improve cross-client gradient alignment in federated learning. We first analyze an idealized rank-$r$ truncation of each client gradient onto the leading eigendirections

of its local Hessian. This is not meant to claim that FedLoRU explicitly computes Hessian eigenspaces. Rather, the theorem identifies the rank-$r$ subspace that is optimal from the perspective of alignment, and serves as a reference object for understanding why low-rank local training can regularize client updates. Intuitively, as the approximation rank $r$ decreases, the components of each local gradient become more concentrated along the most significant directions of its Hessian, which in turn improves similarity across different clients.

While FedLoRU operates practically in the factor space spanned by the low-rank matrices $(A, B)$, this update space and our analyzed rank-$r$ eigenspace are closely linked through learning dynamics. Recent studies (Zhang et al., 2025; Xu et al., 2025) demonstrate that optimization trajectories naturally align with the top singular eigenvectors of the Hessian. Thus, the span of $(A, B)$ driven by optimization acts as a data-adaptive approximation of the ideal subspace identified by our theorem. This is further supported by our empirical observations (e.g., Appendix F.5), where low-rank updates initially exhibit lower alignment but quickly surpass full-rank updates as the factors adapt over the course of training.

**Limiting Eigenvector Transition**  Building on results from Benaych-Georges & Nadakuditi (2011), we know the limiting eigenvector transition. For $i \in \mathcal{P}_N \cup \mathcal{Q}_N$, let $v_i$ be the unit-norm eigenvector associated with the eigenvalue $\theta_i$ of $H_{\text{true}}^R$ and let $u_i$ be the corresponding unit-norm eigenvector of $\boldsymbol{H}_N^R$. Then for $j \in \{j \in \mathcal{P}_N \cup \mathcal{Q}_N : j \neq i\}$, we have

$$|\langle v_i, u_i \rangle|^2 \to 1 - \frac{\sigma^2 s_N^2}{\theta_i^2}, \quad |\langle v_j, u_i \rangle|^2 \to 0. \tag{5}$$

In other words, each limiting eigenvector of $\boldsymbol{H}_N^R$ lies in a cone around the corresponding eigenvector of $\boldsymbol{H}_{\text{true}}^R$. When $N$ is small, $\langle v_i, u_i \rangle$ remains farther from unity. This implies that the similarity between the eigenvectors of $\boldsymbol{H}_N^R$ and $\boldsymbol{H}_{\text{true}}^R$ is diminished in the regime of small $N$. Moreover, for a client operating with a dataset size $M < N$, the spectral similarity $\langle v_i, u_i \rangle$ becomes smaller than that of a client with a larger dataset. This phenomenon can degrade performance when a client holds very limited local data, as its local Hessian captures fewer reliable directions than one computed from a larger dataset. Further, we assume that the bulk eigenvectors are random vectors residing in the subspace orthogonal to that spanned by the edge eigenvectors as numerous studies (Anderson et al., 2010; Antti Knowles, 2013) have demonstrated.

**Gradient alignment**  We define the full-rank approximation of $\nabla f_N(h^R, \omega^R)$ with respect to $\boldsymbol{H}_N^R$ as $\nabla \hat{f}_{N,\text{full}}(h^R, \omega^R) = \sum_{i=1}^s \partial_{u_i} f_N(h^R, \omega^R) u_i$, where $u_1, \ldots, u_s$ are eigenvectors of $\boldsymbol{H}_N^R$ associated with the eigenvalues $\theta_1, \ldots, \theta_s$, ordered by magnitude, and $\partial_u f_N(h^R, \omega^R)$ is the directional derivative of $f_N(h^R, \omega^R)$ with respect to $u$. A rank-$r$ approximation then restricts this sum to only the top-$r$ eigenvectors as $\nabla \hat{f}_{N,r} = \sum_{i=1}^r \partial_{u_i} f_N(h^R, \omega^R) u_i$.

Given $K$ clients, each with a dataset of size $N$, we denote their corresponding Hessians by $\boldsymbol{H}_N^{(k)}$ for $k \in \{1, \ldots, K\}$. Let $C_{N,r}^R(k_1, k_2) = \cos\left(\nabla \hat{f}_{N,r}^{(k_1)}, \nabla \hat{f}_{N,r}^{(k_2)}\right)$ be the cosine similarity between the rank-$r$ approximations of the gradients of clients $k_1$ and $k_2$.

**Theorem 3.3.** *For any $r \in \mathbb{N}$ and $k_1, k_2 \in \{1, \ldots, K\}$ with $k_1 \neq k_2$,*

$$\left| \mathbb{E}\left[ C_{N,r}^R(k_1, k_2) - C_{N,r+1}^R(k_1, k_2) \right] - g_N(r) \right| \to 0 \tag{6}$$

*as $R \to \infty$, where $g_N(r)$ is strictly positive and expressed in the proof in Appendix A.5.*

According to Theorem 3.3, the expected cosine similarity between two clients' rank-$r$ gradient approximations decreases as $r$ increases for large $R$. Specifically, once $r$ is large enough to include all dominant directions, adding an additional component contributes random noise from the bulk eigenvectors, thereby reducing the directional alignment. For small $r$, incorporating the $(r+1)$-th principal direction also reduces similarity, because although it remains more critical than the bulk noise directions, it contributes less universally aligned signal than the top-$r$ directions.

**Scope of the Theoretical Model**  The analysis in this section considers a homogeneous setting in which a client dataset is modeled as a random sub-dataset of the full dataset. Although this does not directly model

non-IID client heterogeneity, it provides a useful baseline mechanism for understanding how limited local data can affect the local loss landscape.

# 4    Federated Low-Rank Update

Theorems 3.2 and 3.3 together reveal a rank paradox in FL: each client faces a higher-rank landscape, but better cross-client alignment arises when local update rank is low. Further, many works, e.g., Hu et al. (2021) and Ren et al. (2024), show that low-rank training mitigates overfitting on small datasets. Our theoretical analysis suggests that local Hessians exhibit a more complex loss landscape, which can lead to potential gradient divergence across local clients. Low-rank updates help align client updates along shared directions, thereby reducing client discrepancies. Building on this insight, we propose FedLoRU, wherein client optimization is constrained to a low-rank subspace to enhance communication efficiency and improve client alignment, while the global model achieves a higher rank by accumulating those subspaces over time.

## 4.1    FedLoRU Algorithm

Consider a federated learning system with $K$ clients, where each client $k$ has its own loss function $f^{(k)}$ : $\mathbb{R}^{m \times n} \to \mathbb{R}$. The server aims to find a global model $\boldsymbol{W} \in \mathbb{R}^{m \times n}$ that minimizes the aggregated loss function $f(\boldsymbol{W}) = \sum_{k=1}^{K} p^{(k)} f^{(k)}(\boldsymbol{W})$, where $p^{(k)}$ is the weight of client $k$.

---

**Algorithm 1** FedLoRU.

---

**Require:** model $\boldsymbol{W}_0$, initial low-rank matrices $\boldsymbol{A}_0, \boldsymbol{B}_0$, scaling factor $\alpha$, accumulation cycle $\tau$, total round $T$
    **Initialize:** Server sends $\boldsymbol{W}_0$ to each client.
    **for** $t = 0, \cdots, T-1$ **do**
        Server selects participating clients $\mathcal{K}_t$ and distributes $\boldsymbol{A}_t, \boldsymbol{B}_t$ to clients in $\mathcal{K}_t$.
        **for** each client $k \in \mathcal{K}_t$ **do**
            Find $\boldsymbol{A}_{t,E}^{(k)}, \boldsymbol{B}_{t,E}^{(k)}$ by solving (7) starting from $\boldsymbol{A}_t, \boldsymbol{B}_t$.
            Send $\boldsymbol{A}_{t,E}^{(k)}, \boldsymbol{B}_{t,E}^{(k)}$ to the server.
        **end for**
        **Server aggregation:**
        $\boldsymbol{A}_{t+1} \leftarrow \sum_{k \in \mathcal{K}_t} p_t^{(k)} \boldsymbol{A}_{t,E}^{(k)}, \boldsymbol{B}_{t+1} \leftarrow \sum_{k \in \mathcal{K}_t} p_t^{(k)} \boldsymbol{B}_{t,E}^{(k)}$.
        **if** $(t+1) \bmod \tau = 0$ **then**
            Server distributes $\boldsymbol{A}_{t+1}, \boldsymbol{B}_{t+1}$ to all clients .
            Each client $k$ updates its local copy of the global model: $\boldsymbol{W}_{t+1} \leftarrow \boldsymbol{W}_t + \alpha \boldsymbol{A}_{t+1} \boldsymbol{B}_{t+1}$.
            Server re-initializes $\boldsymbol{A}_{t+1}, \boldsymbol{B}_{t+1}$.
        **else**
            Server and clients update $\boldsymbol{W}_{t+1} \leftarrow \boldsymbol{W}_t$.
        **end if**
    **end for**
    **Return:** $\boldsymbol{W}_T = \boldsymbol{W}_0 + \alpha \sum_{t=0:\ (t+1) \bmod \tau = 0}^{T-1} \boldsymbol{A}_{t+1} \boldsymbol{B}_{t+1}$.

---

At communication round $t$, a subset of clients $\mathcal{K}_t$ performs local training. We denote by $p_t^{(k)}$ the aggregation weight of client $k \in \mathcal{K}_t$ at round $t$, with $\sum_{k \in \mathcal{K}_t} p_t^{(k)} = 1$. Following the LoRA (Hu et al., 2021) approach[1], client $k$ freezes a local copy of the global model $\boldsymbol{W}_t$ and finds low-rank matrices by solving $\arg \min_{\boldsymbol{A}, \boldsymbol{B}} f^{(k)}(\boldsymbol{W}_t + \alpha \boldsymbol{A}\boldsymbol{B})$, where $\alpha$ is a fixed scaling hyperparameter. Specifically, local training is carried out by $E$ local gradient-descent steps on $\boldsymbol{A}$ and $\boldsymbol{B}$ for $i = 0, \ldots, E-1$:

$$
\begin{aligned}
\boldsymbol{A}_{t,i+1}^{(k)} &= \boldsymbol{A}_{t,i}^{(k)} \ - \ \eta \, \nabla_A \, f^{(k)}\big(\boldsymbol{W}_t + \alpha \, \boldsymbol{A}_{t,i}^{(k)} \boldsymbol{B}_{t,i}^{(k)}; \xi_{t,i}^{(k)}\big), \\
\boldsymbol{B}_{t,i+1}^{(k)} &= \boldsymbol{B}_{t,i}^{(k)} \ - \ \eta \, \nabla_B \, f^{(k)}\big(\boldsymbol{W}_t + \alpha \, \boldsymbol{A}_{t,i}^{(k)} \boldsymbol{B}_{t,i}^{(k)}; \xi_{t,i}^{(k)}\big),
\end{aligned}
\tag{7}
$$

---

[1]While we use a low-rank factorized model, alternatives like LoKr (Edalati et al., 2022) or LoHa (Hyeon-Woo et al., 2021) can be employed, differing only in the factorization scheme but based on the same principles.

where $\nabla f^{(k)}(\boldsymbol{W}; \xi^{(k)})$ is a stochastic gradient evaluated on a randomly sampled subset $\xi^{(k)}$ from client $k$'s local data. After each round, the server collects $\boldsymbol{A}_{t,E}^{(k)}$ and $\boldsymbol{B}_{t,E}^{(k)}$ and aggregates them: $\boldsymbol{A}_{t+1} = \sum_{k \in \mathcal{K}_t} p_t^{(k)} \boldsymbol{A}_{t,E}^{(k)}$, $\boldsymbol{B}_{t+1} = \sum_{k \in \mathcal{K}_t} p_t^{(k)} \boldsymbol{B}_{t,E}^{(k)}$. After aggregation, the server sends $\boldsymbol{A}_{t+1}$ and $\boldsymbol{B}_{t+1}$ to clients, who continue local training using these matrices as starting $\boldsymbol{A}$ and $\boldsymbol{B}$.

Unlike LoRA, FedLoRU periodically accumulates low-rank updates into the global model after aggregation to achieve a higher-rank global model. This accumulation mechanism is inspired by ReLoRA (Lialin et al., 2023), which increases model expressiveness through successive low-rank updates in centralized training; FedLoRU adapts this idea to federated learning by combining periodic server-side accumulation with client-side low-rank optimization . Clients subsequently update their local copies of the global model by $\boldsymbol{W}_t \leftarrow \boldsymbol{W}_t + \alpha \boldsymbol{A}_t \boldsymbol{B}_t$. When low-rank updates are accumulated every $\tau$ rounds from the initial global model $\boldsymbol{W}_0$, the final global model at round $T$ is $\boldsymbol{W}_T = \boldsymbol{W}_0 + \alpha \sum_{t=0:\,(t+1) \bmod \tau = 0}^{T-1} \boldsymbol{A}_{t+1} \boldsymbol{B}_{t+1}$.

We average each matrix $\boldsymbol{A}$ and $\boldsymbol{B}$ individually, but acknowledge that alternative low-rank approaches, such as freezing one factor or alternating updates, may offer different mathematical justifications. In practice, however, we have found that our chosen scheme is the most effective among them. Furthermore, since our primary objective is to demonstrate the practicality and implicit regularization effect of low-rank updates, we defer a deeper investigation of these alternatives to future work.

**FedLoRU for Fine-tuning** For fine-tuning tasks, FedLoRU retains the series of accumulated low-rank matrices alongside the frozen pre-trained model. Although storing multiple low-rank matrices requires more memory than storing a single matrix, their combined size remains significantly smaller than that of the original model. This enables a modular, plug-and-play approach where low-rank matrices can be easily integrated with the pre-trained model. Consequently, FedLoRU maintains the same level of flexibility and extensibility as LoRA. The detailed fine-tuning algorithm is provided in Appendix C.1.

**Practical Advantages** FedLoRU enables training a higher-rank global model alongside low-rank local updates. With each accumulation of low-rank update matrices, the global model's rank is incrementally enhanced, enabling the initiation of new learning phases. Moreover, by constraining updates to a low-rank subspace, FedLoRU implicitly regularizes local training, aligning local updates along major directions and reducing client divergence. Such regularization addresses one of the most significant challenges in federated learning: performance degradation when scaling to many clients.

FedLoRU also reduces communication overhead from $Kmn$ to $Kr(m+n)$ when $r \ll m$ or $r \ll n$. Additionally, since no compression process is involved, there is no additional computation compared to conventional compression-based communication-efficient federated learning algorithms.

## 4.2 Convergence Analysis

We present a convergence result for the proposed FedLoRU algorithm; full details of the technical assumptions and proof are provided in Appendix B. To facilitate the convergence analysis of the proposed method, we make the following standard assumptions.

**Assumption 4.1.** There exist constants $L, G, C_A, C_B, \sigma^2 > 0$ such that for any client $k$, any weight matrices $\boldsymbol{W}, \boldsymbol{W}'$, communication round $t$, and local round $i$, the following hold:

$$\|\nabla f^{(k)}(\boldsymbol{W}) - \nabla f^{(k)}(\boldsymbol{W}')\|_F \leq L \|\boldsymbol{W} - \boldsymbol{W}'\|_F, \tag{A.1}$$

$$\mathbb{E}\big\|\nabla f^{(k)}(\boldsymbol{W}_{t,i}^{(k)})\big\|_F^2 \leq G^2, \tag{A.2}$$

$$\|\boldsymbol{A}_{t,i}^{(k)}\|_F \leq C_A, \quad \|\boldsymbol{B}_{t,i}^{(k)}\|_F \leq C_B, \tag{A.3}$$

$$\mathbb{E}\big\|\nabla f^{(k)}(\boldsymbol{W}; \xi^{(k)}) - \nabla f^{(k)}(\boldsymbol{W})\big\|_F^2 \leq \sigma^2, \tag{A.4}$$

$$\mathbb{E}\big[\nabla f^{(k)}(\boldsymbol{W}; \xi^{(k)})\big] = \nabla f^{(k)}(\boldsymbol{W}), \tag{A.5}$$

where $\boldsymbol{W}_{t,i}^{(k)} = \boldsymbol{W}_t + \alpha \boldsymbol{A}_{t,i}^{(k)} \boldsymbol{B}_{t,i}^{(k)}$ denotes the model parameters of client $k$ evaluated at local step $i$ in communication round $t$.

Showing convergence of LoRA-type algorithms is challenging because factorization does not preserve the smoothness or convexity of the original objective function. Convergence analyses of LoRA-type algorithms (e.g., Dec-LoRA(Ghiasvand et al., 2025), COLA(Xia et al., 2024), RAC-LoRA(Malinovsky et al., 2024), FedSA-LoRA(Ghiasvand et al., 2025)) address this through algorithmic design (e.g., freezing one of two low-rank matrices, or performing only one local step) often sacrificing performance or making strong assumptions (e.g., the descent lemma of the LoRA step). By contrast, our analysis establishes convergence under standard assumptions, without resorting to these concessions. The following theorem confirms that FedLoRU attains the same $\mathcal{O}(T^{-1/2})$ convergence rate in the reparameterized low-rank variables as classical FedAvg (Wang et al., 2020) when the step size is chosen as $\eta = \Theta(T^{-1/2})$.

**Theorem 4.2** (Convergence of FedLoRU). *Let Assumptions (A.1)-(A.5) hold and let $\{(\boldsymbol{W}_t, \boldsymbol{A}_t, \boldsymbol{B}_t)\}_{t=0}^T$ be the iterates produced by FedLoRU. For any fixed step size $\eta > 0$, we define $\widetilde{\boldsymbol{W}}_t$ as the effective model at the start of round $t$ and $\Delta_0 = f(\widetilde{\boldsymbol{W}}_0) - f^\star$, where $f^\star$ is the optimal value of $f$. Then*

$$\frac{1}{T}\sum_{t=0}^{T-1}\mathbb{E}\Big[\|\nabla_{\boldsymbol{A}} f(\widetilde{\boldsymbol{W}}_t)\|_F^2 + \|\nabla_{\boldsymbol{B}} f(\widetilde{\boldsymbol{W}}_t)\|_F^2\Big] \;\leq\; \frac{4\Delta_0}{3\,\eta T E} \;+\; K_1\eta \;+\; K_2\eta^2,$$

*where the positive constants $K_1, K_2$ depend only on $(C_A, C_B, G, L, E, \sigma^2)$ (see Appendix B). Choosing the step size $\eta = \Theta(T^{-1/2})$ yields*

$$\min_{0\leq t\leq T-1}\mathbb{E}\big[\|\nabla_{\boldsymbol{A}} f(\widetilde{\boldsymbol{W}}_t)\|_F^2 + \|\nabla_{\boldsymbol{B}} f(\widetilde{\boldsymbol{W}}_t)\|_F^2\big] = \mathcal{O}\big(T^{-1/2}\big).$$

## 5 Experiments

### 5.1 Experiment setup

**Datasets and Baseline Algorithms** We evaluate our proposed algorithms on five datasets: Fashion MNIST (Xiao et al., 2017), CIFAR-10, CIFAR-100 (Krizhevsky & Hinton, 2009), Alpaca (Taori et al., 2023), and GLUE (Wang et al., 2018). ResNet-10 and ResNet-18 (He et al., 2016) are used for the image datasets, LLaMA2-3B (Touvron et al., 2023) is used for fine-tuning on Alpaca, and TinyLlama 1.1B (Zhang et al., 2024) is used for GLUE (CoLA, SST-2, STS-B, MRPC). For the image datasets, we allocated 10,000 samples each for the validation and test sets, while for the Alpaca dataset, we partitioned the data into training, validation, and test sets consisting of 48,000, 2,000, and 2,000 samples, respectively. We compare FedLoRU with several benchmarks: FedAvg (McMahan et al., 2017), the standard federated learning algorithm that trains full-rank models; FedLoRA (Zhang et al., 2023), which trains low-rank modules without accumulating low-rank updates; and FedHM (Yao et al., 2021), the prior state-of-the-art in communication-efficient federated learning.

**Implementation** During pre-training on the image datasets, we vary the number of clients from 20 to 400, sampling 50% of clients per round, as is standard in the FL literature, with each client training for 5 local epochs. For instruction tuning on Alpaca (LLaMA2-3B), we use 10 clients with a 50% participation rate and 1 local epoch per round. For GLUE (TinyLlama 1.1B), we use 5 clients with full participation (IID split), use 1 local epoch per round, and train for 30 communication rounds; FedLoRU uses accumulation cycles of 5 or 10 rounds. Learning rates and accumulation cycles are selected via grid search, and different rank configurations are tested for FedHM, FedLoRA, and FedLoRU. In fact, while we use FedAvg as the training scheme, FedLoRU techniques can be easily integrated into other federated learning schemes such as FedAdam and FedAdagrad (Reddi et al., 2020). Model parameters are initialized following LoRA best practices, Kaiming initialization (He et al., 2015) for $\boldsymbol{A}$-matrix, and zeros for $\boldsymbol{B}$-matrix. For full details of the implementation, including the selection of parameters such as $\alpha$, $\tau$, and $T$, as well as their sensitivity, see Appendix D. We run each setting 3 times and the numbers reported in the tables are averages. In the statistically heterogeneous setting, we generate disjoint non-IID client data using a Dirichlet distribution, $\text{Dir}(\psi)$, with a concentration parameter $\psi$ set to 0.5, as described in Hsu et al. (2019).

Table 1: Test accuracy comparison with different communication-efficient federated learning methods under various FL settings. The parameter ratio refers to the proportion of trainable parameters in the model compared to the full-rank model and it implies the rank.

(a) Fashion-MNIST

| Setting | IID - #clients=20 | | | IID - #clients=100 | | | NonIID - #clients=20 | | |
|---|---|---|---|---|---|---|---|---|---|
| Param Ratio | 44% | 33% | 22% | 44% | 33% | 22% | 44% | 33% | 22% |
| FedLoRA | 91.22 | 90.29 | 90.15 | 88.63 | 88.14 | 88.01 | 73.89 | 74.00 | 73.19 |
| FedHM | 91.16 | 91.10 | **90.94** | **89.43** | **89.37** | **88.86** | 85.15 | **85.45** | **85.33** |
| FedLoRU | **91.25** | **91.16** | 90.59 | 89.01 | 88.88 | 88.37 | **85.33** | 80.02 | 80.17 |

(b) CIFAR-10

| Setting | IID - #clients=20 | | | IID - #clients=100 | | | NonIID - #clients=20 | | |
|---|---|---|---|---|---|---|---|---|---|
| Param Ratio | 41% | 31% | 21% | 41% | 31% | 21% | 41% | 31% | 21% |
| FedLoRA | 91.65 | 88.96 | 89.35 | 79.48 | 85.71 | 85.06 | 69.60 | 66.13 | 67.61 |
| FedHM | 90.76 | 90.32 | 90.77 | 81.41 | 81.58 | 82.12 | 70.55 | 66.39 | 65.48 |
| FedLoRU | **92.43** | **90.71** | **90.85** | **81.46** | **86.01** | **86.10** | **75.19** | **69.71** | **67.88** |

(c) CIFAR-100

| Setting | IID - #clients=20 | | | IID - #clients=100 | | | NonIID - #clients=20 | | |
|---|---|---|---|---|---|---|---|---|---|
| Param Ratio | 41% | 31% | 21% | 41% | 31% | 21% | 41% | 31% | 21% |
| FedLoRA | 65.53 | 57.36 | 55.14 | 53.79 | 52.20 | 51.20 | 14.41 | 10.58 | 12.97 |
| FedHM | 59.43 | 58.40 | 58.52 | 43.35 | 41.84 | 41.62 | **16.88** | 15.04 | 14.13 |
| FedLoRU | **66.81** | **60.78** | **61.42** | **57.96** | **53.25** | **53.53** | 16.46 | **15.70** | **14.52** |

## 5.2 Performance Evaluation

**Performance of Pre-training**  We evaluate the Top-1 accuracy of models with varying parameter sizes in both IID and Non-IID scenarios across different federated learning configurations. Table 1 shows the performance of FedLoRU and baseline algorithms. The standard deviation for each setting is relatively small in the IID scenario, with a maximum value of 0.382. In contrast, the non-IID setting exhibits a relatively higher standard deviation, with a maximum of 0.785. However, these variations do not impact the overall comparison between the algorithms.

In our experimental evaluation, FedLoRU consistently achieves competitive or superior accuracy compared to FedAvg, whose results can be found in Appendix F. Although FedLoRU's accuracy is slightly lower than FedAvg's in most settings, the difference is minimal given the significant reduction in parameters, with at most a 5% decrease and typically only a 1-2% difference. Notably, in the CIFAR-10 and CIFAR-100 IID settings with 100 clients, FedLoRU surpasses FedAvg. Overall, FedLoRU achieves the best accuracy in 20 out of 27 cases and demonstrates improvements over FedHM ranging from -6% to 33.7%. Furthermore, FedLoRU consistently outperforms FedLoRA, underscoring that accumulated low-rank updates recover high-rank expressiveness while preserving the local-regularization advantage. The observed performance enhancement grows with the number of clients, matching our theory that low-rank constraints mitigate client-side overfitting and enhances inter-client gradient alignment. Additional evidence of alignment of low-rank local training is presented in Appendix F.5.

Additional baselines, $E=1$ stress tests, accumulation cycle sensitivity, and sparsification compatibility are provided in Appendix G. These results show that FedLoRU remains competitive with other full-rank baselines, continues to outperform low-rank alternatives even with $E=1$, and blends seamlessly with quantization-based sparsification.

**Scalability and Performance of FedLoRU in Large-Client Federated Learning**  Table 7 and Figure 2 compare FedAvg and FedLoRU in varying numbers of clients. As the number of clients increases, the

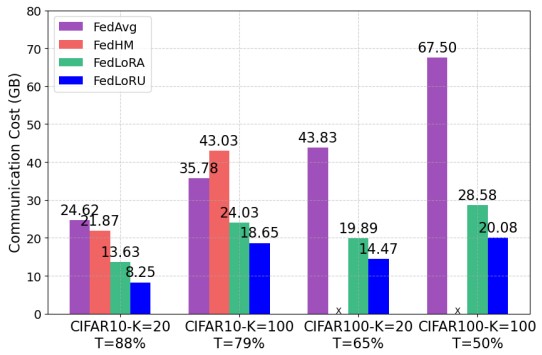

Figure 3: Communication cost of low-rank FL methods to reach target accuracy (X: not reached).

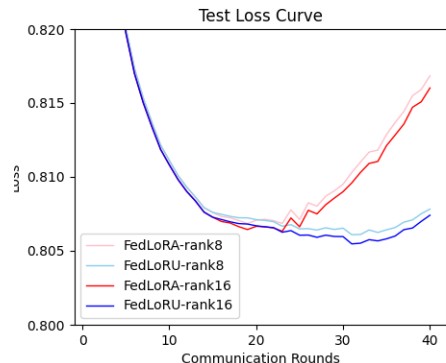

Figure 4: Test loss curve of FedLoRU and FedLoRA for fine-tuning LLaMA2-3B.

Table 2: GLUE fine-tuning results. Report (Mean ± std) over 3 random seeds.

| Dataset | CoLA (Mcc) | SST-2 (Acc) | STS-B (Pearson) | MRPC (Acc) |
|---|---|---|---|---|
| **FedLoRU** | 0.6338 ±0.003 | 0.9572 ±0.005 | 0.8791 ±0.008 | 0.8227 ±0.001 |
| **FedLoRA** | 0.6216 ±0.011 | 0.9562 ±0.005 | 0.8712 ±0.006 | 0.8153 ±0.012 |

scalability of the algorithm becomes a crucial factor. Our experiments show a sharp decline in FedAvg's performance, demonstrating its difficulty in maintaining accuracy as the number of clients grows.

In contrast, FedLoRU and FedLoRA outperform FedAvg when the number of clients exceeds 100 and 200, respectively. This trend is further reinforced in settings with a lower participation ratio, as shown in Table 8. Furthermore, the performance gap between low-rank algorithms and FedAvg continues to expand as $K$ increases. These findings emphasize that constraining updates to a low-rank subspace is particularly beneficial in federated learning environments with a large number of clients, and FedLoRU provides the most effective strategy among the compared low-rank approaches.

**Performance of LLM Fine-tuning** Figure 4 presents the loss curves of FedLoRA and FedLoRU during fine-tuning of the LLaMA2-3B model on the Alpaca dataset. The train loss curves show that both algorithms achieve similar convergence rates, with minimal differences in training optimization. However, a notable distinction emerges in the test loss results, where FedLoRU consistently outperforms FedLoRA after the 25th communication round.

To test whether these gains transfer beyond instruction tuning, we fine-tune TinyLlama 1.1B on four GLUE tasks. Table 2 summarizes the results, and FedLoRU consistently outperforms FedLoRA across tasks. These findings support our hypothesis that periodic accumulation of low-rank updates recovers high-rank expressiveness while preserving client-side regularization.

## 6 Conclusion

In this paper, we theoretically show that client-side optimization exhibits a higher-rank structure compared to server-side optimization and hypothesize that using low-rank updates in client-side optimization can promote an implicit regularization effect across clients. To the best of our knowledge, we are the first to establish a theoretical foundation supporting the use of low-rank updates in federated learning. Our proposed algorithm, FedLoRU, achieves comparable performance to FedAvg while significantly reducing the number of communicated parameters. Moreover, as the number of clients increases, FedLoRU consistently outperforms FedAvg, highlighting its scalability and effectiveness in large-scale federated learning environments.

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

# A  Proof of the Main Theorems

In this section, we provide proofs of Proposition 3.1, Theorem 3.2, and Proposition A.4. We first introduce precise definitions and problem setup, and state several auxiliary lemmas essential to our analysis. We then proceed with the formal proofs of the propositions and the theorem.

**Problem setup and more discussion on decoupled additive perturbed models**  We begin by introducing the set $\Omega^R(\theta_1, \cdots, \theta_k)$ and $\Omega(\theta_1, \cdots, \theta_k)$, over which we will establish convergence. For non-zero real numbers $\theta_1, \cdots, \theta_k$, we define $\Omega^R(\theta_1, \cdots, \theta_k)$ as the family of pairs $(h^R, \omega^R)$, where $h^R$ is an $R$-dimensional prediction function and $\omega^R$ is a weight vector, such that the true Hessian has non-zero eigenvalues $\theta_1, \cdots, \theta_k$. Specifically, $\Omega^R(\theta_1, \cdots, \theta_k) = \{(h^R, \omega^R) : \boldsymbol{H}_{\text{true}}(h^R, \omega^R) \text{ has non-zero eigenvalues } \theta_1, \cdots, \theta_k\}$. Let $\Omega(\theta_1, \cdots, \theta_k) = \bigcup_R \Omega^R(\theta_1, \cdots, \theta_k)$, representing the union of $\Omega^R(\theta_1, \cdots, \theta_k)$ over all dimensions $R$. We aim to show that the difference in the stable rank between the Hessians of the server and a client eventually becomes positive as dimension $R$ approaches infinity within the space of $\Omega(\theta_1, \cdots, \theta_k)$, which contains infinitely many $R$ for which $\Omega^R(\theta_1, \cdots, \theta_k) \neq \emptyset$, as proved in Appendix A.2.

To characterize the limiting spectral behavior of the empirical Hessians, we use the two decoupled additive perturbed model of the true Hessian. In our framework, we express the perturbed Hessians as

$$\boldsymbol{H}_N(h^R, \omega^R) = \boldsymbol{H}_{\text{true}}(h^R, \omega^R) + \epsilon_N^R,$$

with the error matrices defined as $\epsilon_N^R = s_N X^R$, where $X^R$ is a Wigner matrix. Wigner matrices have long been established as a canonical model for random perturbations in high-dimensional settings, such as perturbations in quantum systems (Guhr et al., 1998; Brody et al., 1981) or as noise models in signal processing (Tulino et al., 2004), making them particularly well-suited as error matrices in our additive perturbation model. The use of a Wigner matrix is justified by its ability to capture intrinsic statistical fluctuations in the eigenvalues and eigenvectors, a property that has been extensively verified both theoretically and empirically in Random Matrix Theory.

Additionally, we scale the variance of the entries of $X^R$ by $\sigma^2/R$ rather than $\sigma^2$. This scaling is crucial because it prevents the eigenvalues of the perturbed Hessian from diverging as the matrix dimension $R$ increases. If a variance of $\sigma^2$ were used, the eigenvalues of $\boldsymbol{H}_N(h^R, \omega^R)$ would diverge. In practice, the loss landscape displays controlled fluctuations, and the $\sigma^2/R$ scaling maintains consistency with the reasonable distribution of eigenvalues.

Our formulation also corrects a limitation in prior work. Baskerville et al. (2022) and Granziol et al. (2022) employs the model $\boldsymbol{H}_M^R = \boldsymbol{H}_N^R + \epsilon^R$, implying a dependency structure between $\boldsymbol{H}_M^R$ and $\boldsymbol{H}_N^R$. However, their analysis assumes independence between these matrices, which is problematic given the underlying model and practical considerations. In contrast, we address this issue by introducing two decoupled additive perturbed models.

## A.1  Useful Lemmas

We provide some lemmas that are required for our analysis.

**Lemma A.1** (Theorem 2.2 from Pielaszkiewicz & Singull (2015))**.** *Let $\mu_n$ be a sequence of probability measures on $\mathbb{R}$ and let $g_{\mu_n}$ denote the Stieltjes transform of $\mu_n$. We have*

a) if $\mu_n \to \mu$ weakly, where $\mu$ is a measure on $\mathbb{R}$, then $g_{\mu_n}(z) \to g_\mu(z)$ pointwise for any $z \in \{z \in \mathbb{C} : z = u + iv, v > 0\}$

b) if $g_{\mu_n}(z) \to g(z)$ pointwise, for all $z \in \{z \in \mathbb{C} : z = u + iv, v > 0\}$, then there exists a unique non-negative and finite measure such that $g = g_\mu$ and $\mu_n \to \mu$ weakly.

**Lemma A.2** (cf. Capitaine (2013)). *Let $\boldsymbol{X}_N$ be an $N \times N$ random real-symmetric Wigner matrix, and let $\boldsymbol{D}$ be a $N \times N$ deterministic symmetric matrix with uniformly bounded operator norm $\|\boldsymbol{D}\|$ in $N$. Let $\hat{\mu}_{\boldsymbol{X}}, \hat{\mu}_{\boldsymbol{D}}$ be the empirical spectral measures of the sequence of matrices $\boldsymbol{X}, \boldsymbol{D}$ and assume there exist deterministic limit measures $\mu_{\boldsymbol{X}}, \mu_{\boldsymbol{D}}$. Then $\boldsymbol{H} = \boldsymbol{X} + \boldsymbol{D}$ has a limiting spectral measure and is given by the free convolution $\mu_{\boldsymbol{X}} \boxplus \mu_{\boldsymbol{D}}$.*

Lemma A.2 states that in our additive perturbed model $\boldsymbol{H}_N(h^R, \omega^R) = \boldsymbol{H}_{\text{true}}(h^R, \omega^R) + \epsilon_N^R$, the matrix $\boldsymbol{H}_N(h^R, \omega^R)$ has a limiting spectral measure given by the free additive convolution $\mu_\nu \boxplus \mu_{\epsilon_N^R}$, where $\mu_\nu$ is the limiting spectral measure of $\boldsymbol{H}_{\text{true}}(h^R, \omega^R)$ and $\mu_{\epsilon_N^R}$ corresponds to the limiting spectral measure of $\epsilon_N^R$. The subsequent lemma, Weyl's inequality, examines the changes to eigenvalues of an Hermitian matrix that is perturbed.

**Lemma A.3** (Weyl's inequality). *For Hermitian matrices $\boldsymbol{A}, \boldsymbol{B} \in \mathbb{C}^{n \times n}$ and $i, j \in \{1, 2, \cdots, n\}$,*

$$\lambda_{i+j-1}(\boldsymbol{A} + \boldsymbol{B}) \leq \lambda_i(\boldsymbol{A}) + \lambda_j(\boldsymbol{B}), \quad i + j \leq n + 1, \tag{9}$$
$$\lambda_{i+j-n}(\boldsymbol{A} + \boldsymbol{B}) \geq \lambda_i(\boldsymbol{A}) + \lambda_j(\boldsymbol{B}), \quad i + j \geq n + 1, \tag{10}$$

*where $\lambda_i(\boldsymbol{D})$ is $i$-th eigenvalue of $\boldsymbol{D}$.*

## A.2 Proof of the Richness of $\Omega(\theta_1, \cdots, \theta_k)$.

In our theoretical analysis, we show the difference in stable rank between the Hessians of a server and a client eventually becomes positive as dimension $R$ approaches infinity within the space of $\Omega(\theta_1, \cdots, \theta_k)$. In this section, we discuss about the richness of $\Omega(\theta_1, \cdots, \theta_k)$ and characteristics of $\Omega^R(\theta_1, \cdots, \theta_k)$. They are defined as:

$$\Omega^R(\theta_1, \cdots, \theta_k) = \{(h^R, \omega^R) : \boldsymbol{H}_{\text{true}}(h^R, \omega^R) \text{ has non-zero eigenvalues } \theta_1, \cdots, \theta_k\}, \tag{11}$$
$$\Omega(\theta_1, \cdots, \theta_k) = \bigcup_R \Omega^R(\theta_1, \cdots, \theta_k). \tag{12}$$

In fact, the set of all possible pairs $(h^R, \omega^R)$ is represented by the union over all dimensions $R$, integers $k \leq R$, and non-zero real values $\theta_1, \cdots, \theta_k$ as follows:

$$\bigcup_{R=1}^{\infty} \{(h^R, \omega^R) : \text{any pair } (h^R, \omega^R) \text{ of dimension } R\} = \bigcup_{R=1}^{\infty} \bigcup_{k=1}^{R} \bigcup_{(\theta_1, \cdots, \theta_k) \in \mathbb{R}^k} \Omega^R(\theta_1, \cdots, \theta_k).$$

Thus, for any given pair $(h^R, \omega^R)$, there exist $\theta_1, \cdots, \theta_k$ such that $(h^R, \omega^R) \in \Omega^R(\theta_1, \cdots, \theta_k)$. According to the following proposition, either the set $\Omega(\theta_1, \cdots, \theta_k)$ is empty or there exist infinitely many values of $R$ for which $\Omega^R(\theta_1, \cdots, \theta_k) \neq \emptyset$.

**Proposition A.4.** *Let $\theta_1, \cdots, \theta_k$ be fixed non-zero real numbers, and suppose there exists $\tilde{R} > k$ such that $\Omega^{\tilde{R}}(\theta_1, \cdots, \theta_k)$ is non-empty. Then $\Omega^R(\theta_1, \cdots, \theta_k)$ is non-empty for all $R \geq \tilde{R}$.*

*Proof.* To establish the proposition, it suffices to demonstrate that $\Omega^{\tilde{R}}(\theta_1, \cdots, \theta_k) \neq \emptyset$ implies $\Omega^{\tilde{R}+1}(\theta_1, \cdots, \theta_k) \neq \emptyset$. To this end, let $(h^{\tilde{R}}, \omega^{\tilde{R}}) \in \Omega^{\tilde{R}}(\theta_1, \cdots, \theta_k)$. Our objective is to show that there exists $(h^{\tilde{R}+1}, \omega^{\tilde{R}+1}) \in \Omega^{\tilde{R}+1}(\theta_1, \cdots, \theta_k)$. To construct a prediction function $h^{\tilde{R}+1}$ and a weight $\omega^{\tilde{R}+1}$ of dimension

$\tilde{R}+1$ such that the true Hessian retains the same non-zero eigenvalues, we define $h^{\tilde{R}+1} : \mathbb{R}^{d_x} \times \mathbb{R}^{\tilde{R}+1} \to \mathbb{R}^{d_y}$ and $\omega^{\tilde{R}+1} \in \mathbb{R}^{\tilde{R}+1}$ as

$$h^{\tilde{R}+1}(x;\omega) = \tilde{h}^{\tilde{R}+1}(x;\omega), \ \ \forall x \in \mathbb{R}^{d_x}, \forall \omega \in \mathbb{R}^{\tilde{R}+1}, \tag{13}$$

$$\omega^{\tilde{R}+1} = (\omega^{\tilde{R}}, 0) \tag{14}$$

where $\tilde{h}^{\tilde{R}+1} : \mathbb{R}^{d_x} \times \mathbb{R}^{\tilde{R}+1} \to \mathbb{R}^{d_y}$ is defined as

$$\tilde{h}^{\tilde{R}+1}(x;\omega^{\tilde{R}+1}) = \tilde{h}^{\tilde{R}+1}(x;(\omega^{\tilde{R}},0)) = h^{\tilde{R}}(x;\omega^{\tilde{R}}) \tag{15}$$

which is independent of the last variable $z \in \mathbb{R}$ for all $\omega \in \mathbb{R}^R$. Expanding the Hessian of the loss function at $\omega^{\tilde{R}+1}$ for any $(x,y) \sim \psi$, we obtain

$$\begin{aligned}
\nabla_\omega^2 \ell(h^{\tilde{R}+1}(x;\omega^{\tilde{R}+1}), y) &= \mathbf{J}_\omega(h^{\tilde{R}+1}(x;\omega^{\tilde{R}+1}))^T \nabla_y^2 \ell(h^{\tilde{R}+1}(x;\omega^{\tilde{R}+1}), y) \mathbf{J}_\omega(h^{\tilde{R}+1}(x;\omega^{\tilde{R}+1})) \\
&\quad + \sum_{i=1}^{d_y} \frac{\partial \ell}{\partial y_i}(h^{\tilde{R}+1}(x;\omega^{\tilde{R}+1}), y) \cdot \nabla_\omega^2 h_i^{\tilde{R}+1}(x;\omega^{\tilde{R}+1})
\end{aligned} \tag{16}$$

where $h^{\tilde{R}+1} = [h_1^{\tilde{R}+1}, \cdots, h_{d_y}^{\tilde{R}+1}]^T$ and $\mathbf{J}_\omega(h^{\tilde{R}+1}(x;\omega^{\tilde{R}+1}))$ is the Jacobian of the function $h^{\tilde{R}+1}$ with respect to $R+1$ dimensional input $\omega$. Then, by the definition of $h^{\tilde{R}+1}$ and $\omega^{\tilde{R}+1}$, we have:

$$h^{\tilde{R}+1}(x;\omega^{\tilde{R}+1}) = h^{\tilde{R}}(x;\omega^{\tilde{R}}), \tag{17}$$

$$\mathbf{J}_\omega(h^{\tilde{R}+1}(x;\omega^{\tilde{R}+1})) = \begin{bmatrix} \mathbf{J}_\omega(h^{\tilde{R}}(x;\omega^{\tilde{R}})) & \begin{matrix} | \\ 0 \\ | \end{matrix} \end{bmatrix}, \tag{18}$$

$$\nabla_\omega^2 h_i^{\tilde{R}+1}(x;\omega^{\tilde{R}+1}) = \begin{bmatrix} \nabla_\omega^2 h_i^{\tilde{R}}(x;\omega^{\tilde{R}}) & \begin{matrix} | \\ 0 \\ | \end{matrix} \\ \hline & 0 & \end{bmatrix}, \ \forall i. \tag{19}$$

Substituting these expressions into the expanded Hessian equation (16), we conclude that $\nabla_w^2 \ell(h^{\tilde{R}+1}(x;\omega^{\tilde{R}+1}), y)$ is identical to $\nabla_w^2 \ell(h^{\tilde{R}}(x;\omega^{\tilde{R}}), y)$ for any $(x,y) \sim \psi$ except for a final zero row and column. Thus, $(h^{\tilde{R}+1}, \omega^{\tilde{R}+1})$ and $(h^{\tilde{R}}, \omega^{\tilde{R}})$ have the same true Hessian, except for the zero-row and the zero-column, which have no impact on the non-zero eigenvalues of the Hessian. It follows that $(h^{\tilde{R}+1}, \omega^{\tilde{R}+1}) \in \Omega^{\tilde{R}+1}(\theta_1, \cdots, \theta_k)$.

For example, if we consider feedforward neural networks as prediction functions, one can easily construct a larger neural network that maintains the same non-zero eigenvalues by adding an additional neuron with a single connection to a neuron in the previous layer. This additional neuron does not affect the final output, thereby preserving the desired eigenvalue properties. $\qquad \square$

## A.3 Proof of Proposition 3.1

Numerous studies (Benaych-Georges & Nadakuditi, 2011; 2012; Chen et al., 2021; Péché, 2006) have investigated the eigenvalue behavior of perturbed matrices. In this proposition, we analyze the limiting eigenvalues of a perturbed random matrix when the perturbation is given by a Wigner matrix and the original matrix has fixed eigenvalues.

To prove Proposition 3.1, we decompose the eigenvalue analysis into two distinct parts. First, we demonstrate that the $i$-th eigenvalues, where $i \in \{p+1, \cdots, R-q-1\}$, converge to the upper or lower bounds of the spectral density of $\mu_N$. Here, $\mu_N$ is the limiting spectral density of $\epsilon_N^R$. This portion of the proof parallels the approach employed by Benaych-Georges & Nadakuditi (2011). Second, we show that the remaining eigenvalues converge to the Stieltjes transformation. This part of the proof follows the methodology outlined by Baskerville et al. (2022).

*Proof.* In this proof, we drop dependency on $(h^R, \omega^R)$ and simplify the notation by representing $\boldsymbol{H}_N(h^R, \omega^R)$ and $\boldsymbol{H}_{\text{true}}(h^R, \omega^R)$ as $\boldsymbol{H}_N^R$ and $\boldsymbol{H}_{\text{true}}^R$, respectively. Let us consider $\lambda_i(\boldsymbol{H}_N^R)$ for the index range $p < i < R - q$. Applying Lemma A.3, we obtain

$$\lambda_i(\boldsymbol{H}_N^R) \leq \lambda_{1+i-j}(\boldsymbol{H}_{\text{true}}^R) + \lambda_{1+i-k}(\epsilon^R(N)), \quad i = j + k - 1 \leq R,\ j, k \in \{1, \cdots, R\}, \tag{20}$$

$$\lambda_i(\boldsymbol{H}_N^R) \geq \lambda_{R+i-j}(\boldsymbol{H}_{\text{true}}^R) + \lambda_{R+i-k}(\epsilon^R(N)), \quad i = j + k - R \geq 1,\ j, k \in \{1, \cdots, R\}. \tag{21}$$

By letting $k = 1 + p$ in (20) and $k = R - q$ in (21), we derive

$$\lambda_i(\boldsymbol{H}_N^R) \leq \lambda_{1+i-j}(\boldsymbol{H}_{\text{true}}^R) + \lambda_{i-p}(\epsilon^R(N)), \quad i = j + p \leq R,\ j \in \{1, \cdots, R\}, \tag{22}$$

$$\lambda_i(\boldsymbol{H}_N^R) \geq \lambda_{R+i-j}(\boldsymbol{H}_{\text{true}}^R) + \lambda_{i+q}(\epsilon^R(N)), \quad i = j - q \geq 1,\ j \in \{1, \cdots, R\}. \tag{23}$$

By substituting $i - j = p$ in (22) and $i - j = -q$ in (23), and utilizing the facts that $\lambda_{1+p}(\boldsymbol{H}_{\text{true}}^R) = 0$ and $\lambda_{R-q}(\boldsymbol{H}_{\text{true}}^R) = 0$, we deduce

$$\lambda_{i+q}(\epsilon_N^R) \leq \lambda_i(\boldsymbol{H}_N^R) \leq \lambda_{i-p}(\epsilon_N^R), \quad \forall i \in \{1, \cdots, R\}, \tag{24}$$

where $\lambda_k(\epsilon_N^R) = -\infty$ if $k > R$, and $+\infty$ if $k \leq 0$. Additionally, since $\epsilon_N^R$ has the limiting spectral density $\mu_N$ and $L_N, U_N$ are lower and upper bounds of $\mu_N$, we have, for all $i \geq 1$ fixed,

$$\liminf_{R \to \infty} \lambda_i(\epsilon_N^R) \geq U_N \quad \text{and} \quad \limsup_{R \to \infty} \lambda_{R+1-i}(\epsilon_N^R) \leq L_N, \tag{25}$$

$$\lambda_1(\epsilon_N^R) \to U_N \quad \text{and} \quad \lambda_R(\epsilon_N^R) \to L_N. \tag{26}$$

From these relations, it follows that for any fixed $i \geq 1$, $\lambda_i(\epsilon_N^R)$ converges to $U_N$ and $\lambda_{R+1-i}(\epsilon_N^R)$ converges to $L_N$ as $R \to \infty$. By applying (25) in (24), we obtain, for all fixed $i \geq 1$,

$$\liminf_{R \to \infty} \lambda_i(\boldsymbol{H}_N^R) \geq U_N \quad \text{and} \quad \limsup_{R \to \infty} \lambda_i(\boldsymbol{H}_N^R) \leq L_N \tag{27}$$

By combining (24), (26), and (27), for all $i > p$ (respectively, $i \geq q$) fixed, we have

$$\lambda_i(\boldsymbol{H}_N^R) \to U_N \quad (\text{respectively,} \ \lambda_{R-i}(\boldsymbol{H}_N^R) \to L_N). \tag{28}$$

Next, we aim to prove the behavior of the remaining eigenvalues $\lambda_i(\boldsymbol{H}_N^R)$ for $i \in \{1, \cdots, p, R-q+1, \cdots R\}$. Note that, since $p + q \ll R$ when $R$ is sufficiently large, the limiting spectral density of $H_{\text{true}}^R$ converges to $\nu = \delta_0$. Furthermore, because $X^R$ is a Wigner matrix, its limiting spectral density is given by the semicircular distribution, denoted by $\mu$.

Let us consider $\lambda_i(\boldsymbol{H}_N^R)$ where $i \leq p$ or $i \geq R - q$. According to Lemma A.2, the limiting spectral density $\mu_{\boldsymbol{H}_N^R}$ of $\boldsymbol{H}_N^R$ is given by $\mu_N \boxplus \nu$, where $\mu_N$ is the limiting spectral density of $\epsilon_N^R$. By Lemma A.1, the Stieltjes transform $g_{\mu_{\boldsymbol{H}_N^R}}(z)$ converges pointwise to $g_{\nu \boxplus \mu_N}(z)$ for any $z \in \{z : z \in \mathbb{C},\ z = u + iv, v > 0\}$. Consequently, we have:

$$\hat{g}_{\boldsymbol{H}_N^R}(z) = g_{\mu_{\boldsymbol{H}_N^R}}(z) + o(1)$$

$$= g_{\mu_N \boxplus \nu}(z) + o(1)$$

$$= g_\nu(k(z)) + o(1) \tag{29}$$

$$= \hat{g}_{\boldsymbol{H}_{\text{true}}^R}(k(z)) + o(1),$$

where $k$ is the subordination function such that $g_{\mu_N \boxplus \nu}(z) = g_\nu(k(z))$.

Let $\lambda \in \mathbb{R} \backslash \text{supp}(\mu_N \boxplus \nu)$ be an eigenvalue of $\boldsymbol{H}_N^R$. Then $\hat{g}_{\boldsymbol{H}_N^R}$ has a singularity at $\lambda$, and thus $\hat{g}_{\boldsymbol{H}_{\text{true}}^R}$ must also have a singularity at $k(\lambda)$. Thus, for any $R$, this singularity persists, implying that $k(\lambda)$ must correspond to one of the outlier eigenvalues of $\boldsymbol{H}_N^R$. In other words, $\theta_i$ is an outlier eigenvalue of $\boldsymbol{H}_{\text{true}}^R$ if and only if there exists an eigenvalue $\lambda$ of $\boldsymbol{H}_N^R$ in $\mathbb{R} \backslash \text{supp}(\mu_N \boxplus \nu)$ such that $k(\lambda) = \theta_i$. Thus, the family of the outliers of $\boldsymbol{H}_N^R$ can be expressed as

$$\{k^{-1}(\theta_j) : k^{-1}(\theta_j) \in \mathbb{R} \backslash \text{supp}(\mu_N \boxplus \nu)\}. \tag{30}$$

Note that $\text{supp}(\mu_N \boxplus \nu) = \text{supp}(\mu_N \boxplus \delta_0) = \text{supp}(\mu_N)$. Our next goal is to determine the form of $k^{-1}(\theta_j)$. From the subordination function relation, we have:

$$k^{-1}(\theta) = g_{\mu_N \boxplus \nu}^{-1}(g_\nu(\theta))$$

$$= \mathcal{R}_{\mu_N}(g_\nu(\theta) + g_\nu^{-1}(g_\nu(\theta))) \tag{31}$$

$$= \mathcal{R}_{\mu_N}(1/\theta) + \theta.$$

Note that by the definition of Stieltjes transformation and $\mathcal{R}$-transform, we have $g_\nu(\theta) = g_{\delta_0}(\theta) = 1/\theta$.

Let $m_n^{(\mu)}$ denote the $n$-th moment of a distribution $\mu$, and let $C_n^{(\mu)}$ denote the $n$-th cumulant of $\mu$. The relationship between $m_n^{(\mu)}$ and $C_n^{(\mu)}$ is given by Anderson et al. (2010) as

$$m_n^{(\mu)} = \sum_{r=1}^n \sum_{\substack{0 \le i_1, \cdots, i_r \le n-r \\ i_1 + \cdots + i_r = n-r}} C_r^{(\mu)} \left[ \Pi_{j=1}^r m_{i_j}^{(\mu)} \right]. \tag{32}$$

Using the scaling property of moments, $m_n^{\mu_N} = s_N^n m_n^\mu$, we can derive the corresponding scaling relation for the cumulants as $C_n^{(\mu_N)} = s_N^n C_n^{(\mu)}$. Consequently, the $\mathcal{R}$-transform exhibits the scaling property

$$\mathcal{R}_{\mu_N}(\theta) = s_N \mathcal{R}_\mu(s_N \theta). \tag{33}$$

Finally, we have an expression for the outliers of $\boldsymbol{H}_N^R$ as

$$k^{-1}(\theta) = s_N \mathcal{R}_\mu(s_N/\theta) + \theta. \tag{34}$$

Since $\mathcal{R}$-transform of a semicircle law $\mu$ is given by $\mathcal{R}_\mu(x) = \sigma^2 x$, we have $k^{-1}(\theta) = \theta + \frac{\sigma^2 s_N^2}{\theta}$.

$\square$

## A.4 Proof of Theorem 3.2

*Proof.* Define the sets $\mathcal{P}_N = \{i \le p : g_N^{-1}(\theta_i) > U_N\} = \{i \le p : \lambda_i(\boldsymbol{H}_N^R) \to g_N^{-1}(\theta_i)\}$ and $\mathcal{Q}_N = \{i > p : g_N^{-1}(\theta_i) < L_N\} = \{i > p : \lambda_i(\boldsymbol{H}_N^R) \to g_N^{-1}(\theta_i)\}$, which represent the indices of eigenvalues $\lambda_i(\boldsymbol{H}_N^R)$ converging to $g_N^{-1}(\theta_i)$. Let $N_u = |\{i : \lambda_i(\boldsymbol{H}_N^R) \to U_N\}|$ and $N_l = |\{i : \lambda_i(\boldsymbol{H}_N^R) \to L_N\}|$ denote their cardinalities of the

set of indices whose corresponding limiting eigenvalues converge to $U_N$ and $L_N$, respectively. Similarly, define $\mathcal{P}_M$, $\mathcal{Q}_M$, $M_u$, and $M_l$ for $\boldsymbol{H}_M^R$ analogously.

It is possible that $g_N^{-1}(\theta_i) \leq U_N$ for all $i \in \{1, \cdots, p\}$ or $g_N^{-1}(\theta_i) \geq L_N$ for all $i \in \{p+1, \cdots, p+q\}$ . In this case, we can just let $\mathcal{P}_N = \emptyset$ or $\mathcal{Q}_N = \emptyset$, respectively.

Define $\theta_0 = \theta_1 \cdot \mathbf{1}_{|\theta_1| \geq |\theta_{p+q}|} + \theta_{p+q} \cdot \mathbf{1}_{|\theta_1| < |\theta_{p+q}|}$ to represent the limiting eigenvalue based on the larger magnitude between $\theta_1$ and $\theta_{p+q}$. Using the limiting eigenvalues of $\boldsymbol{H}_N^R$, define the estimated stable rank as:

$$\hat{\mathrm{srank}}(\boldsymbol{H}_N^R) = \sum_{j \in \mathcal{P}_N \cup \mathcal{Q}_N} \frac{g_N^{-1}(\theta_j)^2}{g_N^{-1}(\theta_0)^2} + N_u \frac{U_N^2}{g_N^{-1}(\theta_0)^2} + N_l \frac{L_N^2}{g_N^{-1}(\theta_0)^2}. \tag{35}$$

Similarly, $\hat{\mathrm{srank}}(\boldsymbol{H}_M^R)$ is defined in the same manner. By Proposition 3.1, it follows that $\left| \mathrm{srank}(\boldsymbol{H}_N^R) - \hat{\mathrm{srank}}(\boldsymbol{H}_N^R) \right| \to 0$ and $\left| \mathrm{srank}(\boldsymbol{H}_M^R) - \hat{\mathrm{srank}}(\boldsymbol{H}_M^R) \right| \to 0$. Consequently, we have

$$\left| \left( \mathrm{srank}(\boldsymbol{H}_M^R) - \mathrm{srank}(\boldsymbol{H}_N^R) \right) - \left( \hat{\mathrm{srank}}(\boldsymbol{H}_M^R) - \hat{\mathrm{srank}}(\boldsymbol{H}_N^R) \right) \right| \to 0. \tag{36}$$

Given that $U_N < U_M$ and $L_N > L_M$, it follows that $\mathcal{P}_N \subseteq \mathcal{P}_M$ and $\mathcal{Q}_N \subseteq \mathcal{Q}_M$. Furthermore, since $U_N^2 = L_N^2$, by matching the indices in $\hat{\mathrm{srank}}(\boldsymbol{H}_N^R)$ and $\hat{\mathrm{srank}}(\boldsymbol{H}_M^R)$, we can express the difference between the limiting stable rank as

$$\begin{aligned}
\hat{\mathrm{srank}}(\boldsymbol{H}_M^R) - \hat{\mathrm{srank}}(\boldsymbol{H}_N^R) = & \sum_{j \in \mathcal{P}_N \cup \mathcal{Q}_N} \left( \frac{g_M^{-1}(\theta_j)^2}{g_M^{-1}(\theta_0)^2} - \frac{g_N^{-1}(\theta_j)^2}{g_N^{-1}(\theta_0)^2} \right) \\
& + \sum_{j \in (\mathcal{P}_N^c \cap \mathcal{P}_M) \cup (\mathcal{Q}_N^c \cap \mathcal{Q}_M)} \left( \frac{g_M^{-1}(\theta_j)^2}{g_M^{-1}(\theta_0)^2} - \frac{U_N^2}{g_N^{-1}(\theta_0)^2} \right) \\
& + (M_u + M_l) \left( \frac{U_M^2}{g_M^{-1}(\theta_0)^2} - \frac{U_N^2}{g_N^{-1}(\theta_0)^2} \right).
\end{aligned} \tag{37}$$

(i) We begin by showing that the first summation term, $\sum_{j \in \mathcal{P}_N \cup \mathcal{Q}_N} \left( \frac{g_M^{-1}(\theta_j)^2}{g_M^{-1}(\theta_0)^2} - \frac{g_N^{-1}(\theta_j)^2}{g_N^{-1}(\theta_0)^2} \right)$, is positive and increasing with respect to $M$. To achieve this, we analyze the individual term $F_j = \frac{g_M^{-1}(\theta_j)^2}{g_M^{-1}(\theta_0)^2} - \frac{g_N^{-1}(\theta_j)^2}{g_N^{-1}(\theta_0)^2}$ for $j \in \mathcal{P}_N \cup \mathcal{Q}_N$, which appears in the first summation of (37).

Expanding $F_j$ and factoring the numerator, we have

$$\begin{aligned}
F_j &= \frac{g_M^{-1}(\theta_j)^2 g_N^{-1}(\theta_0)^2 - g_N^{-1}(\theta_j)^2 g_M^{-1}(\theta_0)^2}{g_M^{-1}(\theta_0)^2 g_N^{-1}(\theta_0)^2} \\
&= \frac{\left( g_M^{-1}(\theta_j) g_N^{-1}(\theta_0) + g_N^{-1}(\theta_j) g_M^{-1}(\theta_0) \right) \left( g_M^{-1}(\theta_j) g_N^{-1}(\theta_0) - g_N^{-1}(\theta_j) g_M^{-1}(\theta_0) \right)}{g_M^{-1}(\theta_0)^2 g_N^{-1}(\theta_0)^2}
\end{aligned}$$

Substituting $g_M^{-1}(\theta) = \theta + \frac{\sigma^2 s_M^2}{\theta}$ and $g_N^{-1}(\theta) = \theta + \frac{\sigma^2 s_N^2}{\theta}$ and simplifying, we can express the difference as

$$\begin{aligned}
g_M^{-1}(\theta_j) g_N^{-1}(\theta_0) - g_N^{-1}(\theta_j) g_M^{-1}(\theta_0) &= \left( \theta_j + \frac{\sigma^2 s_M^2}{\theta_j} \right) \left( \theta_0 + \frac{\sigma^2 s_N^2}{\theta_0} \right) - \left( \theta_j + \frac{\sigma^2 s_N^2}{\theta_j} \right) \left( \theta_0 + \frac{\sigma^2 s_M^2}{\theta_0} \right) \\
&= \sigma^2 (s_M^2 - s_N^2) \left( \frac{\theta_0}{\theta_j} - \frac{\theta_j}{\theta_0} \right).
\end{aligned}$$

Thus, $F_j$ becomes

$$F_j = \frac{g_M^{-1}(\theta_j)g_N^{-1}(\theta_0) + g_N^{-1}(\theta_j)g_M^{-1}(\theta_0)}{g_M^{-1}(\theta_0)^2 g_N^{-1}(\theta_0)^2} \cdot \sigma^2(s_M^2 - s_N^2)\left(\frac{\theta_0}{\theta_j} - \frac{\theta_j}{\theta_0}\right). \tag{38}$$

For the sign analysis, the term $g_M^{-1}(\theta_j)g_N^{-1}(\theta_0) + g_N^{-1}(\theta_j)g_M^{-1}(\theta_0)$ takes the sign of $\theta_0\theta_j$, as the sign of $g_M^{-1}(\theta)$ and $g_N^{-1}(\theta)$ are dependent of the sign of $\theta$. The difference $s_M^2 - s_N^2$ is positive, and the term $\frac{\theta_0}{\theta_j} - \frac{\theta_j}{\theta_0}$ also has the sign of $\theta_0\theta_j$. Combining these observations, the overall sign of $F_j$ is positive because all contributing terms either maintain a positive sign or do not introduce a sign change. Further, since $g_M^{-1}(\theta_j) \geq U_M$ for $j \in \mathcal{P}_M$ or $g_M^{-1}(\theta_j) \leq L_M$ for $j \in \mathcal{Q}_M$, a lower bound for $F_j$ can be established as follows:

$$F_j \geq \frac{8\sigma^4 s_M s_N (s_M^2 - s_N^2)}{g_M^{-1}(\theta_0)^2 g_N^{-1}(\theta_0)^2} \cdot \left|\frac{\theta_0}{\theta_j} - \frac{\theta_j}{\theta_0}\right|. \tag{39}$$

To show that the first summation term $\sum_{j \in \mathcal{P}_N \cup \mathcal{Q}_N} F_j$ is a decreasing function with respect to $M$, we compute the derivative of $F_j$ with respect to $M$. The derivative can be expressed as

$$\frac{\partial F_j}{\partial M} = \frac{\partial F_j}{\partial s_M} \cdot \frac{\partial s_M}{\partial M} = \frac{4\sigma^2 s_M\left(\theta_0^2 - \theta_j^2\right)}{\theta_0\theta_j} \cdot \frac{g_M^{-1}(\theta_j)}{g_M^{-1}(\theta_0)^3} \cdot \frac{\partial s_M}{\partial M} \tag{40}$$

Since $s_M$ is a decreasing function of $M$, it follows that $\frac{\partial s_M}{\partial M} < 0$. Additionally, the term $\frac{4\sigma^2 s_M\left(\theta_0^2 - \theta_j^2\right)}{\theta_0\theta_j} \cdot \frac{g_M^{-1}(\theta_j)}{g_M^{-1}(\theta_0)^3}$ is positive as same way in the sign analysis. Consequently, the product is negative, implying $\frac{\partial F_j}{\partial M} < 0$. This shows that $F_j$ decreases with $M$. Therefore, the first summation term, which is a sum of such $\sum_{j \in \mathcal{P}_N \cup \mathcal{Q}_N} F_j$ is a decreasing function of $M$.

(ii) We next show the lower bound of remaining terms $\sum_{j \in (\mathcal{P}_N^c \cap \mathcal{P}_M) \cup (\mathcal{Q}_N^c \cap \mathcal{Q}_M)} \left(\frac{g_M^{-1}(\theta_j)^2}{g_M^{-1}(\theta_0)^2} - \frac{U_N^2}{g_N^{-1}(\theta_0)^2}\right) + (M_u + M_l)\left(\frac{U_M^2}{g_M^{-1}(\theta_0)^2} - \frac{U_N^2}{g_N^{-1}(\theta_0)^2}\right)$ is positive and decreases with $M$. Since $g_M^{-1}(\theta_j) \geq U_M$ for $j \in (\mathcal{P}_N^c \cap \mathcal{P}_M) \cup (\mathcal{Q}_N^c \cap \mathcal{Q}_M)$, it follows that

$$\sum_{j \in (\mathcal{P}_N^c \cap \mathcal{P}_M) \cup (\mathcal{Q}_N^c \cap \mathcal{Q}_M)} \left(\frac{g_M^{-1}(\theta_j)^2}{g_M^{-1}(\theta_0)^2} - \frac{U_N^2}{g_N^{-1}(\theta_0)^2}\right) + (M_u + M_l)\left(\frac{U_M^2}{g_M^{-1}(\theta_0)^2} - \frac{U_N^2}{g_N^{-1}(\theta_0)^2}\right)$$
$$\geq \sum_{j \in (\mathcal{P}_N^c \cap \mathcal{P}_M) \cup (\mathcal{Q}_N^c \cap \mathcal{Q}_M)} \left(\frac{U_M^2}{g_M^{-1}(\theta_0)^2} - \frac{U_N^2}{g_N^{-1}(\theta_0)^2}\right) + (M_u + M_l)\left(\frac{U_M^2}{g_M^{-1}(\theta_0)^2} - \frac{U_N^2}{g_N^{-1}(\theta_0)^2}\right) \tag{41}$$
$$= B_N \left(\frac{U_M^2}{g_M^{-1}(\theta_0)^2} - \frac{U_N^2}{g_N^{-1}(\theta_0)^2}\right),$$

where $B_N = |\{i : \lambda_i(H_N^R) \to U_N \text{ or } L_N\}|$. Now consider the difference $\frac{U_M^2}{g_M^{-1}(\theta_0)^2} - \frac{U_N^2}{g_N^{-1}(\theta_0)^2}$. By expanding and simplifying, we have

$$\frac{U_M^2}{g_M^{-1}(\theta_0)^2} - \frac{U_N^2}{g_N^{-1}(\theta_0)^2} = \frac{U_M^2 g_N^{-1}(\theta_0)^2 - U_N^2 g_M^{-1}(\theta_0)^2}{g_M^{-1}(\theta_0)^2 g_N^{-1}(\theta_0)^2}$$
$$= \frac{4\sigma^2 s_M^2\left(\theta_0 + \sigma^2 s_N^2/\theta_0\right)^2 - 4\sigma^2 s_N^2\left(\theta_0 + \sigma^2 s_M^2/\theta_0\right)^2}{g_M^{-1}(\theta_0)^2 g_N^{-1}(\theta_0)^2}$$
$$= \frac{4\sigma^2\left(\theta_0^2(s_M^2 - s_N^2) + \sigma^4 s_M^2 s_N^2\left(s_N^2/\theta_0^2 - s_M^2/\theta_0^2\right)\right)}{g_M^{-1}(\theta_0)^2 g_N^{-1}(\theta_0)^2} \tag{42}$$
$$= \frac{4\sigma^2(s_M^2 - s_N^2)\left(\theta_0^2 - \sigma^4 s_M^2 s_N^2/\theta_0^2\right)}{g_M^{-1}(\theta_0)^2 g_N^{-1}(\theta_0)^2}.$$

Given the assumption that $\theta_0^2 \geq \sigma^2 s_M^2 > \sigma^2 s_M s_N$, the numerator is positive, ensuring that (42) is positive. To establish that this bound decreases with $M$, we compute the derivative with respect to $M$:

$$
\frac{\partial}{\partial M}\left(\frac{U_M^2}{g_M^{-1}(\theta_0)^2} - \frac{U_N^2}{g_N^{-1}(\theta_0)^2}\right) = \frac{\partial}{\partial s_M}\left(\frac{U_M^2}{g_M^{-1}(\theta_0)^2} - \frac{U_N^2}{g_N^{-1}(\theta_0)^2}\right) \cdot \frac{\partial s_M}{\partial M}
$$

$$
= \frac{2U_M(U_M)'g_M^{-1}(\theta_0)^2 - 2g_M^{-1}(\theta_0)(g_M^{-1}(\theta_0))'U_M^2}{g_M^{-1}(\theta_0)^4} \cdot \frac{\partial s_M}{\partial M}
$$

$$
= \frac{4\sigma U_M g_M^{-1}(\theta_0) - 4\sigma^2 s_M U_M^2 g_M^{-1}(\theta_0)/\theta_0}{g_M^{-1}(\theta_0)^4} \cdot \frac{\partial s_M}{\partial M}
$$

$$
= \frac{4\sigma U_M(\theta_0^2 - \sigma^2 s_M^2)}{\theta_0 g_M^{-1}(\theta_0)^3} \cdot \frac{\partial s_M}{\partial M}. \tag{43}
$$

Since $\theta_0^2 \geq \sigma^2 s_M^2$ and $\frac{\partial s_M}{\partial M} < 0$, the derivative is negative, indicating that the lower bound of (41) is a decreasing function of $M$.

By (i) and (ii), the difference in the limiting stable rank between $\boldsymbol{H}_N^R$ and $\boldsymbol{H}_M^R$ is

$$
\hat{\mathrm{srank}}(\boldsymbol{H}_M) - \hat{\mathrm{srank}}(\boldsymbol{H}_N) \geq \frac{s_M^2 - s_N^2}{g_M^{-1}(\theta_0)^2 g_N^{-1}(\theta_0)^2}\left[\sum_{j \in \mathcal{P}_N \cup \mathcal{Q}_N} 8\sigma^4 s_M s_N \left|\frac{\theta_0}{\theta_j} - \frac{\theta_j}{\theta_0}\right| + 4\sigma^2 B_N\left(\theta_0^2 - \frac{\sigma^4 s_M^2 s_N^2}{\theta_0^2}\right)\right],
$$
$$\tag{44}$$

thus it is positive and its lower bound is a decreasing function of $M$. $\qquad\square$

## A.5 Proof of Theorem 3.3

We rearrange the indices such that eigenvalues $\theta_1, \cdots, \theta_{p+q}$ of $H_{\mathrm{true}}^R$ satisfy $|\theta_1| \geq \cdots \geq |\theta_{p+q}|$. Let $v_i$ be the unit-norm eigenvector associated with the eigenvalue $\theta_i$ of $H_{\mathrm{true}}^R$, and let $u_i^{(k)}$ be the unit-norm eigenvector of $\boldsymbol{H}_N^{(k)}$ corresponding to the eigenvalue whose limiting eigenvalue is $g_N^{-1}(\theta_i)$. Define $U^{(k)}$ be the subspace spanned by $\{u_i^{(k)}\}$. For all $k \in \{1, \cdots, K\}$, the dimension of $U^{(k)}$ is identical for each client and is denoted as $\tilde{r} = |U^{(k)}|$. Additionally, let $W_N^{(k)} = \{w_i^{(k)}\}_{i=1}^{l_k}$ be remaining limiting eigenvectors of $H_N^{(k)}(h^R, \omega^R)$. The indices of $\{w_i^{(k)}\}_{i=1}^{l_k}$ are rearranged in descending order based on the magnitudes of their associated singular values. We formalize the assumption stated in the main text:

**Assumption A.5.** $\{w_i^{(k)}\}_{i=1}^{l_k}$ are random unit-norm orthonormal vectors such that $w_i^{(k)} \perp U_N^{(k)}, \forall i$, and the limiting value of expected directional derivative $\mathbb{E}[\partial_{w_i^{(k)}} f_N^{(k)}(h^R, \omega^R)]$ have same values for all $i$ and $k$.

Define $\phi_i = \sqrt{1 - \frac{\sigma^2 s_N^2}{\theta_i^2}}$, and note that $|\langle v_i, u_i^{(k)}\rangle| \to \phi_i$ by (5). The following lemma provides the limiting value of the expected inner product between eigenvectors of different clients, which is used in proving Theorem 3.3.

**Lemma A.6.** For any $k_1 \neq k_2 \in \{1, \cdots, K\}$ and for any $i, j$, the limiting value of the expected inner product between eigenvectors of different clients $k_1$ and $k_2$ is as follows:

a) $\mathbb{E}\left[\langle u_i^{(k_1)}, u_j^{(k_2)}\rangle\right] \to \phi_i\phi_j \mathbb{1}\{i = j\},$

b) $\mathbb{E}\left[\langle w_i^{(k_1)}, w_j^{(k_2)}\rangle\right] \to 0,$

c) $\mathbb{E}\left[\langle u_i^{(k_1)}, w_j^{(k_2)}\rangle\right] \to 0.$

*Proof.* Let $\phi_i^{(k)}$ be the cosine of the angle between $v_i$ and $u_i^{(k)}$. By (5), $\left(\phi_i^{(k)}\right)^2 \to \phi_i^2$. Using this, $u_i^{(k_1)}$ and $u_j^{(k_2)}$ can be expressed as

$$u_i^{(k_1)} = \phi_i^{(k_1)} v_i + \sqrt{1 - (\phi_i^{(k_1)})^2}\, r_i^{(k_1)}, \tag{45}$$

$$u_j^{(k_2)} = \phi_j^{(k_2)} v_j + \sqrt{1 - (\phi_j^{(k_2)})^2}\, r_j^{(k_2)}, \tag{46}$$

where $r_i^{(k_1)}$ and $r_j^{(k_2)}$ are random vectors orthogonal to $v_i$ and $v_j$, respectively. The inner product between $u_i^{(k_1)}$ and $u_j^{(k_2)}$ is given by

$$\begin{aligned}
\langle u_i^{(k_1)}, u_j^{(k_2)} \rangle &= \langle \phi_i^{(k_1)} v_i, \phi_j^{(k_2)} v_j \rangle + \langle \phi_i^{(k_1)} v_i, \sqrt{1 - (\phi_j^{(k_2)})^2}\, r_j^{(k_2)} \rangle \\
&+ \langle \sqrt{1 - (\phi_i^{(k_1)})^2}\, r_i^{(k_1)}, \phi_j^{(k_2)} v_j \rangle + \langle \sqrt{1 - (\phi_i^{(k_1)})^2}\, r_i^{(k_1)}, \sqrt{1 - (\phi_j^{(k_2)})^2}\, r_j^{(k_2)} \rangle.
\end{aligned} \tag{47}$$

Since $r_i^{(k_1)}$ and $r_j^{(k_2)}$ are uniformly distributed on the subspaces orthogonal to $v_i$ and $v_j$, respectively, all cross terms involving $r_i^{(k_1)}$ and $r_j^{(k_2)}$ average to zero as $R \to \infty$. Consequently, the expected value reduces to $\mathbb{E}\left[\langle u_i^{(k_1)}, u_j^{(k_2)} \rangle\right] \to \phi_i \phi_j \mathbb{1}\{i = j\}$.

For eigenvectors $w_i^{(k_1)}$ and $w_j^{(k_2)}$, these are independent random vectors uniformly distributed within $(U^{(k_1)})^\perp$ and $(U^{(k_2)})^\perp$, respectively, i.e., they are chosen uniformly on the sphere in $(U^{(k_1)})^\perp$ and $(U^{(k_2)})^\perp$. Due to the rotational symmetry of these spaces, the expected inner product averages to zero:

$$\mathbb{E}\left[\langle w_i^{(k_1)}, w_j^{(k_2)} \rangle\right] \to 0. \tag{48}$$

Similarly, since $w_i^{(k_1)} \perp U_N^{(k_1)}$ and $u_j^{(k_2)} \in U_N^{(k_2)}$, the expected inner product between $w_i^{(k_1)}$ and $u_j^{(k_2)}$ also averages to zero:

$$\mathbb{E}[\langle u_i^{(k_1)}, w_j^{(k_2)} \rangle] \to 0. \tag{49}$$

$\square$

Now we provide the proof for Theorem 3.3.

*Proof.* Let $\alpha_i^R = \mathbb{E}[\partial_{u_i^{(k)}} f_N^{(k)}(h^R, \omega^R)]$ and $\beta = \mathbb{E}[\partial_{w_i^{(k)}} f_N^{(k)}(h^R, \omega^R)]$. Since the dataset $\mathcal{D}_N^{(k)}$, $\forall k \in \{1, \cdots, K\}$, is random and the eigenvector distributions are identical across clients, the expected values of the directional derivatives are the same for all clients. The cosine similarity between two rank-$r$ approximations of client $k_1$ and $k_2$ is $C_{N,r}^R(k_1, k_2) = \cos\left(\nabla \hat{f}_{N,r}^{(k_1)}, \nabla \hat{f}_{N,r}^{(k_2)}\right) = \frac{\langle \nabla \hat{f}_{N,r}^{(k_1)}, \nabla \hat{f}_{N,r}^{(k_2)} \rangle}{\|\nabla \hat{f}_{N,r}^{(k_1)}\| \cdot \|\nabla \hat{f}_{N,r}^{(k_2)}\|}$.

(i) For $r < \tilde{r}$, we can write rank-$r$ approximation of $\nabla f_N^{(k_1)}(h^R, \omega^R)$ and $\nabla f_N^{(k_2)}(h^R, \omega^R)$ as

$$\nabla \hat{f}_{N,r}^{(k_1)} = \sum_{i \leq r} \partial_{u_i^{(k_1)}} f_N^{(k_1)}(h^R, \omega^R)\, u_i^{(k_1)},$$

$$\nabla \hat{f}_{N,r}^{(k_2)} = \sum_{i \leq r} \partial_{u_i^{(k_2)}} f_N^{(k_2)}(h^R, \omega^R)\, u_i^{(k_2)}.$$

We drop $h^R$ and $\omega^R$ since the context is clear. The expectation of the cosine similarity between these two rank-$r$ approximations is

$$\mathbb{E}\left[C_{N,r}^R(k_1, k_2)\right] = \mathbb{E}\left[\frac{\langle \sum_{i \leq r} \partial_{u_i^{(k_1)}} f_N^{(k_1)} u_i^{(k_1)}, \sum_{i \leq r} \partial_{u_i^{(k_2)}} f_N^{(k_2)} u_i^{(k_2)} \rangle}{\|\nabla \hat{f}_{N,r}^{(k_1)}\| \cdot \|\nabla \hat{f}_{N,r}^{(k_2)}\|}\right]. \tag{50}$$

The denominator in (50) is $\mathbb{E}\left[\|\nabla \hat{f}_{N,r}^{(k_1)}\| \cdot \|\nabla \hat{f}_{N,r}^{(k_2)}\|\right] = \sum_{i \leq r}(\alpha_i^R)^2$ because of the independence between client $k_1$ and $k_2$. The numerator can be expressed as

$$\mathbb{E}\left[\langle \sum_{i \leq r} \partial_{u_i^{(k_1)}} f_N^{(k_1)} u_i^{(k_1)}, \sum_{i \leq r} \partial_{u_i^{(k_2)}} f_N^{(k_2)} u_i^{(k_2)} \rangle\right]$$

$$= \mathbb{E}\left[\sum_{i \leq r} \partial_{u_i^{(k_1)}} f_N^{(k_1)} \partial_{u_i^{(k_2)}} f_N^{(k_2)} \langle u_i^{(k_1)}, u_i^{(k_2)} \rangle\right] + \mathbb{E}\left[\sum_{i \neq j \leq r} \partial_{u_i^{(k_1)}} f_N^{(k_1)} \partial_{u_j^{(k_2)}} f_N^{(k_2)} \langle u_i^{(k_1)}, u_j^{(k_2)} \rangle\right] \tag{51}$$

By Lemma A.6, we know $\mathbb{E}\left[\langle u_i^{(k_1)}, u_i^{(k_2)} \rangle\right] \to \phi_i^2$ and $\mathbb{E}\left[\langle u_i^{(k_1)}, u_j^{(k_2)} \rangle\right] \to 0$ for $i \neq j$, thus the numerator satisfies

$$\left| \mathbb{E}\left[\langle \sum_{i \leq r} \partial_{u_i^{(k_1)}} f_N^{(k_1)} u_i^{(k_1)}, \sum_{i \leq r} \partial_{u_i^{(k_2)}} f_N^{(k_2)} u_i^{(k_2)} \rangle\right] - \sum_{i \leq r}(\alpha_i^R)^2 \phi_i^2 \right| \to 0. \tag{52}$$

Therefore we have

$$\left| \mathbb{E}\left[C_{N,r}^R(k_1, k_2)\right] - \frac{\sum_{i \leq r}(\alpha_i^R)^2 \phi_i^2}{\sum_{i \leq r}(\alpha_i^R)^2} \right| \to 0. \tag{53}$$

Finally, we have the following term

$$\left| \mathbb{E}\left[C_{N,r}^R(k_1, k_2) - C_{N,r+1}^R(k_1, k_2)\right] - \frac{(\alpha_{r+1}^R)^2 \sum_{i \leq r}(\alpha_i^R)^2(\phi_i^2 - \phi_{r+1}^2)}{\sum_{i \leq r}(\alpha_i^R)^2 \cdot \sum_{i \leq r+1}(\alpha_i^R)^2} \right| \to 0 \tag{54}$$

and here $g(r) = \frac{(\alpha_{r+1}^R)^2 \sum_{i \leq r}(\alpha_i^R)^2(\phi_i^2 - \phi_{r+1}^2)}{\sum_{i \leq r}(\alpha_i^R)^2 \cdot \sum_{i \leq r+1}(\alpha_i^R)^2}$ is strictly positive since $\phi_i^2 = 1 - \frac{\sigma^2 s_N^2}{\theta_i^2} \geq 1 - \frac{\sigma^2 s_N^2}{\theta_{r+1}^2} = \phi_{r+1}^2$.

(ii) For $r \geq \tilde{r}$, we can write rank-$r$ approximation of $\nabla f_N^{(k_1)}(h^R, \omega^R)$ and $\nabla f_N^{(k_2)}(h^R, \omega^R)$ as

$$\nabla \hat{f}_{N,r}^{(k_1)} = \sum_{i \leq \tilde{r}} \partial_{u_i^{(k_1)}} f_N^{(k_1)} u_i^{(k_1)} + \sum_{i \leq r-\tilde{r}} \partial_{w_i^{(k_1)}} f_N^{(k_1)} w_i^{(k_1)},$$

$$\nabla \hat{f}_{N,r}^{(k_2)} = \sum_{i \leq \tilde{r}} \partial_{u_i^{(k_2)}} f_N^{(k_2)} u_i^{(k_2)} + \sum_{i \leq r-\tilde{r}} \partial_{w_i^{(k_2)}} f_N^{(k_2)} w_i^{(k_2)}.$$

By the same argument in (i), we have

$$\mathbb{E}\left[C^R_{N,r}(k_1,k_2)\right] = \mathbb{E}\left[\frac{\langle \sum_{i\leq\tilde r}\partial_{u_i^{(k_1)}}f_N^{(k_1)}u_i^{(k_1)} + \sum_{i\leq r-\tilde r}\partial_{w_i^{(k_1)}}f_N^{(k_1)}w_i^{(k_1)}, \sum_{i\leq\tilde r}\partial_{u^{(k_2)}}f_N^{(k_2)}u_i^{(k_2)} + \sum_{i\leq r-\tilde r}\partial_{w_i^{(k_2)}}f_N^{(k_2)}\rangle}{\|\nabla\hat f_{N,r}^{(k_1)}\|\cdot\|\nabla\hat f_{N,r}^{(k_2)}\|}\right].$$
(55)

and by applying Lemma A.6, we have

$$\left|\mathbb{E}\left[C^R_{N,r}(k_1,k_2)\right] - \frac{\sum_{i\leq\tilde r}(\alpha_i^R)^2\phi_i^2}{(r-\tilde r)\beta^2 + \sum_{i\leq\tilde r}(\alpha_i^R)^2}\right| \to 0.$$
(56)

Finally,

$$\left|\mathbb{E}\left[C^R_{N,r}(k_1,k_2) - C^R_{N,r+1}(k_1,k_2)\right] - \frac{\beta^2\sum_{i\leq\tilde r}(\alpha_i^R)^2\phi_i^2}{\left((r-\tilde r)\beta^2 + \sum_{i\leq\tilde r}(\alpha_i^R)^2\right)\left((r-\tilde r+1)\beta^2 + \sum_{i\leq\tilde r}(\alpha_i^R)^2\right)}\right| \to 0 \quad (57)$$

and here $g(r) = \frac{\beta^2\sum_{i\leq\tilde r}(\alpha_i^R)^2\phi_i^2}{\left((r-\tilde r)\beta^2+\sum_{i\leq\tilde r}(\alpha_i^R)^2\right)\left((r-\tilde r+1)\beta^2+\sum_{i\leq\tilde r}(\alpha_i^R)^2\right)}$ is strictly positive.

$\square$

### A.6 Discussion on Theorem 3.3

One major limitation of Theorem 3.3 is that it analyzes the alignment between the gradient approximations of the clients based on a single update step. However, in practical federated learning settings, each local client typically performs multiple gradient descent steps during each round. Thus, it does not directly guarantee that the overall alignment of the representative gradients, defined as the difference between a client's updated model and the global model after one round of training, would exhibit the same behavior. Further analysis is needed, but we leave this for future research.

## B Convergence Analysis

In this appendix, we establish the convergence of FedLoRU in the case $\alpha = 1$ (the extension to $\alpha \neq 1$ is straightforward). We begin by restating the required assumptions.

**Assumption B.1** (Smoothness). Each local objective function $f^{(k)}$ is $L$-smooth, that is, for all $W$ and $W'$,

$$\|\nabla f^{(k)}(W) - \nabla f^{(k)}(W')\|_F \leq L\,\|W - W'\|_F.$$
(A.1)

**Assumption B.2** (Trajectory-wise bounded stochastic-gradient norm). Let $\{W_{t,i}^{(k)}\}$ denote the (random) sequence of model points at which client $k$ evaluates stochastic gradients during the execution of the algorithm. There exists a constant $G > 0$ such that for all clients $k$, all communication rounds $t$, and all local steps $i$,

$$\mathbb{E}\left\|\nabla f^{(k)}\left(W_{t,i}^{(k)}\right)\right\|_F^2 \leq G^2.$$
(A.2)

**Assumption B.3** (Bounded low-rank matrices)**.** The local low-rank update matrices remain uniformly bounded: there exist constants $C_A > 0$ and $C_B > 0$ such that for all communication rounds $t$, local update steps $i$, and clients $k$,

$$\|A_{t,i}^{(k)}\|_F \le C_A, \quad \|B_{t,i}^{(k)}\|_F \le C_B. \tag{A.3}$$

**Assumption B.4** (Unbiasedness and bounded variance)**.** The stochastic gradient estimator is unbiased and has bounded variance, that is, for all $W$ there exists $\sigma^2 > 0$ such that

$$\mathbb{E}\big[\nabla f^{(k)}(W; \xi^{(k)})\big] = \nabla f^{(k)}(W), \tag{A.4}$$

$$\mathbb{E}\big\|\nabla f^{(k)}(W; \xi^{(k)}) - \nabla f^{(k)}(W)\big\|_F^2 \le \sigma^2. \tag{A.5}$$

Assumptions (A.1) - (A.5) respectively state that each local objective is $L$-smooth, that the gradient's second moment along the trajectory is bounded by $G^2$, that the low-rank matrices $A_{t,i}^{(k)}$ and $B_{t,i}^{(k)}$ remain bounded by $C_A$ and $C_B$, and that the stochastic gradient estimator is unbiased with variance at most $\sigma^2$. Moreover, since we reinitialize $A$ and $B$ at each accumulation cycle, they can accumulate at most $\tau E$ gradient steps before being reinitialized. Under Assumption A.2, each stochastic gradient has norm at most $G$, and by choosing a sufficiently small step size $\eta$, each update increases the Frobenius norm only modestly, which can justify Assumption B.3.

We now introduce the notation used throughout the convergence proof. At communication round $t$, let $W_t$, $A_t$, $B_t$ denote the initial weight and low-rank update matrices. For brevity, with a slight abuse of notation, we write $f(W_t + AB)$ as $f_t(AB)$ and $f^{(k)}(W_t + AB)$ as $f_t^{(k)}(AB)$. Moreover, unless otherwise indicated, all matrix norms $\|\cdot\|$ denote the Frobenius norm.

During local training on client $k$, we perform $E$ gradient steps on the factors $\{A_{t,i}^{(k)}, B_{t,i}^{(k)}\}_{i=1}^E$, using independent mini-batches $\xi_{t,i}^{(k)}$. We then define the averaged accumulated low-rank updates of each client $k$

$$\tilde{\Delta}_{A,t}^{(k)} = \frac{1}{E} \sum_{i=1}^E \nabla_A f_t^{(k)}(A_{t,i}^{(k)} B_{t,i}^{(k)}; \xi_{t,i}^{(k)}), \quad \tilde{\Delta}_{B,t}^{(k)} = \frac{1}{E} \sum_{i=1}^E \nabla_B f_t^{(k)}(A_{t,i}^{(k)} B_{t,i}^{(k)}; \xi_{t,i}^{(k)}), \tag{58}$$

and their non-stochastic analogues

$$\Delta_{A,t}^{(k)} = \frac{1}{E} \sum_{i=1}^E \nabla_A f_t^{(k)}(A_{t,i}^{(k)} B_{t,i}^{(k)}), \quad \Delta_{B,t}^{(k)} = \frac{1}{E} \sum_{i=1}^E \nabla_B f_t^{(k)}(A_{t,i}^{(k)} B_{t,i}^{(k)}). \tag{59}$$

After all clients complete local training, the server aggregates via

$$A_{t+1} = \sum_{k \in \mathcal{K}_t} p_t^{(k)} A_{t,E}^{(k)}, \quad B_{t+1} = \sum_{k \in \mathcal{K}_t} p_t^{(k)} B_{t,E}^{(k)}.$$

We can also express the update of each low-rank matrices as sum of the averaged accumulated low-rank updates as

$$A_{t+1} - A_t = -\eta E \sum_{k \in \mathcal{K}_t} p_t^{(k)} \tilde{\Delta}_{A,t}^{(k)}, \tag{60}$$

$$B_{t+1} - B_t = -\eta E \sum_{k \in \mathcal{K}_t} p_t^{(k)} \tilde{\Delta}_{B,t}^{(k)}. \tag{61}$$

Whenever $(t+1) \bmod \tau = 0$ (the accumulation cycle), we add the the low-rank matrices into the global model by $W_{t+1} = W_t + A_{t+1} B_{t+1}$, then reinitialize the low-rank matrices so that they satisfy $\bar{A}_{t+1} \bar{B}_{t+1} = 0$. We use a bar notation to distinguish the post-aggregation factors $A_{t+1}, B_{t+1}$ from their reinitialized counterparts $\bar{A}_{t+1}, \bar{B}_{t+1}$ at each accumulation cycle. By construction $f(W_t + A_{t+1} B_{t+1}) = f(W_{t+1} + \bar{A}_{t+1} \bar{B}_{t+1})$, so this re-initialization does not affect the loss.

### B.1 Technical Lemmas

We next collect several foundational results that will be used in the convergence proof.

**Lemma B.5** (Partial Smoothness). *Assume Assumption B.1 and Assumption B.3. Each local loss $f^{(k)}$ is $LC_B^2$-smooth in $A$ (for fixed $B$) and $LC_A^2$-smooth in $B$ (for fixed $A$). Consequently, the global objective $f$ inherits the same smoothness properties.*

*Proof.* Fix client $k$, round $t$, matrices $A, A'$ and $B$. By the chain rule and the definition of $\nabla_A$,

$$\nabla_A f_t^{(k)}(AB) = \nabla_W f_t^{(k)}(AB)\, B^\top,$$

so

$$\left\|\nabla_A f_t^{(k)}(AB) - \nabla_A f_t^{(k)}(A'B)\right\| = \left\|(\nabla_W f_t^{(k)}(AB) - \nabla_W f_t^{(k)}(A'B))B^\top\right\|.$$

Applying submultiplicativity of the Frobenius norm and the $L$-smoothness of $f^{(k)}$ in $W$ (Assumption B.1) gives

$$\begin{aligned}
\left\|\nabla_A f_t^{(k)}(AB) - \nabla_A f_t^{(k)}(A'B)\right\| &\le \left\|\nabla_W f_t^{(k)}(AB)B^\top - \nabla_W f_t^{(k)}(A'B)B^\top\right\| \\
&\le \left\|\nabla_W f_t^{(k)}(AB) - \nabla_W f_t^{(k)}(A'B)\right\| \|B\| \\
&\le LC_B \|AB - A'B\| \\
&\le LC_B^2 \|A - A'\|.
\end{aligned}$$

An identical argument establishes that $f_t^{(k)}$ is $LC_A^2$-smooth in $B$ when $A$ is held fixed. $\qquad\square$

**Lemma B.6.** *For any $t$, we have the following bound:*

$$\mathbb{E}\left[\|\nabla_B f_t(A_{t+1}B_t) - \nabla_B f_t(A_t B_t)\|^2\right] \le 2(L^2 C_A^2 C_B^2 + G^2)\mathbb{E}\|A_{t+1} - A_t\|^2.$$

*Proof.* Fix $t$. Noticing that $\nabla_B f_t(AB) = A^\top \nabla_W f_t(AB)$, we have

$$\begin{aligned}
\mathbb{E}&\|\nabla_B f_t(A_{t+1}B_t) - \nabla_B f_t(A_t B_t)\|^2 \\
&= \mathbb{E}\|A_{t+1}^\top \nabla_W f_t(A_{t+1}B_t) - A_t^\top \nabla_W f_t(A_t B_t)\|^2 \\
&= \mathbb{E}\|A_{t+1}^\top[\nabla_W f_t(A_{t+1}B_t) - \nabla_W f_t(A_t B_t)] + (A_{t+1} - A_t)^\top \nabla_W f_t(A_t B_t)\|^2 \\
&\le 2\,\mathbb{E}\|A_{t+1}^\top[\nabla_W f_t(A_{t+1}B_t) - \nabla_W f_t(A_t B_t)]\|^2 + 2\,\mathbb{E}\|(A_{t+1} - A_t)^\top \nabla_W f_t(A_t B_t)\|^2,
\end{aligned}$$

where we used $\|X + Y\|^2 \le 2\|X\|^2 + 2\|Y\|^2$.

Since $f_t$ is $L$-smooth with respect to $W$ and low-rank matrices are uniformly bounded,

$$\begin{aligned}
\mathbb{E}\left\|A_{t+1}^\top\left(\nabla_W f_t(A_{t+1}B_t) - \nabla_W f_t(A_t B_t)\right)\right\|^2 &\le \mathbb{E}\|A_{t+1}\|^2\,\mathbb{E}\left\|\nabla_W f_t(A_{t+1}B_t) - \nabla_W f_t(A_t B_t)\right\|^2 \\
&\le L^2 C_A^2\,\mathbb{E}\|A_{t+1}B_t - A_t B_t\|^2 \\
&\le L^2 C_A^2 C_B^2\,\mathbb{E}\|A_{t+1} - A_t\|^2.
\end{aligned}$$

By Assumption B.2, we have

$$\mathbb{E}\left\|(A_{t+1}-A_t)^\top \nabla_W f_t(A_t B_t)\right\|^2 = \mathbb{E}\left[\|A_{t+1}-A_t\|^2\,\|\nabla_W f_t(A_t B_t)\|^2\right]$$
$$\leq G^2 \mathbb{E}\,\|A_{t+1}-A_t\|^2.$$

Putting these together gives

$$\mathbb{E}\|\nabla_B f_t(A_{t+1}B_t) - \nabla_B f_t(A_t B_t)\|^2 \leq 2\left(L^2 C_A^2 C_B^2 + G^2\right)\mathbb{E}\|A_{t+1}-A_t\|^2,$$

as claimed. $\qquad\square$

**Lemma B.7** (Drift of aggregated and local factors). *Assume Assumptions B.2–B.4. Then for every communication round $t$,*

$$\mathbb{E}\|A_{t+1}-A_t\|_F^2 \leq 2\eta^2 E^2 (\sigma^2 + G^2)\, C_B^2,$$
$$\mathbb{E}\|B_{t+1}-B_t\|_F^2 \leq 2\eta^2 E^2 (\sigma^2 + G^2)\, C_A^2.$$

*Moreover, for any client $k$ and any local step $i \leq E$,*

$$\mathbb{E}\|A_{t,i}^{(k)} - A_{t,0}^{(k)}\|_F^2 \leq 2\eta^2 i^2 (\sigma^2 + G^2)\, C_B^2,$$
$$\mathbb{E}\|B_{t,i}^{(k)} - B_{t,0}^{(k)}\|_F^2 \leq 2\eta^2 i^2 (\sigma^2 + G^2)\, C_A^2.$$

*Proof.* We prove the bounds for $A$; the bounds for $B$ are identical by symmetry.

From (60),

$$A_{t+1} - A_t = -\eta E \sum_{k\in\mathcal{K}_t} p_t^{(k)}\,\tilde{\Delta}_{A,t}^{(k)}.$$

Since $x \mapsto \|x\|_F^2$ is convex and $\sum_{k\in\mathcal{K}_t} p_t^{(k)} = 1$, we apply Jensen's inequality to have

$$\mathbb{E}\|A_{t+1}-A_t\|^2 = \eta^2 E^2\,\mathbb{E}\bigg\|\sum_{k\in\mathcal{K}_t} p_t^{(k)}\tilde{\Delta}_{A,t}^{(k)}\bigg\|^2 \leq \eta^2 E^2\,\mathbb{E}\bigg[\sum_{k\in\mathcal{K}_t} p_t^{(k)}\|\tilde{\Delta}_{A,t}^{(k)}\|^2\bigg]. \qquad (62)$$

Fix any client $k$. Since

$$\tilde{\Delta}_{A,t}^{(k)} = \frac{1}{E}\sum_{i=1}^{E}\nabla_A f_t^{(k)}\big(A_{t,i}^{(k)} B_{t,i}^{(k)};\xi_{t,i}^{(k)}\big),$$

another application of Jensen yields

$$\mathbb{E}\|\tilde{\Delta}_{A,t}^{(k)}\|^2 \leq \frac{1}{E}\sum_{i=1}^{E}\mathbb{E}\Big\|\nabla_A f_t^{(k)}\big(A_{t,i}^{(k)} B_{t,i}^{(k)};\xi_{t,i}^{(k)}\big)\Big\|^2. \qquad (63)$$

Using $\nabla_A f_t^{(k)}(AB;\xi) = \nabla_W f_t^{(k)}(AB;\xi)\,B^\top$ and $\|XY^\top\| \leq \|X\|\,\|Y\|$, we have

$$\Big\|\nabla_A f_t^{(k)}\big(A_{t,i}^{(k)} B_{t,i}^{(k)};\xi_{t,i}^{(k)}\big)\Big\|^2 \leq \Big\|\nabla_W f_t^{(k)}\big(A_{t,i}^{(k)} B_{t,i}^{(k)};\xi_{t,i}^{(k)}\big)\Big\|^2 \|B_{t,i}^{(k)}\|^2$$
$$\leq C_B^2\Big\|\nabla_W f_t^{(k)}\big(A_{t,i}^{(k)} B_{t,i}^{(k)};\xi_{t,i}^{(k)}\big)\Big\|^2, \qquad (64)$$

where we used Assumption B.3 in the second inequality.

Let

$$\tilde{\nabla}_W f_{t,i}^{(k)} := \nabla_W f_t^{(k)}\big(A_{t,i}^{(k)} B_{t,i}^{(k)};\xi_{t,i}^{(k)}\big), \qquad \nabla_W f_{t,i}^{(k)} := \nabla_W f_t^{(k)}\big(A_{t,i}^{(k)} B_{t,i}^{(k)}\big).$$

Then by $\|u + v\|^2 \leq 2\|u\|^2 + 2\|v\|^2$, Assumption B.4, and Assumption B.2,

$$
\begin{aligned}
\mathbb{E}\|\tilde{\nabla}_W f_{t,i}^{(k)}\|^2 &= \mathbb{E}\|\tilde{\nabla}_W f_{t,i}^{(k)} - \nabla_W f_{t,i}^{(k)} + \nabla_W f_{t,i}^{(k)}\|^2 \\
&\leq 2\,\mathbb{E}\|\tilde{\nabla}_W f_{t,i}^{(k)} - \nabla_W f_{t,i}^{(k)}\|^2 + 2\,\mathbb{E}\|\nabla_W f_{t,i}^{(k)}\|^2 \\
&\leq 2\sigma^2 + 2G^2.
\end{aligned}
\tag{65}
$$

Combining (63)–(65) gives

$$
\mathbb{E}\|\tilde{\Delta}_{A,t}^{(k)}\|^2 \leq \frac{1}{E} \sum_{i=1}^{E} 2(\sigma^2 + G^2)\, C_B^2 = 2(\sigma^2 + G^2)\, C_B^2.
$$

Substituting this into (62) and using $\sum_{k \in \mathcal{K}_t} p_t^{(k)} = 1$ yields

$$
\mathbb{E}\|A_{t+1} - A_t\|^2 \leq \eta^2 E^2\, \mathbb{E}\left[\sum_{k \in \mathcal{K}_t} p_t^{(k)} \cdot 2(\sigma^2 + G^2)\, C_B^2\right] = 2\eta^2 E^2 (\sigma^2 + G^2)\, C_B^2.
$$

For the local iterates, for any client $k$ and $i \leq E$,

$$
A_{t,i}^{(k)} - A_{t,0}^{(k)} = -\eta \sum_{j=1}^{i} \nabla_A f_t^{(k)}\big(A_{t,j}^{(k)} B_{t,j}^{(k)}; \xi_{t,j}^{(k)}\big).
$$

Using $\|\sum_{j=1}^{i} X_j\|^2 \leq i \sum_{j=1}^{i} \|X_j\|^2$ and the bound $\mathbb{E}\|\nabla_A f_t^{(k)}(A_{t,j}^{(k)} B_{t,j}^{(k)}; \xi_{t,j}^{(k)})\|^2 \leq 2(\sigma^2 + G^2)C_B^2$ (from the same argument above), we obtain

$$
\mathbb{E}\|A_{t,i}^{(k)} - A_{t,0}^{(k)}\|^2 \leq \eta^2 i \sum_{j=1}^{i} 2(\sigma^2 + G^2)\, C_B^2 = 2\eta^2 i^2 (\sigma^2 + G^2)\, C_B^2.
$$

The bounds for $B_{t+1} - B_t$ and $B_{t,i}^{(k)} - B_{t,0}^{(k)}$ follow identically, replacing $C_B$ by $C_A$ and using $\nabla_B f_t^{(k)}(AB; \xi) = A^\top \nabla_W f_t^{(k)}(AB; \xi)$. $\qquad\square$

**Lemma B.8.** *For any $t$, we have*

$$
\mathbb{E}\left\| \sum_{k \in \mathcal{K}_t} p_t^{(k)} \Delta_{A,t}^{(k)} - \nabla_A f_t(A_t B_t) \right\|^2 \leq 4\eta^2 D_A,
$$

$$
\mathbb{E}\left\| \sum_{k \in \mathcal{K}_t} p_t^{(k)} \Delta_{B,t}^{(k)} - \nabla_B f_t(A_t B_t) \right\|^2 \leq 4\eta^2 D_B,
$$

*where*

$$
D_A := E^2(\sigma^2 + G^2)\Big(2L^2\, C_B^2(C_A^4 + C_B^4) + G^2\, C_A^2\Big), \qquad D_B := E^2(\sigma^2 + G^2)\Big(2L^2\, C_A^2(C_A^4 + C_B^4) + G^2\, C_B^2\Big).
$$

*Proof.* Observe that

$$
\nabla_A f_t(A_t B_t) = \sum_{k \in \mathcal{K}_t} p_t^{(k)}\, \nabla_A f_{t,0}^{(k)},
$$

and therefore

$$
\begin{aligned}
\sum_{k \in \mathcal{K}_t} p_t^{(k)} \Delta_{A,t}^{(k)} - \nabla_A f_t(A_t B_t) &= \sum_{k \in \mathcal{K}_t} p_t^{(k)}\Big(\Delta_{A,t}^{(k)} - \nabla_A f_{t,0}^{(k)}\Big) \\
&= \sum_{k \in \mathcal{K}_t} p_t^{(k)} \frac{1}{E} \sum_{i=1}^{E} \big[\nabla_A f_{t,i}^{(k)} - \nabla_A f_{t,0}^{(k)}\big].
\end{aligned}
$$

Since $x \mapsto \|x\|^2$ is convex and $\sum_{k \in \mathcal{K}_t} p_t^{(k)} = 1$, Jensen's inequality yields

$$
\mathbb{E}\Big\| \sum_{k \in \mathcal{K}_t} p_t^{(k)} \Delta_{A,t}^{(k)} - \nabla_A f_t(A_t B_t) \Big\|^2 \leq \mathbb{E}\Big[ \sum_{k \in \mathcal{K}_t} p_t^{(k)} \Big\| \frac{1}{E} \sum_{i=1}^{E} (\nabla_A f_{t,i}^{(k)} - \nabla_A f_{t,0}^{(k)}) \Big\|^2 \Big]
$$

$$
\leq \mathbb{E}\Big[ \sum_{k \in \mathcal{K}_t} p_t^{(k)} \frac{1}{E} \sum_{i=1}^{E} \| \nabla_A f_{t,i}^{(k)} - \nabla_A f_{t,0}^{(k)} \|^2 \Big]. \tag{66}
$$

Using $\nabla_A f = \nabla_W f \, B^\top$, we can expand the difference as

$$
\nabla_A f_{t,i}^{(k)} - \nabla_A f_{t,0}^{(k)} = (\nabla_W f_{t,i}^{(k)} - \nabla_W f_{t,0}^{(k)})(B_{t,i}^{(k)})^\top + \nabla_W f_{t,0}^{(k)}(B_{t,i}^{(k)} - B_{t,0}^{(k)})^\top.
$$

Hence, by $\|X + Y\|^2 \leq 2\|X\|^2 + 2\|Y\|^2$,

$$
\mathbb{E}\Big\| \sum_{k \in \mathcal{K}_t} p_t^{(k)} \Delta_{A,t}^{(k)} - \nabla_A f_t(A_t B_t) \Big\|^2 \leq \mathbb{E}\Big[ \sum_{k \in \mathcal{K}_t} p_t^{(k)} \frac{1}{E} \sum_{i=1}^{E} \Big( 2\|\nabla_W f_{t,i}^{(k)} - \nabla_W f_{t,0}^{(k)}\|^2 \, \|B_{t,i}^{(k)}\|^2
$$

$$
+ 2\|\nabla_W f_{t,0}^{(k)}\|^2 \, \|B_{t,i}^{(k)} - B_{t,0}^{(k)}\|^2 \Big) \Big]. \tag{67}
$$

Finally, invoking $\|B_{t,i}^{(k)}\| \leq C_B$ and $\mathbb{E}\|\nabla_W f_{t,0}^{(k)}\|^2 \leq G^2$ gives

$$
\mathbb{E}\Big\| \sum_{k \in \mathcal{K}_t} p_t^{(k)} \Delta_{A,t}^{(k)} - \nabla_A f_t(A_t B_t) \Big\|^2 \leq \mathbb{E}\Big[ \sum_{k \in \mathcal{K}_t} p_t^{(k)} \frac{1}{E} \sum_{i=1}^{E} \Big( 2C_B^2 \, \|\nabla_W f_{t,i}^{(k)} - \nabla_W f_{t,0}^{(k)}\|^2 + 2G^2 \, \|B_{t,i}^{(k)} - B_{t,0}^{(k)}\|^2 \Big) \Big]. \tag{68}
$$

By Assumption B.1 and Assumption B.3,

$$
\begin{aligned}
\|\nabla_W f_{t,i}^{(k)} - \nabla_W f_{t,0}^{(k)}\|^2 &\leq L^2 \|A_{t,i}^{(k)} B_{t,i}^{(k)} - A_{t,0}^{(k)} B_{t,0}^{(k)}\|^2 \\
&= L^2 \|(A_{t,i}^{(k)} - A_{t,0}^{(k)}) B_{t,i}^{(k)} + A_{t,0}^{(k)}(B_{t,i}^{(k)} - B_{t,0}^{(k)})\|^2 \\
&\leq 2L^2 \|(A_{t,i}^{(k)} - A_{t,0}^{(k)}) B_{t,i}^{(k)}\|^2 + 2L^2 \|A_{t,0}^{(k)}(B_{t,i}^{(k)} - B_{t,0}^{(k)})\|^2 \\
&\leq 2L^2 C_B^2 \|A_{t,i}^{(k)} - A_{t,0}^{(k)}\|^2 + 2L^2 C_A^2 \|B_{t,i}^{(k)} - B_{t,0}^{(k)}\|^2.
\end{aligned}
$$

Substituting this into (68) yields

$$
\mathbb{E}\Big\| \sum_{k \in \mathcal{K}_t} p_t^{(k)} \Delta_{A,t}^{(k)} - \nabla_A f_t(A_t B_t) \Big\|^2 \leq \mathbb{E}\Big[ \sum_{k \in \mathcal{K}_t} p_t^{(k)} \frac{1}{E} \sum_{i=1}^{E} \Big( 4L^2 C_B^4 \|A_{t,i}^{(k)} - A_{t,0}^{(k)}\|^2 + (4L^2 C_A^2 C_B^2 + 2G^2) \|B_{t,i}^{(k)} - B_{t,0}^{(k)}\|^2 \Big) \Big].
$$

Applying Lemma B.7 and using $\frac{1}{E} \sum_{i=1}^{E} i^2 \leq E^2$ and $\sum_{k \in \mathcal{K}_t} p_t^{(k)} = 1$,

$$
\mathbb{E}\Big\| \sum_{k \in \mathcal{K}_t} p_t^{(k)} \Delta_{A,t}^{(k)} - \nabla_A f_t(A_t B_t) \Big\|^2 \leq 4L^2 C_B^4 \cdot 2\eta^2 E^2 (\sigma^2 + G^2) C_B^2 + (4L^2 C_A^2 C_B^2 + 2G^2) \cdot 2\eta^2 E^2 (\sigma^2 + G^2) C_A^2
$$

$$
= 4\eta^2 E^2 (\sigma^2 + G^2) \Big( 2L^2 C_B^2 (C_A^4 + C_B^4) + G^2 C_A^2 \Big) = 4\eta^2 D_A.
$$

The bound for $B$ is obtained analogously, which gives

$$
\mathbb{E}\Big\| \sum_{k \in \mathcal{K}_t} p_t^{(k)} \Delta_{B,t}^{(k)} - \nabla_B f_t(A_t B_t) \Big\|^2 \leq 4\eta^2 D_B.
$$

$\square$

## B.2 Proof of Theorem 4.2

*Proof.* Fix a communication round $t$ with initial parameters $(W_t, A_t, B_t)$. We obtain

$$
\begin{aligned}
\mathbb{E}\left[f_t(A_{t+1}B_{t+1})\right] &\overset{(a)}{\leq} \mathbb{E}\left[f_t(A_{t+1}B_t)\right] + \mathbb{E}\left[\langle \nabla_B f_t(A_{t+1}B_t), B_{t+1} - B_t\rangle\right] + \frac{LC_A^2}{2}\mathbb{E}\left\|B_{t+1} - B_t\right\|^2 \\
&= \mathbb{E}\left[f_t(A_{t+1}B_t)\right] + \frac{LC_A^2}{2}\mathbb{E}\left\|B_{t+1} - B_t\right\|^2 \\
&\quad + \mathbb{E}\left[\langle \nabla_B f_t(A_{t+1}B_t) - \nabla_B f_t(A_tB_t) + \nabla_B f_t(A_tB_t), B_{t+1} - B_t\rangle\right] \\
&= \mathbb{E}\left[f_t(A_{t+1}B_t)\right] + \frac{LC_A^2}{2}\mathbb{E}\left\|B_{t+1} - B_t\right\|^2 + \mathbb{E}\left[\langle \nabla_B f_t(A_tB_t), B_{t+1} - B_t\rangle\right] \\
&\quad + \mathbb{E}\left[\langle \nabla_B f_t(A_{t+1}B_t) - \nabla_B f_t(A_tB_t), B_{t+1} - B_t\rangle\right] \\
&\overset{(b)}{\leq} \mathbb{E}\left[f_t(A_{t+1}B_t)\right] + \frac{LC_A^2}{2}\mathbb{E}\left\|B_{t+1} - B_t\right\|^2 - \eta E\,\mathbb{E}\left[\langle \nabla_B f_t(A_tB_t), \sum_k p_t^{(k)}\tilde{\Delta}_{B,t}^{(k)}\rangle\right] \\
&\quad + \mathbb{E}\left\|B_{t+1} - B_t\right\|^2 + \frac{1}{4}\mathbb{E}\left[\left\|\nabla_B f_t(A_{t+1}B_t) - \nabla_B f_t(A_tB_t)\right\|^2\right],
\end{aligned}
\tag{69}
$$

where we apply Lemma B.5 in (a), and use inequality $\langle a, b\rangle \leq \frac{1}{4}\|a\|^2 + \|b\|^2$ and (61) in (b).

For the inner product term $\mathbb{E}\left[\langle \nabla_B f_t(A_tB_t), \sum_k p_t^{(k)}\tilde{\Delta}_{B,t}^{(k)}\rangle\right]$, we can take full expectation $\mathbb{E} = \mathbb{E}_{\xi_{t,1}}\ldots\mathbb{E}_{\xi_{t,E}}$ to get

$$
\begin{aligned}
\mathbb{E}\left[\langle \nabla_B f_t(A_tB_t), \sum_k p_t^{(k)}\tilde{\Delta}_{B,t}^{(k)}\rangle\right] &= \mathbb{E}_{\xi_{t,1}}\ldots\mathbb{E}_{\xi_{t,E}}\left[\langle \nabla_B f_t(A_tB_t), \sum_k p_t^{(k)}\frac{1}{E}\sum_{i=1}^E \nabla_B f_t^{(k)}(A_{t,i}^{(k)}B_{t,i}^{(k)}; \xi_{t,i}^{(k)})\rangle\right] \\
&= \mathbb{E}\left[\langle \nabla_B f_t(A_tB_t), \sum_k p_t^{(k)}\frac{1}{E}\sum_{i=1}^E \nabla_B f_t^{(k)}(A_{t,i}^{(k)}B_{t,i}^{(k)})\rangle\right] \\
&= \mathbb{E}\left[\langle \nabla_B f_t(A_tB_t), \sum_k p_t^{(k)}\Delta_{B,t}^{(k)}\rangle\right],
\end{aligned}
\tag{70}
$$

where $\xi_{t,i} = \{\xi_{t,i}^{(k)}\}_{k=1}^K$ is the set of random samples of all clients at communication round $t$ and local training round $i$. This follows from the fact that $\nabla_B f_t(A_tB_t)$ depends only on the history $\{\xi_\tau\}_{\tau=1}^{t-1}$, which is independent of $\xi_t = \{\xi_{t,i}\}_{i=1}^E$. Further, by applying $-\langle a, b\rangle \leq \frac{1}{4}\|a\|^2 + \|b\|^2$ again in (c), we have

$$
\begin{aligned}
&-\eta E\,\mathbb{E}\left[\langle \nabla_B f_t(A_tB_t), \sum_k p_t^{(k)}\Delta_{B,t}^{(k)}\rangle\right] \\
&= -\eta E\,\mathbb{E}\left[\langle \nabla_B f_t(A_tB_t), \sum_k p_t^{(k)}\Delta_{B,t}^{(k)} - \nabla_B f_t(A_tB_t) + \nabla_B f_t(A_tB_t)\rangle\right] \\
&= -\eta E\,\mathbb{E}\left[\langle \nabla_B f_t(A_tB_t), \sum_k p_t^{(k)}\Delta_{B,t}^{(k)} - \nabla_B f_t(A_tB_t)\rangle\right] - \eta E\,\mathbb{E}\left\|\nabla_B f_t(A_tB_t)\right\|^2 \\
&\overset{(c)}{\leq} \eta E\,\mathbb{E}\left[\left\|\sum_k p_t^{(k)}\Delta_{B,t}^{(k)} - \nabla_B f_t(A_tB_t)\right\|^2\right] - \frac{3\eta E}{4}\mathbb{E}\left\|\nabla_B f_t(A_tB_t)\right\|^2.
\end{aligned}
\tag{71}
$$

Combining (70) and (71) into (69) yields

$$
\begin{aligned}
\mathbb{E}\left[f_t(A_{t+1}B_{t+1})\right] &\leq \mathbb{E}\left[f_t(A_{t+1}B_t)\right] + \left(1 + \frac{LC_A^2}{2}\right)\mathbb{E}\left\|B_{t+1} - B_t\right\|^2 - \frac{3\eta E}{4}\mathbb{E}\left\|\nabla_B f_t(A_tB_t)\right\|^2 \\
&\quad + \frac{1}{4}\mathbb{E}\left[\left\|\nabla_B f_t(A_{t+1}B_t) - \nabla_B f_t(A_tB_t)\right\|^2\right] + \eta E\,\mathbb{E}\left[\left\|\sum_k p_t^{(k)}\Delta_{B,t}^{(k)} - \nabla_B f_t(A_tB_t)\right\|^2\right] \\
&\overset{(d)}{\leq} \mathbb{E}\left[f_t(A_{t+1}B_t)\right] + \left(1 + \frac{LC_A^2}{2}\right)\mathbb{E}\left\|B_{t+1} - B_t\right\|^2 - \frac{3\eta E}{4}\mathbb{E}\left\|\nabla_B f_t(A_tB_t)\right\|^2 \\
&\quad + \frac{1}{2}\left(L^2C_A^2C_B^2 + G^2\right)\mathbb{E}\left\|A_{t+1} - A_t\right\|^2 + \eta E\,\mathbb{E}\left[\left\|\sum_k p_t^{(k)}\Delta_{B,t}^{(k)} - \nabla_B f_t(A_tB_t)\right\|^2\right],
\end{aligned}
\tag{72}
$$

where we apply Lemma B.6 in (d).

Next, we bound the term $\mathbb{E}\left[f_t(A_{t+1}B_t)\right]$. By $LC_B^2$-smoothness in Lemma B.5, one has

$$
\begin{aligned}
\mathbb{E}\left[f_t(A_{t+1}B_t)\right] &\leq \mathbb{E}\left[f_t(A_tB_t)\right] + \mathbb{E}\left[\langle\nabla_A f_t(A_tB_t), A_{t+1} - A_t\rangle\right] + \frac{LC_B^2}{2}\mathbb{E}\left\|A_{t+1} - A_t\right\|^2 \\
&= \mathbb{E}\left[f_t(A_tB_t)\right] + \frac{LC_B^2}{2}\mathbb{E}\left\|A_{t+1} - A_t\right\|^2 - \eta E\,\mathbb{E}\left[\langle\nabla_A f_t(A_tB_t), \sum_k p_t^{(k)}\tilde{\Delta}_{A,t}^{(k)}\rangle\right].
\end{aligned}
\tag{73}
$$

Since $\nabla_A f_t(A_tB_t)$ is independent of the current mini-batches $\xi_{t,1}, \ldots, \xi_{t,E}$, taking full expectation $\mathbb{E} = \mathbb{E}_{\xi_{t,1}} \ldots \mathbb{E}_{\xi_{t,E}}$ yields

$$
\mathbb{E}\left[\langle\nabla_A f_t(A_tB_t), \sum_k p_t^{(k)}\tilde{\Delta}_{A,t}^{(k)}\rangle\right] = \mathbb{E}\left[\langle\nabla_A f_t(A_tB_t), \sum_k p_t^{(k)}\Delta_{A,t}^{(k)}\rangle\right].
\tag{74}
$$

Applying $-\langle a, b\rangle \leq \frac{1}{4}\|a\|^2 + \|b\|^2$, we obtain

$$
\begin{aligned}
&-\eta E\,\mathbb{E}\left[\langle\nabla_A f_t(A_tB_t), \sum_k p_t^{(k)}\Delta_{A,t}^{(k)}\rangle\right] \\
&= -\eta E\,\mathbb{E}\left[\langle\nabla_A f_t(A_tB_t), \sum_k p_t^{(k)}\Delta_{A,t}^{(k)} - \nabla_A f_t(A_tB_t) + \nabla_A f_t(A_tB_t)\rangle\right] \\
&= -\eta E\,\mathbb{E}\left[\langle\nabla_A f_t(A_tB_t), \sum_k p_t^{(k)}\Delta_{A,t}^{(k)} - \nabla_A f_t(A_tB_t)\rangle\right] - \eta E\,\mathbb{E}\left\|\nabla_A f_t(A_tB_t)\right\|^2 \\
&\leq \eta E\,\mathbb{E}\left[\left\|\sum_k p_t^{(k)}\Delta_{A,t}^{(k)} - \nabla_A f_t(A_tB_t)\right\|^2\right] - \frac{3\eta E}{4}\mathbb{E}\left\|\nabla_A f_t(A_tB_t)\right\|^2.
\end{aligned}
\tag{75}
$$

By combining (74) and (75) into (73), we get

$$\mathbb{E}\left[f_t(A_{t+1}B_t)\right] \le \mathbb{E}\left[f_t(A_tB_t)\right] + \frac{LC_B^2}{2}\mathbb{E}\left\|A_{t+1} - A_t\right\|^2$$
$$- \frac{3\eta E}{4}\mathbb{E}\left\|\nabla_A f_t(A_tB_t)\right\|^2 + \eta E\,\mathbb{E}\left[\left\|\sum_k p^{(k)}\Delta_{A,t}^{(k)} - \nabla_A f_t(A_tB_t)\right\|^2\right]. \tag{76}$$

By substituting (76) into (72), we have

$$\mathbb{E}\left[f_t(A_{t+1}B_{t+1})\right] \le \mathbb{E}\left[f_t(A_tB_t)\right] - \frac{3\eta E}{4}\left(\mathbb{E}\left\|\nabla_B f_t(A_tB_t)\right\|^2 + \mathbb{E}\left\|\nabla_A f_t(A_tB_t)\right\|^2\right)$$
$$+ \frac{1}{2}\left(LC_B^2 + L^2 C_A^2 C_B^2 + G^2\right)\mathbb{E}\left\|A_{t+1} - A_t\right\|^2 + \left(1 + \frac{LC_A^2}{2}\right)\mathbb{E}\left\|B_{t+1} - B_t\right\|^2$$
$$+ \eta E\,\mathbb{E}\left[\left\|\sum_k p_t^{(k)}\Delta_{B,t}^{(k)} - \nabla_B f_t(A_tB_t)\right\|^2\right] \tag{77}$$
$$+ \eta E\,\mathbb{E}\left[\left\|\sum_k p_t^{(k)}\Delta_{A,t}^{(k)} - \nabla_A f_t(A_tB_t)\right\|^2\right].$$

By using Lemma B.7 and Lemma B.8, we have

$$\mathbb{E}\left[f_t(A_{t+1}B_{t+1})\right] \le \mathbb{E}\left[f_t(A_tB_t)\right] - \frac{3\eta E}{4}\left(\mathbb{E}\left\|\nabla_B f_t(A_tB_t)\right\|^2 + \mathbb{E}\left\|\nabla_A f_t(A_tB_t)\right\|^2\right)$$
$$+ 2\eta^2(\sigma^2 + G^2)E^2\left\{\frac{C_B^2}{2}\left(LC_B^2 + L^2 C_A^2 C_B^2 + G^2\right) + C_A^2\left(1 + \frac{LC_A^2}{2}\right)\right\} \tag{78}$$
$$+ 4\eta^3 E(D_A + D_B).$$

Let $\widetilde{W}_t$ denote the model parameter at the start of round $t$. By construction,

$$f_t(A_tB_t) = f(\widetilde{W}_t), \quad f_t(A_{t+1}B_{t+1}) = f(\widetilde{W}_{t+1}),$$

since accumulation does not alter the initial parameter. Rearranging the bound obtained above then gives

$$\mathbb{E}\left[\left\|\nabla_A f(\widetilde{W}_t)\right\|^2 + \left\|\nabla_B f(\widetilde{W}_t)\right\|^2\right] \le \frac{4}{3\eta E}\left(\mathbb{E}\left[f(\widetilde{W}_t)\right] - \mathbb{E}\left[f(\widetilde{W}_{t+1})\right]\right) + K_1\eta + K_2\eta^2, \tag{79}$$

where

$$K_1 = \frac{8}{3}(\sigma^2 + G^2)E\left\{\frac{C_B^2}{2}\left(LC_B^2 + L^2 C_A^2 C_B^2 + G^2\right) + C_A^2\left(1 + \frac{LC_A^2}{2}\right)\right\},$$
$$K_2 = \frac{16}{3}(D_A + D_B).$$

Summing this inequality over $t = 0, \ldots, T - 1$ and dividing by $T$ yields

$$\frac{1}{T}\sum_{t=0}^{T-1}\mathbb{E}\left[\left\|\nabla_A f(\widetilde{W}_t)\right\|^2 + \left\|\nabla_B f(\widetilde{W}_t)\right\|^2\right] \le \frac{4}{3\eta TE}\left(\mathbb{E}\left[f(\widetilde{W}_0)\right] - \mathbb{E}\left[f(\widetilde{W}_T)\right]\right) + K_1\eta + K_2\eta^2. \tag{80}$$

Since $f$ is bounded below by $f^\star$, we have

$$\mathbb{E}\big[f(\widetilde{W}_T)\big] \geq f^\star,$$

so letting $\Delta_0 = f(W_0) - f^\star = \mathbb{E}[f(\widetilde{W}_0)] - f^\star$ yields

$$\frac{1}{T} \sum_{t=0}^{T-1} \mathbb{E}\big[\|\nabla_A f(\widetilde{W}_t)\|^2 + \|\nabla_B f(\widetilde{W}_t)\|^2\big] \;\leq\; \frac{4\,\Delta_0}{3\,\eta\,E\,T} \;+\; K_1\,\eta \;+\; K_2\,\eta^2. \tag{81}$$

Taking the minimum over $t = 0, \dots, T-1$ on the left and observing that each term is nonnegative gives the same upper bound for $\min_{t<T} \mathbb{E}[\|\nabla_A f(\widetilde{W}_t)\|^2 + \|\nabla_B f(\widetilde{W}_t)\|^2]$.

Finally, by choosing the classical diminishing stepsize $\eta = \eta_0\,T^{-1/2}$ for a constant $\eta_0 > 0$, each term on the right-hand side of (equation 81) scales as $\mathcal{O}(T^{-1/2})$. Hence

$$\min_{0 \leq t < T} \mathbb{E}\big[\|\nabla_A f(\widetilde{W}_t)\|^2 + \|\nabla_B f(\widetilde{W}_t)\|^2\big] = \mathcal{O}(T^{-1/2}),$$

which completes the proof of Theorem 4.2. $\qquad\square$

## C   Detail of the algorithms

In this section, we provide detailed explanation of fine-tuning version of FedLoRU and introduces variants of FedLoRU to adapt to environments with statistical and model heterogeneity by employing multiple or hierarchical low-rank updates.

### C.1   FedLoRU for Fine-Tuning

In the fine-tuning version of FedLoRU, the approach deliberately avoids merging the low-rank update matrices into the frozen pre-trained model. Instead, these low-rank matrices are stored separately, enabling a plug-and-play mechanism. This design choice allows the pre-trained model to remain intact while the task-specific adaptations are provided solely by the auxiliary low-rank matrices. As a result, this framework not only minimizes storage overhead and communication costs but also maintains flexibility during fine-tuning — clients can easily swap or update the low-rank components without altering the core model, ensuring efficient and adaptable federated learning.

### C.2   Personalized Federated Low-Rank Updates (pFedLoRU)

We develop the personalized FedLoRU (pFedLoRU) algorithm to address statistical heterogeneity (non-IID) in federated learning, building on the FedLoRU approach. The pFedLoRU algorithm enables each client $k$ to train a personalized model adapted to its data distribution.

In pFedLoRU, each client $k$ maintains a local copy of the global model $\boldsymbol{W}_t$, global low-rank matrices $\boldsymbol{A}_t^{(k)}$ and $\boldsymbol{B}_t^{(k)}$, and personal matrices $\boldsymbol{L}_t^{(k)}$ and $\boldsymbol{U}_t^{(k)}$. The matrices $\boldsymbol{A}_t^{(k)}$ and $\boldsymbol{B}_t^{(k)}$ are shared with the server to update the global model, while $\boldsymbol{L}_t^{(k)}$ and $\boldsymbol{U}_t^{(k)}$ are tailored to adapt to the local distribution. In each round $t$, client $k$ optimizes the personal matrices for $E_{\text{per}}$ epochs and the global matrices for $E_{\text{global}}$ by solving

$$\boldsymbol{L}_{t+1}^{(k)},\ \boldsymbol{U}_{t+1}^{(k)} = \underset{\boldsymbol{L},\,\boldsymbol{U}}{\arg\min}\, f^{(k)}(\boldsymbol{W}_t + \alpha_{\text{global}}\boldsymbol{A}_t\boldsymbol{B}_t + \alpha_{\text{per}}\boldsymbol{L}\boldsymbol{U}), \tag{82}$$

$$\boldsymbol{A}_{t+1}^{(k)},\ \boldsymbol{B}_{t+1}^{(k)} = \underset{\bar{\boldsymbol{A}},\,\bar{\boldsymbol{B}}}{\arg\min}\, f^{(k)}(\boldsymbol{W}_t + \alpha_{\text{global}}\bar{\boldsymbol{A}}\bar{\boldsymbol{B}} + \alpha_{\text{per}}\boldsymbol{L}_{t+1}^{(k)}\boldsymbol{U}_{t+1}^{(k)}). \tag{83}$$

Subsequently, the server collects the global update matrices $\boldsymbol{A}_{t+1}^{(k)}$ and $\boldsymbol{B}_{t+1}^{(k)}$ from the clients, performs aggregation $\boldsymbol{A}_{t+1} \leftarrow \sum_{k \in \mathcal{K}_t} p_t^{(k)}\boldsymbol{A}_{t+1}^{(k)}$, $\boldsymbol{B}_{t+1} \leftarrow \sum_{k \in \mathcal{K}_t} p_t^{(k)}\boldsymbol{B}_{t+1}^{(k)}$, and broadcasts $\boldsymbol{A}_{t+1}$ and $\boldsymbol{B}_{t+1}$ to the

---

**Algorithm 2** pFedLoRU.

---

**Require:** model $\boldsymbol{W}_0$, initial global low-rank update matrices $\boldsymbol{A}_0, \boldsymbol{B}_0$
**Require:** initial personal low-rank update matrices $\boldsymbol{L}_0, \boldsymbol{U}_0$
**Require:** scaling factors $\alpha_{\text{global}}$ and $\alpha_{\text{per}}$, accumulation cycle $\tau$, total round $T$
  **Initialize:** Server sends $\boldsymbol{W}_0$ to each client.
  **for** $t = 0, \cdots, T - 1$ **do**
    Server selects participating clients $\mathcal{K}_t$ and distributes $\boldsymbol{A}_t, \boldsymbol{B}_t$ to the clients in $\mathcal{K}_t$.
    **for** each client $k \in \mathcal{K}_t$ **do**
      **Local training:**
        Find $\boldsymbol{L}_{t+1}^{(k)}, \boldsymbol{U}_{t+1}^{(k)}$ by solving (82) starting from $\boldsymbol{L}_t^{(k)}, \boldsymbol{U}_t^{(k)}$.
        Find $\boldsymbol{A}_{t+1}^{(k)}, \boldsymbol{B}_{t+1}^{(k)}$ by solving (83) starting from $\boldsymbol{A}_t, \boldsymbol{B}_t$.
      Send $\boldsymbol{A}_{t+1}^{(k)}, \boldsymbol{B}_{t+1}^{(k)}$ to the server.
    **end for**
    **Server aggregation:** $\boldsymbol{A}_{t+1} \leftarrow \sum_{k \in \mathcal{K}_t} p_t^{(k)} \boldsymbol{A}_{t+1}^{(k)}$, $\boldsymbol{B}_{t+1} \leftarrow \sum_{k \in \mathcal{K}_t} p_t^{(k)} \boldsymbol{B}_{t+1}^{(k)}$.
    **if** $(t+1) \bmod \tau = 0$ **then**
      Server distributes $\boldsymbol{A}_{t+1}, \boldsymbol{B}_{t+1}$ to all clients.
      Each client $k$ updates its local copy of the global model: $\boldsymbol{W}_{t+1} \leftarrow \boldsymbol{W}_t + \alpha_{\text{global}} \boldsymbol{A}_{t+1} \boldsymbol{B}_{t+1}$.
      Server re-initializes $\boldsymbol{A}_{t+1}, \boldsymbol{B}_{t+1}$.
    **else**
      Server and clients update $\boldsymbol{W}_{t+1} \leftarrow \boldsymbol{W}_t$.
    **end if**
  **end for**
  **Return:** The final model for client $k$ is $\boldsymbol{W}_T + \alpha_{\text{per}} \boldsymbol{L}_T^{(k)} \boldsymbol{U}_T^{(k)}$.

---

clients. The clients then accumulate the low-rank updates accordingly as in FedLoRU. If client $k$ performs inference, it is based on model $\boldsymbol{W}_T + \alpha_{\text{per}} \boldsymbol{L}_T^{(k)} \boldsymbol{U}_T^{(k)}$.

In pFedLoRU, the communication between the server and clients involves only the low-rank matrices $\boldsymbol{A}^{(k)}$ and $\boldsymbol{B}^{(k)}$, which substantially reduces communication overhead. In practice, since the global model incorporates general knowledge from all clients' datasets, and the personalized model is essentially a fine-tuned version of the global model, we typically assign higher ranks to $\boldsymbol{A}^{(k)}$ and $\boldsymbol{B}^{(k)}$. Additionally, although we use the same rank for $\boldsymbol{L}^{(k)}$ and $\boldsymbol{U}^{(k)}$ across all clients in our experiments, each client can, in practice, use different ranks based on the complexity and size of their local dataset. It is also noteworthy that different ranks for $\boldsymbol{A}^{(k)}$ and $\boldsymbol{B}^{(k)}$ can be employed by integrating pFedLoRU and mFedLoRU.

### C.3 Model-Heterogeneous Federated Low-Rank Updates (mFedLoRU)

When local clients possess varying hardware resources, it becomes impractical to use uniform low-rank matrices across all clients. To address this issue, we develop the model-heterogeneous FedLoRU (mFedLoRU) algorithm, which employs hierarchical low-rank updates that allows clients to use their adaptive update ranks. In mFedLoRU, at each round $t$, each client $k$ receives $\boldsymbol{A}_t$ and $\boldsymbol{B}_t$ and updates its local copy of the global model as in FedLoRU. For local training, each client $k$ generates and optimizes nested low-rank matrices $\boldsymbol{A}_{\text{d}}^{(k)} \boldsymbol{A}_{\text{u}}^{(k)}$ and $\boldsymbol{B}_{\text{d}}^{(k)} \boldsymbol{B}_{\text{u}}^{(k)}$ by solving

$$\boldsymbol{A}_{\text{d}}^{(k)}, \boldsymbol{A}_{\text{u}}^{(k)}, \boldsymbol{B}_{\text{d}}^{(k)}, \boldsymbol{B}_{\text{u}}^{(k)} = \underset{\bar{\boldsymbol{A}}_{\text{d}}, \bar{\boldsymbol{A}}_{\text{u}}, \bar{\boldsymbol{B}}_{\text{d}}, \bar{\boldsymbol{B}}_{\text{u}}}{\arg\min} f^{(k)}(\boldsymbol{W}_t + \alpha(\boldsymbol{A}_t + \alpha_A^{(k)} \bar{\boldsymbol{A}}_{\text{d}} \bar{\boldsymbol{A}}_{\text{u}})(\boldsymbol{B}_t + \alpha_B^{(k)} \bar{\boldsymbol{B}}_{\text{d}} \bar{\boldsymbol{B}}_{\text{u}})). \tag{84}$$

Here, $\boldsymbol{A}_t \boldsymbol{B}_t$ are the rank-$r$ low-rank matrices, and $\boldsymbol{A}_{\text{d}}^{(k)} \boldsymbol{A}_{\text{u}}^{(k)}$ and $\boldsymbol{B}_{\text{d}}^{(k)} \boldsymbol{B}_{\text{u}}^{(k)}$ are rank-$r_A$ and rank-$r_B$ low-rank matrices used to update $\boldsymbol{A}_t$ and $\boldsymbol{B}_t$. After local training, the server collects $\boldsymbol{A}_{\text{d}}^{(k)}, \boldsymbol{A}_{\text{u}}^{(k)}$, recovers the low-rank update matrix $\boldsymbol{A}_{t+1}^{(k)} \leftarrow \boldsymbol{A}_t + \alpha_A^{(k)} \boldsymbol{A}_{\text{d}}^{(k)} \boldsymbol{A}_{\text{u}}^{(k)}$, and finally aggregates $\boldsymbol{A}_{t+1} \leftarrow \sum_{k \in \mathcal{K}_t} p_t^{(k)} \boldsymbol{A}_{t+1}^{(k)}$. The same process applies for the low-rank matrices $\boldsymbol{B}_{\text{d}}^{(k)}$ and $\boldsymbol{B}_{\text{u}}^{(k)}$.

---

**Algorithm 3** mFedLoRU.

---

**Require:** model $\boldsymbol{W}_0$, initial low-rank update matrices $\boldsymbol{A}_0, \boldsymbol{B}_0$
**Require:** scaling factors $\alpha, \alpha_A^{(k)}, \alpha_B^{(k)}$
**Require:** accumulation cycle $\tau$, total round $T$
  **Initialize:** Server sends $\boldsymbol{W}_0$ to each client.
  **for** $t = 0, \cdots, T-1$ **do**
      Server selects participating clients $\mathcal{K}_t$ and distributes $\boldsymbol{A}_t, \boldsymbol{B}_t$.
      **for** each client $k \in \mathcal{K}_t$ **do**
          Initializes nested low-rank updates $\boldsymbol{A}_{\mathrm{d}}^{(k)}, \boldsymbol{A}_{\mathrm{u}}^{(k)}$ and $\boldsymbol{B}_{\mathrm{d}}^{(k)}, \boldsymbol{B}_{\mathrm{u}}^{(k)}$.
          **Local training:**
              Find $\boldsymbol{A}_{\mathrm{d}}^{(k)}, \boldsymbol{A}_{\mathrm{u}}^{(k)}, \boldsymbol{B}_{\mathrm{d}}^{(k)}, \boldsymbol{B}_{\mathrm{u}}^{(k)}$ by solving (84)
              starting from $\boldsymbol{W}_t + \alpha(\boldsymbol{A}_t + \alpha_A^{(k)} \boldsymbol{A}_{\mathrm{d}}^{(k)} \boldsymbol{A}_{\mathrm{u}}^{(k)})(\boldsymbol{B}_t + \alpha_B^{(k)} \boldsymbol{B}_{\mathrm{d}}^{(k)} \boldsymbol{B}_{\mathrm{u}}^{(k)})$.
          Sends $\boldsymbol{A}_{\mathrm{d}}^{(k)}, \boldsymbol{A}_{\mathrm{u}}^{(k)}$ and $\boldsymbol{B}_{\mathrm{d}}^{(k)}, \boldsymbol{B}_{\mathrm{u}}^{(k)}$ to the server.
      **end for**
      Recover rank-$r$ low-rank updates from hierarchical low-rank updates:
          $\boldsymbol{A}_{t+1}^{(k)} \leftarrow \boldsymbol{A}_t + \alpha_A^{(k)} \boldsymbol{A}_{\mathrm{d}}^{(k)} \boldsymbol{A}_{\mathrm{u}}^{(k)}, \quad \boldsymbol{B}_{t+1}^{(k)} \leftarrow \boldsymbol{B}_t + \alpha_B^{(k)} \boldsymbol{B}_{\mathrm{d}}^{(k)} \boldsymbol{B}_{\mathrm{u}}^{(k)}$.
      **Server aggregation:** $\boldsymbol{A}_{t+1} \leftarrow \sum_{k \in \mathcal{K}_t} p_t^{(k)} \boldsymbol{A}_{t+1}^{(k)}, \boldsymbol{B}_{t+1} \leftarrow \sum_{k \in \mathcal{K}_t} p_t^{(k)} \boldsymbol{B}_{t+1}^{(k)}$.
      **if** $(t+1) \bmod \tau = 0$ **then**
          Server distributes $\boldsymbol{A}_{t+1}, \boldsymbol{B}_{t+1}$ to all clients.
          Each client $k$ updates its local model: $\boldsymbol{W}_{t+1} \leftarrow \boldsymbol{W}_t + \alpha \boldsymbol{A}_{t+1} \boldsymbol{B}_{t+1}$.
          Server re-initializes $\boldsymbol{A}_{t+1}, \boldsymbol{B}_{t+1}$.
      **else**
          Server and clients update $\boldsymbol{W}_{t+1} \leftarrow \boldsymbol{W}_t$.
      **end if**
  **end for**
  **Return:** The final model $\boldsymbol{W}_T = \boldsymbol{W}_0 + \alpha \sum_{t=0:\ (t+1) \bmod \tau = 0}^{T-1} \boldsymbol{A}_{t+1} \boldsymbol{B}_{t+1}$.

---

Model-heterogeneous FedLoRU (mFedLoRU) algorithm enables each client $k$ to utilize a rank tailored to its resource constraints. Similar to FedLoRU, client $k$ maintains low-rank update matrices $\boldsymbol{A}_t^{(k)} \in \mathbb{R}^{m \times r}$ and $\boldsymbol{B}_t^{(k)} \in \mathbb{R}^{r \times n}$, but each client $k$ decides whether to use nested low-rank updates or not. If a client opts out of nested low-rank updates, it updates its low-rank modules like in FedLoRU. However, if client $k$ chooses nested low-rank updates, it determines the locally adapted rank $r_A^{(k)}, r_B^{(k)} < r$ based on its resources. At each round, it initializes nested low-rank update matrices $\boldsymbol{A}_\mathrm{d}^{(k)} \in \mathbb{R}^{m \times r_A^{(k)}}$, $\boldsymbol{A}_\mathrm{u}^{(k)} \in \mathbb{R}^{r_A^{(k)} \times r}$ and $\boldsymbol{B}_\mathrm{d}^{(k)} \in \mathbb{R}^{r \times r_B^{(k)}}$, $\boldsymbol{B}_\mathrm{u}^{(k)} \in \mathbb{R}^{r_B^{(k)} \times n}$ such that $\boldsymbol{A}_\mathrm{d}^{(k)} \boldsymbol{A}_\mathrm{u}^{(k)} = 0$ and $\boldsymbol{B}_\mathrm{d}^{(k)} \boldsymbol{B}_\mathrm{u}^{(k)} = 0$. After local training by solving (84), we update client $k$'s original low-rank matrices as follows:

$$\boldsymbol{A}_{t+1}^{(k)} \leftarrow \boldsymbol{A}_t + \alpha_A^{(k)} \boldsymbol{A}_\mathrm{d}^{(k)} \boldsymbol{A}_\mathrm{u}^{(k)}, \quad \boldsymbol{B}_{t+1}^{(k)} \leftarrow \boldsymbol{B}_t + \alpha_B^{(k)} \boldsymbol{B}_\mathrm{d}^{(k)} \boldsymbol{B}_\mathrm{u}^{(k)}. \tag{85}$$

After local training, to reduce communication overhead, the client does not recover its original low-rank matrices directly. Instead, it sends the nested low-rank matrices to the server, which recovers them into rank-$r$ low-rank matrices $\boldsymbol{A}_{t+1}^{(k)} \leftarrow \boldsymbol{A}_t + \alpha_A^{(k)} \boldsymbol{A}_\mathrm{d}^{(k)} \boldsymbol{A}_\mathrm{u}^{(k)}$ and $\boldsymbol{B}_{t+1}^{(k)} \leftarrow \boldsymbol{B}_t + \alpha_B^{(k)} \boldsymbol{B}_\mathrm{d}^{(k)} \boldsymbol{B}_\mathrm{u}^{(k)}$, and then performs aggregation using these rank-$r$ low-rank matrices as in FedLoRU. By using this strategy, the communication overhead is reduced from $2mn$ to $r(m + n) + r_A(m + r) + r_B(n + r)$.

### C.4 Personalized Federated Low-Rank Adaptation (pFedLoRA)

We outline two variants of the personalized FedLoRA algorithm here. We use these algorithms to compare our pFedLoRU. Both versions of pFedLoRA follow a similar framework, where each client maintains a full-rank global model $\boldsymbol{W}$ and its own personalization models $\boldsymbol{L}^{(k)}$ and $\boldsymbol{U}^{(k)}$.

---

**Algorithm 4** pFedLoRA.

---

**Require:** model $\boldsymbol{W}_0$, initial personal low-rank update matrices $\boldsymbol{L}_0, \boldsymbol{U}_0$
**Require:** scaling factor $\alpha_\mathrm{per}$, total round $T$
  **Initialize:** Server sends $\boldsymbol{W}_0$ to each client.
  **for** $t = 0, \cdots, T - 1$ **do**
     Server selects participating clients $\mathcal{K}_t$ and distributes $\boldsymbol{W}_t$. Each client $k \in \mathcal{K}_t$
     initializes it as a local copy of the global model.
     **for** each client $k \in \mathcal{K}_t$ **do**
        **Local training - pFedLoRA(1):**
           Find $\boldsymbol{L}_{t+1}^{(k)}, \boldsymbol{U}_{t+1}^{(k)}$ by solving (86) starting from $\boldsymbol{L}_t^{(k)}, \boldsymbol{U}_t^{(k)}$.
           Find $\boldsymbol{W}_{t+1}^{(k)}$ by solving (87) starting from $\boldsymbol{W}_t$.
        **Local training - pFedLoRA(2):**
           Find $\boldsymbol{W}_{t+1}^{(k)}, \boldsymbol{L}_{t+1}^{(k)}, \boldsymbol{U}_{t+1}^{(k)}$ together by solving (88) starting from $\boldsymbol{W}_t$ and $\boldsymbol{L}_t^{(k)}, \boldsymbol{U}_t^{(k)}$.
        Send $\boldsymbol{W}_{t+1}^{(k)}$ to the server.
     **end for**
     **Server aggregation:** $\boldsymbol{W}_{t+1} \leftarrow \sum_{k \in \mathcal{K}_t} p_t^{(k)} \boldsymbol{W}_{t+1}^{(k)}$.
  **end for**
  **Return**: The final model for client $k$ is $\boldsymbol{W}_T + \alpha_\mathrm{per} \boldsymbol{L}_T^{(k)} \boldsymbol{U}_T^{(k)}$.

---

In pFedLoRA(1), the first variant, as suggested by Wu et al. (2024) and other FedLoRA algorithms, the personalization models are optimized separately from the global model. Specifically, the algorithm first optimizes the personalization models for $E_\mathrm{per}$ iterations and subsequently optimizes the global full-rank model for $E_\mathrm{global}$ iterations by solving:

$$\boldsymbol{L}_{t+1}^{(k)}, \boldsymbol{U}_{t+1}^{(k)} = \arg\min_{\boldsymbol{L}, \boldsymbol{U}} f^{(k)}(\boldsymbol{W}_t + \alpha_{\mathrm{per}} \boldsymbol{L} \boldsymbol{U}), \tag{86}$$

$$\boldsymbol{W}_{t+1}^{(k)} = \arg\min_{\boldsymbol{W}} f^{(k)}(\boldsymbol{W} + \alpha_{\mathrm{per}} \boldsymbol{L}_{t+1}^{(k)} \boldsymbol{U}_{t+1}^{(k)}). \tag{87}$$

However, pFedLoRA(1) has been found to be less effective compared to our modified version pFedLoRA(2). The second variant, pFedLoRA(2), optimizes both the personalization modules and the global full-rank model simultaneously for $E = E_{\mathrm{per}} + E_{\mathrm{global}}$ iterations by solving:

$$\boldsymbol{W}_{t+1}^{(k)}, \boldsymbol{L}_{t+1}^{(k)}, \boldsymbol{U}_{t+1}^{(k)} = \arg\min_{\boldsymbol{W}, \boldsymbol{L}, \boldsymbol{U}} f^{(k)}(\boldsymbol{W} + \alpha_{\mathrm{per}} \boldsymbol{L} \boldsymbol{U}). \tag{88}$$

## D  Detail of the experiment setting

In this section, we provide detailed experimental settings, including datasets, models, hyperparameters, and training protocols. We use PyTorch, CUDA 11.4, and 4 TITAN Xp GPUs. Additionally, we present experiments for pFedLoRU and mFedLoRU that are not included in the main text.

### D.1  Datasets and Models

The federated learning experiments use five datasets: Fashion-MNIST (FMNIST, Xiao et al. (2017)), CIFAR-10, CIFAR-100 (Krizhevsky & Hinton, 2009), Alpaca (Taori et al., 2023), and GLUE (Wang et al., 2018). The Alpaca dataset, consisting of 52,000 instruction and demonstration samples, was divided into 48,000 instances for training and 2,000 each for validation and test datasets in our fine-tuning experiment.

We construct datasets for clients by evenly splitting the training data among $K$ clients in a statistically homogeneous (i.e., IID) federated learning setting. For the heterogeneous statistical setting, we follow the procedure outlined in Hsu et al. (2019), which involves applying Latent Dirichlet Allocation (LDA) over the dataset labels to create clients' datasets. In this approach, each client is assigned a multinomial distribution over the labels, from which its examples are sampled. The multinomial distribution is drawn from a symmetric Dirichlet distribution with parameter $\psi$. For the non-IID setting, we use $\psi = 0.5$ to simulate a severely heterogeneous environment.

### D.2  Implementation and training details

**Detailed implementation of FedLoRA, FedLoRU, and FedHM**  In FedLoRA, FedLoRU, FedHM, and their variant algorithms, we apply low-rank factorization to the convolutional layers in ResNet-based models and to the self-attention modules in LLaMA2-3B and TinyLlama 1.1B. Specifically, for ResNet10 and ResNet18, we factorize the convolutional layers in layer1 through layer4, and for LLaMA2-3B and TinyLlama 1.1B, we factorize the self-attention modules in q_proj, k_proj, v_proj, and o_proj. We explore various low-rank configurations, setting the ranks of the factorized modules to 16, 32, 64, and 128 for FedLoRA and FedLoRU. We use rank $r = 128$ as the largest rank since our initial experiments showed it to have the best performance/memory trade-off. For FedHM, since its factorization scheme differs from that of FedLoRA and FedLoRU, we determine equivalent rank factors that yield the same number of trainable parameters as the ranks used in FedLoRA and FedLoRU.

We employ two strategies for initializing the low-rank update matrices in FedLoRU. For random initialization, as adopted in Hu et al. (2021), we initialize $\boldsymbol{A}$ with a random Kaiming distribution and set $\boldsymbol{B}$ to zero, ensuring that $\boldsymbol{AB}$ is zero at the start. The scheduling of accumulations is also critical due to the varying nature of the training phases across different rounds; in this study, we employ periodic accumulation with the accumulation cycle determined through a grid search over the values {20, 30, 40, 50, 60, 70, 80}, though this area warrants further investigation. We assess the performance by evaluating Top-1 test accuracy across

experiments. In the non-IID setting, due to significant fluctuations in performance, we report the average of the last five test accuracy values.

**Federated learning setting**  The federated learning experiments were conducted using five datasets: FMNIST, CIFAR-10, CIFAR-100, Alpaca, and GLUE. The client sampling rate, representing the proportion of clients selected per communication round, was set at 0.5 for all datasets except GLUE. For GLUE, we use full-participation. Each client performed 5 local epochs per communication round on the image datasets, while client performed 1 local epochs on Alpaca and GLUE.

For training FMNIST, CIFAR-10, and CIFAR-100, we utilized stochastic gradient descent (SGD) with a momentum of 0.9 as the local optimizer. The learning rate was selected through a grid search over 0.3, 0.2, 0.1, 0.05, 0.01, and a Cosine-Annealing learning rate scheduler was applied throughout the training process, with a minimum learning rate of 0.001 and a cycle step set to 50 or the total number of communication rounds. For fine-tuning LLaMA2-3B and TinyLlama 1.1B, we used AdamW (Loshchilov & Hutter, 2017) as the local optimizer, with a learning rate of $3 \times 10^{-4}$ and betas set to (0.9, 0.999), without employing a learning rate scheduler.

**Fine-tuning setting**  We assess the fine-tuning performance of FedLoRA and FedLoRU using two different ranks, 8 and 16. For the low-rank matrix factorization of LLaMA2-3B and TinyLlama 1.1B, we employ the PEFT library (Mangrulkar et al., 2022). In particular, for LLaMA2-3B, the percentage of trainable parameters is 0.124% for rank 8 and 0.248% for rank 16.

**Personalization and model heterogeneous setting**  We compare pFedLoRU against pFedLoRA (Wu et al., 2024), and for mFedLoRU, we compare with the model-heterogeneous version of FedHM. For the model heterogeneous setting, we simulate virtual environments where each client is assigned a different nominal rank, thereby restricting them to use low-rank update matrices of varying ranks. In particular, we tested two different model heterogeneous configurations in mFedLoRU experiments where the clients had different ranks, denoted as $r$, which reflect the computational resources or constraints of each client. For FedHM, we match the number of trainable parameters corresponding to the model with specific rank in mFedLoRU experiments.

Table 3: Detailed model heterogeneous settings in our experiments. Both settings include total 20 clients.

| Rank of a client | | $r = 128$ | $r = 64$ | $r = 32$ | $r = 16$ |
|---|---|---|---|---|---|
| **#Clients** | setting 1 | 5 | 5 | 5 | 5 |
| | setting 2 | - | 6 | 6 | 7 |

The motivation behind these settings is to establish a challenging model heterogeneous environment. This is particularly important as we observed that FedLoRU with $r = 128$ produces similar results to FedAvg with a full-rank model. Therefore, these configurations were designed to test the algorithm's adaptability under more demanding and diverse client conditions. In addition, we set $\alpha_A$ and $\alpha_B$ to satisfy $\alpha_A/r_A = \alpha_A/r_B = 1/2$, as our empirical observations indicate that the choice of $\alpha$ values in the range of $1/4$ to $1$ has minimal effect on overall performance.

### D.3  Detail of the estimated stable rank experiment

We conduct an experiment to support our theoretical analysis that the Hessians of loss functions trained on smaller datasets exhibit larger stable ranks. In this experiment, we randomly select either 50 or 500 samples from the CIFAR-100 dataset and train a ResNet-18 model using only these 50 or 500 samples. Every 5 epochs, we compute an estimated stable rank of the Hessian, as calculating the true stable rank is computationally challenging due to the need to determine all singular values. Instead, we estimate the empirical spectral density using pyHessian (Yao et al., 2020), which provides the empirical singular values $\sigma_i(H)$ of a Hessian $H$ and their corresponding densities $p(\sigma_i)$, $i = 1, \cdots, Q$. Based on this, we calculate the estimated stable rank as follows:

$$\hat{\text{srank}}(H) = \frac{\sum_{i=1}^{Q} p(\sigma_i)\, \sigma_i^2(H)}{p(\sigma_1)\, \sigma_1^2(H)} \tag{89}$$

Figure 1 shows the results of the experiment, demonstrating that the Hessians trained on the smaller dataset ($n = 50$) consistently exhibits higher estimated stable ranks compared to those trained on the larger dataset ($n = 500$).

# E    Experiment Result for pFedLoRU and mFedLoRU

We evaluate the performance of pFedLoRU and mFedLoRU on statistical heterogeneous and model heterogeneous FL environments. Table 4 shows the performance of pFedLoRU and pFedLoRA. We use two variants of pFedLoRA, each utilizing different optimization schemes. For a comprehensive description of pFedLoRA(1) and pFedLoRA(2), see Appendix C.4. Under both non-IID levels ($\psi = 0.1$ and $\psi = 0.5$), pFedLoRU shows a clear advantage in terms of accuracy compared to pFedLoRA. In addition, despite having less than half the number of parameters, pFedLoRU consistently achieves higher accuracy.

Table 4: Comparison of the average test accuracy across local models for pFedLoRA and pFedLoRU with varying non-IIDness ($\psi$) on CIFAR100.

| Algorithm | #params | Non-IIDness | |
|---|---|---|---|
| | | $\psi = 0.1$ | $\psi = 0.5$ |
| pFedLoRA(1) | 11.22M | 45.36 | 42.14 |
| pFedLoRA(2) | 11.22M | 47.45 | 42.28 |
| pFedLoRU | 4.63M | **49.65** | **46.50** |

On the other hand, Table 5 shows the performance of mFedLoRU and FedHM. FedHM outperforms mFedLoRU in both heterogeneous settings (setting 1 and setting 2) on the CIFAR-10 dataset, indicating that FedHM handles model heterogeneity more effectively for simpler tasks. This suggests that FedHM is better suited for less complex datasets such as CIFAR-10, where its approach proves more efficient. However, mFedLoRU outperforms FedHM in both heterogeneous settings for the more complex CIFAR-100 dataset, demonstrating its potential in addressing the model-heterogeneous problem in federated learning. A key advantage of mFedLoRU is that it does not require additional computational steps, such as the weight factorization used in FedHM, making it a more efficient solution in scenarios involving more challenging tasks.

Table 5: Comparison of test accuracy for FedHM and mFedLoRU in two model-heterogeneous settings.

| Dataset | Setting | FedHM | mFedLoRU |
|---|---|---|---|
| CIFAR-10 | setting 1 | **88.09** | 84.81 |
| | setting 2 | **88.68** | 84.36 |
| CIFAR-100 | setting 1 | 49.84 | **51.16** |
| | setting 2 | 50.52 | **50.89** |

# F    Further Discussion on Experimental Results

In this section, we present learning curve plots and additional experimental results that were not included in the main text. Furthermore, we provide a more detailed analysis and discussion of the experimental outcomes.

## F.1    Experiment Results for FedAvg

To emphasize the comparison between FedLoRU and other communication-efficient federated learning algorithms, we have excluded the FedAvg results from the main text. The FedAvg outcomes are instead provided in Table 6.

From Table 1 and Table 6, we observe that FedAvg consistently performs well across different datasets and settings, but its performance tends to drop as the number of clients increases and in non-IID scenarios. For example, in the CIFAR-100 dataset under the IID setting with 100 clients, FedAvg achieves a test accuracy of

Table 6: Top-1 test accuracy of FedAvg under different federated learning settings and datasets

| Dataset | | FMNIST | CIFAR-10 | CIFAR-100 |
|---|---|---|---|---|
| | IID - K=20 | 91.81 | 93.48 | 69.97 |
| FL setting | IID - K=100 | 90.19 | 85.14 | 55.14 |
| | NonIID - K=20 | 80.03 | 79.65 | 19.18 |

55.14%, while its accuracy drops significantly to 19.18% in the non-IID setting with 20 clients. This illustrates FedAvg's limitations in handling large client numbers and heterogeneous data distributions.

In comparison, FedLoRU demonstrates competitive performance relative to FedAvg. While FedLoRU is at most 5% less accurate than FedAvg in some cases, it sometimes outperforms FedAvg, particularly in scenarios with a larger number of clients. For instance, in the CIFAR-100 IID setting with 100 clients, FedLoRU achieves a test accuracy of 57.96%, which surpasses FedAvg's accuracy of 55.14%. This suggests that FedLoRU's low-rank update approach scales better with an increasing number of clients and is more robust in large-scale federated learning environments.

## F.2 Learning Curve Plots For IID Setting

We present the test accuracy curves for experiments conducted under a statistically homogeneous setting. Figure 5, Figure 6 and Figure 7 shows the test accuracy with respect to communication round under the IID setting. The fluctuations observed in the graphs are attributable to the use of a cosine-annealing learning rate scheduler.

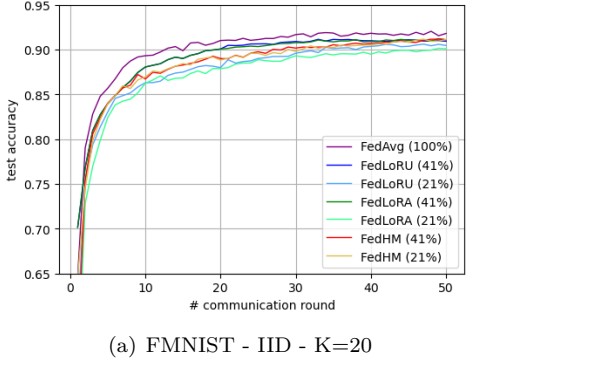
(a) FMNIST - IID - K=20

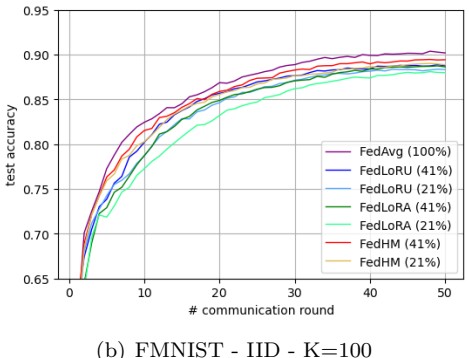
(b) FMNIST - IID - K=100

Figure 5: The test accuracy curves for FMNIST under an IID setting with K=20 and K=100.

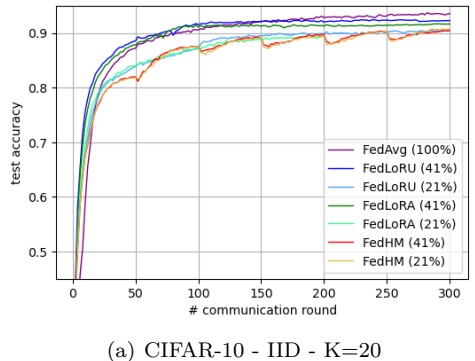
(a) CIFAR-10 - IID - K=20

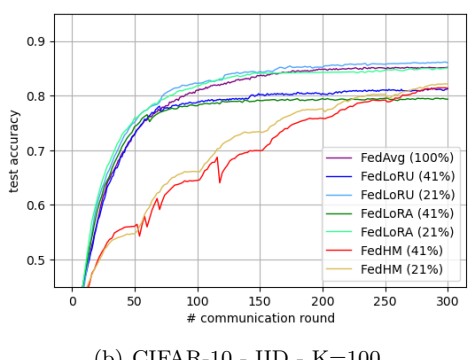
(b) CIFAR-10 - IID - K=100

Figure 6: The test accuracy curves for CIFAR-10 under an IID setting with K=20 and K=100.

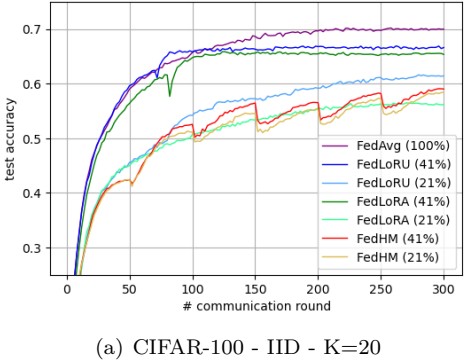
(a) CIFAR-100 - IID - K=20

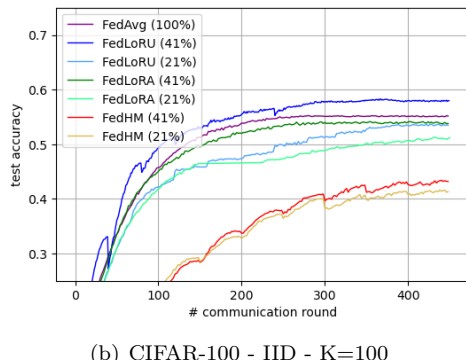
(b) CIFAR-100 - IID - K=100

Figure 7: The test accuracy curves for CIFAR-100 under an IID setting with K=20 and K=100.

### F.3    Discussion on Communication Cost

One of the main motivation of FedLoRU is to reduce the communication cost by using low-rank updates while maintaining reasonable performances. When the original weight matrix $\boldsymbol{W} \in \mathbb{R}^{m \times n}$ requires $mn$ parameters to be communicated, FedLoRU with rank $r$ requires $r(m + n)$ parameters. Additionally, as we can see in Figure 5, Figure 6 and Figure 7, the convergence speed is similar to FedAvg, resulting in much lower communication overheads.

Building on the motivation to reduce communication costs, Figure 3 compares the communication overheads across several federated learning algorithms—FedAvg, FedHM, FedLoRA, and FedLoRU—using the CIFAR-10 and CIFAR-100 datasets. The figure evaluates the communication cost in gigabytes (GB) required to reach specific target test accuracy (denoted as $T\%$) for different numbers of clients ($K$) and datasets. We compute the communication cost as $2 \times (\#\text{clients}) \times (\text{participation rate}) \times (\#\text{parameters}) \times (\text{parameter memory size}) \times (\#\text{round})$. It is evident that FedLoRU consistently achieves significantly lower communication costs compared to the other methods.

### F.4    Relative difference in performance in terms of the number of clients

Table 7 presents a comparison of test accuracy between FedAvg, FedLoRA, and FedLoRU across varying number of clients, illustrating the relative performance of these algorithms as the number of clients increases. FedLoRU consistently outperforms FedAvg when the number of clients exceeds 100, demonstrating its scalability and effectiveness in cross-device federated learning environments. Interestingly, even FedLoRA, which does not accumulate low-rank updates as in FedLoRU, outperforms FedAvg, particularly when the number of clients reaches 200 and above. This result suggests that simply adopting low-rank updates in high-client FL can significantly improve performance. These findings align with our theoretical insights, highlighting the potential benefits of leveraging low-rank structures in federated learning, even without the accumulation strategy employed by FedLoRU.

We extended our experiments to settings with a lower participation ratio and a larger number of clients. Specifically, we examined $K = 100, 200$ with $C = 0.1$, using an IID CIFAR-100 dataset, which is more challenging than FMNIST and CIFAR-10. For these tests, we used the ResNet18 model, applying full parameter training for FedAvg and 41% parameter training for low-rank methods. The results, averaged over three runs with minimal standard deviation ($< 0.005$), are presented in Table 8.

The results indicate that low-rank training methods consistently outperform full-rank training when the participation ratio is low and the number of clients increases. Among the low-rank approaches, FedLoRU achieves the highest accuracy, demonstrating its effectiveness in large-scale federated learning. These findings reinforce the advantages of using low-rank updates, particularly in settings with a large number of clients and limited participation per round.

Table 7: A comparison between FedAvg, FedLoRA, and FedLoRU accuracy across varying client numbers. The ratio is the relative difference in accuracy between two algorithms. Here, we compute the ratio of FedLoRA and FedLoRU compared to FedAvg. For example, ratio of FedLoRU is defined as $\text{Ratio} = \frac{\text{FedLoRU} - \text{FedAvg}}{\text{FedLoRU}}$.

|  |  | FedLoRA | | FedLoRU | |
|---|---|---|---|---|---|
| #Clients | FedAvg | acc | ratio | acc | ratio |
| 20 | 69.97 | 65.53 | -0.063 | 66.81 | -0.046 |
| 50 | 64.68 | 59.87 | -0.074 | 62.45 | -0.034 |
| 100 | 55.14 | 53.79 | -0.024 | 57.96 | +0.051 |
| 200 | 38.85 | 42.42 | +0.092 | 44.85 | +0.154 |
| 300 | 24.94 | 32.69 | +0.311 | 36.79 | +0.475 |
| 400 | 21.44 | 31.41 | +0.465 | 35.86 | +0.673 |

Table 8: A comparison between FedAvg, FedHM, FedLoRA, and FedLoRU accuracy for experiments under large client numbers $K = 100, 200$ with lower participation ratio $C = 0.1$.

|  | FedAvg | FedHM | FedLoRA | FedLoRU |
|---|---|---|---|---|
| K=100 | 0.5382 | 0.5732 | 0.5506 | **0.5837** |
| K=200 | 0.3885 | 0.4872 | 0.5227 | **0.5393** |

### F.5 Model alignment of FedLoRU

Theorem 3.2 shows that clients exhibit higher stable ranks, indicating a more complex loss landscape that exacerbates client discrepancies. Further, Theorem 3.3 demonstrates that low-rank approximations of client gradients are more closely aligned compared to higher-rank approximations. This behavior implies that constraining updates to a low-rank space, as implemented in FedLoRU, inherently regularizes client training by aligning updates along major directions and reducing variations between clients.

To empirically validate the alignment of client updates with global updates, we conducted experiments to calculate the average cosine similarity between the global update (difference between the aggregated global model and the previous global model) and the local updates (differences between the locally trained models and the previous global model). These experiments were conducted on CIFAR-100 in two configurations: (1) 20 clients with a participation rate of 0.5 and (2) 100 clients with a participation rate of 0.1, both in iid setting. The average cosine similarity across clients serves as a proxy for the degree of alignment, with higher values indicating stronger alignment between local and global updates.

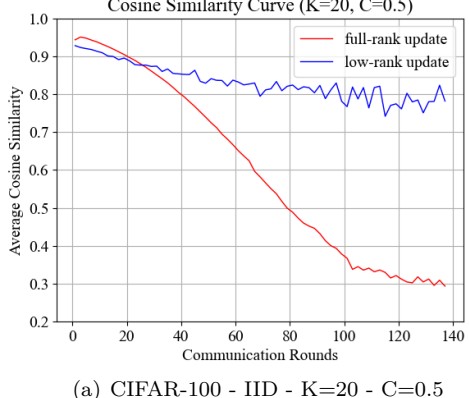
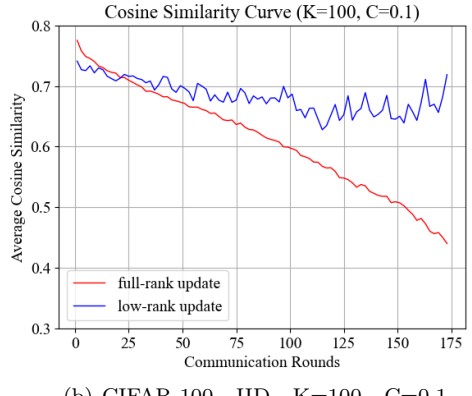

(a) CIFAR-100 - IID - K=20 - C=0.5     (b) CIFAR-100 - IID - K=100 - C=0.1

Figure 8: Average cosine similarity between global updates and local updates was calculated. Model weights were vectorized, and the cosine similarity between each participating client's update and the global update was computed.

In the first configuration with 20 clients, full-rank updates (FedAvg) initially exhibit higher cosine similarity, indicating stronger alignment with the global update in the early training stages. As training progresses, the alignment for both full-rank and low-rank updates decreases. Notably, after approximately 20 communication rounds, the low-rank updates consistently achieve higher cosine similarity than full-rank updates. This observation suggests that while low-rank updates initially align less closely with global updates due to their constrained nature, they adapt over time, improving alignment and maintaining stronger consistency during later communication rounds.

In the second configuration with 100 clients, a similar trend is observed. Full-rank updates initially achieve higher cosine similarity, reflecting better alignment in the early training stages. However, as training proceeds, low-rank updates surpass full-rank updates in alignment. The slightly lower cosine similarity of low-rank updates in the early stages likely reflects the initial adaptation of client updates within the constraints of the low-rank subspace.

## G   Additional Experimental Results

This section presents additional results that complement the main body. We include additional baselines including FedProx (Li et al., 2020) and FedEx-LoRA (Singhal et al., 2024), stress tests with $E = 1$, sensitivity analysis over accumulation cycles, experiments combining our approach with sparsification using FedPAQ (Reisizadeh et al., 2020), and an empirical analysis of Hessian stable rank under non-IID splits. Unless otherwise specified, we report test accuracy. Experiment settings are same as the main body, and hyperparameters are chosen through a grid search.

### G.1   Additional Baselines

Table 9 compares full-rank (FedAvg, FedProx) with low-rank counterparts (FedAvg+FedLoRU, Fed-Prox+FedLoRU) and FedEx-LoRA. FedLoRU is competitive with full-rank methods and, consistent with the main text, surpasses FedAvg as $K$ grows. We include FedEx-LoRA because it is one of the state-of-the-art methods that applies LoRA-style adaptation in federated fine-tuning. While its design is well suited to fine-tuning, our results show that FedLoRU surpasses FedEx-LoRA in pre-training scenarios. FedEx-LoRA exhibits substantially weaker performance across all settings in Table 9.

Table 9: Additional baselines. All low-rank methods use the same rank $r$.

| Algorithm | Setting | CIFAR10 | | | CIFAR100 | | |
|---|---|---|---|---|---|---|---|
| | | K=20 | K=100 | K=20 Diri | K=20 | K=100 | K=20 Diri |
| FedAvg | Full-Gradient | 93.48 | 85.14 | 79.65 | 69.97 | 55.14 | 19.18 |
| FedAvg+FedLoRU | Low-Rank | 92.43 | 81.46 | 75.19 | 66.81 | 57.96 | 16.46 |
| FedProx | Full-Gradient | 92.87 | 83.72 | 79.37 | 69.85 | 52.19 | 26.13 |
| FedProx+FedLoRU | Low-Rank | 91.93 | 84.51 | 75.11 | 69.45 | 53.48 | 23.14 |
| FedEx-LoRA | Low-Rank | 75.36 | 63.54 | 58.14 | 33.58 | 22.41 | 12.31 |

### G.2   Stress Test with $E = 1$

To examine the robustness of the algorithms under more frequent communications, we conduct experiments with a single local epoch ($E$=1). We consider two challenging configurations: (i) a large-scale, limited-participation IID setting (K100-C01-iid: $K$=100 clients, participation $C$=0.1), and (ii) a smaller, statistically heterogeneous non-IID setting (K20-C02-noniid: $K$=20, $C$=0.2, Dirichlet $\psi$=0.5). For each method, we perform a grid search over the learning rate $\eta$ and accumulation cycle $\tau$, and then evaluate the best configuration with a single run.

Even with only one local epoch, FedLoRU continues to outperform FedAvg and FedLoRA in the large-scale IID setting, and remains competitive in the non-IID case. These results indicate that the advantages of low-rank accumulation persist under communication-constrained training regimes.

Table 10: Test accuracy with $E=1$ local epoch.

| Setting | FedAvg | FedLoRU | FedLoRA | FedHM |
|---|---|---|---|---|
| **K100-C01-iid** | 45.43 | **48.03** | 46.36 | 28.70 |
| **K20-C02-noniid** | 21.55 | 23.19 | **23.45** | 15.30 |

### G.3 Sensitivity to the Accumulation Cycle $\tau$

We sweep $\tau \in \{20, 30, 40, 50, 60, 70, 80\}$ for $E=5$, $C=0.5$ across several $K$. Accuracy increases to an optimum and then declines as $\tau$ increases.

Table 11: Accumulation cycle sweep for $E=5$, $C=0.5$. Best values in bold.

| Setting | 20 | 30 | 40 | 50 | 60 | 70 | 80 |
|---|---|---|---|---|---|---|---|
| **E5-K20-C05** | 60.85 | 64.37 | 66.21 | **66.81** | 66.54 | 66.25 | 65.84 |
| **E5-K100-C05** | 52.78 | 55.76 | 57.85 | **57.96** | 56.92 | 55.86 | 55.45 |
| **E5-K200-C05** | 42.98 | **44.85** | 44.45 | 44.06 | 43.95 | 43.79 | 43.58 |
| **E5-K400-C05** | 35.25 | **35.86** | 34.88 | 33.41 | 32.58 | 32.21 | 31.98 |

Table 12: FedLoRU composed with FedPAQ (quantization).

| Dataset / Partition | Full bit | 16-bit | 12-bit | 8-bit |
|---|---|---|---|---|
| CIFAR10   (K=20, IID) | 92.43 | 92.39 | 92.28 | 92.33 |
| CIFAR10   (K=100, IID) | 81.46 | 81.48 | 81.42 | 81.31 |
| CIFAR10   (K=20, non-IID) | 75.19 | 75.38 | 75.19 | 74.86 |
| CIFAR100 (K=20, IID) | 66.81 | 66.79 | 67.05 | 66.58 |
| CIFAR100 (K=100, IID) | 57.96 | 57.95 | 57.75 | 57.88 |
| CIFAR100 (K=20, non-IID) | 16.46 | 16.23 | 15.49 | 15.29 |

### G.4 Compatibility with Sparsification (FedPAQ)

One of our central claims is that FedLoRU can be readily combined with other communication-efficient techniques. To support this, we integrate FedLoRU with FedPAQ (Reisizadeh et al., 2020), a state-of-the-art sparsification and quantization method in FL. In this experiment, we apply FedLoRU with a 41% parameter ratio and then vary the quantization bit-widths used in FedPAQ.

The results in Table 12 show that test accuracy remains stable across bit-widths, with negligible performance degradation even at 8-bit quantization. This demonstrates that FedLoRU complements existing compression approaches and that its benefits are orthogonal to sparsification, enabling further communication savings without sacrificing accuracy.

