# OpenReview forum: "Communication-Efficient Federated Low-Rank Update Algorithm and its Connection to Implicit Regularization"
_TMLR — Under review for TMLR_

### Review · Reviewer_9eT6 · 2026-06-09

**Summary Of Contributions:**

The paper proposes FedLoRU, a FL algorithm that constrains client-side updates to a low rank subspace and periodically accumulates these updates server side to recover higher-rank expressiveness. The paper also provides a theoretical motivation based on stable rank differences between client and server Hessians and argues that low rank updates improve inter client gradient alignment.

**Additional Comments:**

N/A

**Audience:**

Yes

**Audience Explanation:**

The topic is relevant to TMLR because it connects FL, low rank adaptation, communication efficiency, and optimization theory which would be interest of the TMLR audience working on efficient training and federated optimization.

**Claims And Evidence:**

No

**Claims Explanation:**

The theoretical motivation is interesting, I liked this angle of analyzing the heterogeneity through low rank updates in FL but the connection to practical FL is not fully convincing. The paper itself acknowledges that the analysis is based on a homogeneous setting where a client dataset is a random sub-sample of the full dataset. Therefore, the theory supports a limited data effect more strongly than a true statistical heterogeneity effect.


- The “convergence matches FedAvg” claim is too strong. The theorem shows an (O(T^{-1/2})) rate in the low-rank norm, not a full FedAvg equivalent guarantee in the original model parameter space.
- The gradient alignment theorem seems unclear. It analyzes idealized rank r gradient projections onto Hessian eigenspaces, but FedLoRU does not compute or explicitly optimize in those eigenspaces. The claimed link among the analyzed eigenspace projection and the learned (A,B) factor space is understandable, but remains heuristic.
- $$\left(\sum_k p_k A_k\right)\left(\sum_k p_k B_k\right) \neq \sum_k p_k A_k B_k$$ The paper neither proves this is good nor bad,  it simply reports that separate averaging works well empirically and defers analysis. The paper acknowledges that alternative schemes may have different mathematical justifications, but the current justification is not fully satisfactory.
- The claim of being the first theoretical study of rank properties in FL should be softened. \textbf{Possible Fix}: one of the first random-matrix style analyses of client/server Hessian stable rank differences for motivating low rank FL updates.
- The experiments are decent but not fully to the mark. I know that the more related work is LoRA but since few claims are regarding the communication efficiency it would be very good and strong for the paper to add baselines from other communication efficient FL algos like compression/sparsification/ beacause the claim of superiority over SOTA communication efficient FL is stronger than what the experiments establish.
- The dataset split details like $\beta$=0.5 should be included in the main paper than the appendix and an additional $\beta$=0.3 can be useful for stronger analysis.
- The “client Hessian is higher-rank than server Hessian” claim is valid under the paper’s random perturbation model, but maybe not be presented as a general property of FL.
- The renitialization step after accumulation needs to be more precisely specifie. What exactly is reset, and to what?

**Requested Changes:**

Please check above

---

### Review · Reviewer_7YY2 · 2026-06-25

**Summary Of Contributions:**

This paper focuses on improving the communication efficiency of federated learning. The authors propose a novel method named FedLoRU which leverages low rank structures in the gradient updates. The authors analyze the \emph{stable rank} of gradient updates and discovered that the stable rank of Hessians over each client's local dataset tend to be higher than the global Hessian over the full dataset.

**Audience:**

Yes

**Audience Explanation:**

This paper should be interesting to researchers working on federated learning and optimization.

**Broader Impact Concerns:**

The authors can briefly discuss the privacy concerns in the algorithm, as stated in the ``Requested Changes section''. Apart from this, there are no other major ethical concerns.

**Claims And Evidence:**

Yes

**Claims Explanation:**

1. The authors provide theoretical analysis to justify the insight regarding the rank structure of local client gradients.
2. The authors prove convergence rate for the proposed algorithm under standard Lipschitz and gradient norm assumptions, matching that of the standard FedAvg algorithm.
3. Extensive experiments are performed to demonstrate the performance and communication advantage over standard FedAvg and prior works that leverage low rank structure of gradients. These experiments range from small scale datasets and models, to larger scale experiments over open-source large language models.

**Requested Changes:**

Major comment:
1. One main motivation for federated learning is to guarantee privacy. However, extensive studies have shown that local gradient updates can be used to recover clients' data even though without directly sending the original data to the server. The authors should briefly discuss how privacy can be guaranteed given the current approach. One potential approach might be adding suitable noise that guarantees differential privacy. Adding a brief discussion will strengthen the paper.
2. The connection between the proposed FedLoRU algorithm and the insights described in Section 3 is unclear. In particular, it is unclear which parts of the FedLoRU algortihm are inspired by which part of the gradient structure analysis. It would be helpful to clearly state the connections. It would be even better if the authors could provide a simple toy example where FedLoRU succeeds but prior methods such as FedLoRA would fail. Adding these discussions would greatly strengthen the paper.
3. I assume that the rank $r$ should be an important hyperparameter in the training process, but it is not clear how this is chosen in the experiments. The authors should discuss the effect of rank and how to choose it appropriately in practice.


Minor comments and typos:
1. Add a short remark in Section 3.1 on how the singular values are sorted (I assume $\sigma_1\ge...\ge \sigma_{n}\ge 0$).
2. $m$ and $n$ are not defined in the beginning of Section 4.1.
3. Specify in the tables that FedLoRU is the new approach proposed in this work.
4. p.8 second paragraph, line 5, extra space after the word ``optimization''.

---

### Review · Reviewer_FGWk · 2026-07-12

**Summary Of Contributions:**

This paper studies low-rank client updates as a mechanism for both communication efficiency and implicit regularization in federated learning. The authors first analyze a finite-rank-plus-random-noise model of client and server Hessians, arguing that client Hessians have higher stable rank when clients have fewer samples. They then claim that lower-rank approximations of local gradients exhibit stronger cross-client alignment.

Motivated by these results, the paper proposes FedLoRU, where clients optimize low-rank factors $A,B$, the server aggregates these factors, and the resulting updates are periodically accumulated into the global model to recover higher-rank expressiveness. The paper also presents personalized and model-heterogeneous variants, a convergence analysis, and experiments on image classification and language-model fine-tuning.

I find the problem important and the algorithmic idea potentially useful. In particular, periodically accumulating low-rank updates is a natural way to avoid the expressiveness limitation of a single fixed-rank adapter. The experiments also suggest that FedLoRU can be competitive with full-rank training and may become relatively more effective when clients have limited local data.

**Audience:**

Yes

**Audience Explanation:**

Communication-efficient federated learning and low-rank adaptation are both relevant to the TMLR audience. The empirical observation that low-rank client optimization may become more useful when local datasets are small is interesting, and the periodic accumulation mechanism is practically motivated. Even independently of the full theoretical interpretation, the algorithm and its behavior in large-client experiments are worth investigating.

**Broader Impact Concerns:**

No such concerns.

**Claims And Evidence:**

Yes

**Claims Explanation:**

I would say partially, the claims are supported by evidence. The empirical evidence for the practical usefulness of FedLoRU is promising. However, I am not yet fully convinced by several parts of the theoretical argument, and some experiments do not directly measure the quantities appearing in the theory.

### Stable-rank analysis

I may be missing a detail in the asymptotic argument, but there appears to be a possible issue in the treatment of the Wigner bulk. Proposition 3.1 seems to establish convergence to the spectral edges for fixed indices near the top or bottom of the spectrum. It is less clear to me that this implies that all $O(R)$ interior eigenvalues can be replaced by the edge values, as appears to be done in Equation (35). For stable rank, I would instead expect the bulk contribution to involve the second moment of the limiting spectral distribution.

It would therefore be helpful for the authors to clarify how Proposition 3.1 justifies the summation over the entire bulk in Theorem 3.2. The assumed scaling of the Hessian perturbation with dataset size also deserves more discussion. The paper gives $s_N=1/N$ as a typical example, whereas fluctuations of empirical averages often scale closer to $N^{-1/2}$, depending on the assumptions.

### Gradient-alignment theorem

I also have some uncertainty about the proof of Theorem 3.3. In Equations (50)--(53), the expected cosine similarity seems to be reduced to a ratio involving expected numerator and denominator terms. Since cosine similarity has a random normalization, independence between clients alone may not be sufficient for this step; some concentration or inverse-moment argument may be needed.

Relatedly, squared gradient norms generally involve second moments of directional derivatives rather than squared first moments. There may also be a sign-convention issue because eigenvectors are only defined up to sign, whereas the preceding random-matrix result controls squared absolute overlap. These concerns may be resolvable with additional assumptions, but I think they should be stated explicitly. The strict positivity claim may also require non-degeneracy conditions.

### Connection between the theory and FedLoRU

The theoretical approximation projects a full parameter-space gradient onto leading Hessian eigendirections. FedLoRU instead imposes a layerwise low-matrix-rank parameterization $AB$. These are intuitively related, but they are not obviously the same notion of rank. A rank-$r$ matrix update need not lie in the span of the leading $r$ Hessian eigenvectors.

The empirical alignment experiment is useful, but it measures similarity between each local update and the aggregated global update. The theorem instead concerns pairwise similarity between clients' truncated gradients. Since the global update is constructed from the local updates, this metric also contains a self-correlation component. Direct pairwise client similarities, or measurements of how much leading-Hessian-eigenspace energy is captured by FedLoRU updates, would more directly support the proposed mechanism.

### Convergence and factor aggregation

The convergence theorem controls gradients with respect to \(A\) and \(B\), rather than the gradient with respect to the effective model $W$. Because bilinear factorization has scaling symmetries and potentially degenerate stationary points, I think the comparison to FedAvg should be qualified as convergence in the reparameterized factor variables unless a relationship to model-space stationarity is established.

I would also appreciate clarification on partial participation. In several proof steps, the global gradient appears to be identified with a weighted sum over participating clients. An explicit sampling model or unbiasedness argument seems necessary if the theorem is meant to apply to the full global objective.

Finally, separately averaging $A$ and $B$ gives $\left(\sum_k p_k A_k\right)\left(\sum_k p_k B_k\right)$, which is generally different from $\sum_k p_k A_kB_k$. The factorization is also non-identifiable under $A_kB_k=(A_kQ)(Q^{-1}B_k)$. Common initialization may help in practice, but an ablation against product averaging followed by recompression, factor alignment, or freezing one factor would make this design choice more convincing.

### Empirical scope and communication accounting

The experiments show that low-rank methods become relatively stronger as the number of clients increases. However, because the total dataset is fixed, increasing the client count also decreases the number of samples per client. Thus, the experiments do not distinguish an effect of client population size from the proposed small-local-dataset regularization effect. It would be useful to vary client count and local sample size independently.

Similarly, the stable-rank experiment trains separate models on 50 and 500 samples, so the Hessians are evaluated at different learned parameters. Evaluating different sample subsets at common checkpoints would more directly test the theoretical statement.

The communication results are encouraging, but the accounting should clarify whether the all-client broadcasts at accumulation rounds are included. The main tables would also benefit from reporting standard deviations directly. Some claims regarding robustness to non-IID data and model heterogeneity should be moderated, since FedLoRU does not uniformly outperform the strongest baselines in those settings.

**Requested Changes:**

1. Clarify or revise the proof of Theorem 3.2, particularly the contribution of the interior Wigner spectrum to the stable-rank numerator, and justify the assumed sample-size scaling of the perturbation.

2. Revisit Theorem 3.3 to handle the random denominator in cosine similarity, distinguish first and second moments of directional derivatives, specify an eigenvector sign convention, and state any required non-degeneracy assumptions.

3. Provide a clearer connection between Hessian-eigenspace truncation and the layerwise matrix-rank constraint used by FedLoRU. Direct pairwise client-alignment or eigenspace-overlap experiments would be useful.

4. Qualify the convergence claim as factor-space stationarity unless model-space stationarity can be established. Please also clarify the treatment of partial participation and the bounded-factor assumption.

5. Analyze the separate averaging of $A$ and $B$, including factor non-identifiability, and compare it with at least one alternative aggregation strategy.

6. Strengthen the empirical validation by varying local sample size and client count independently, evaluating stable rank at common model checkpoints, reporting uncertainty in the main tables, and providing complete communication accounting.